

# Spring phytoplankton communities of the Labrador Sea (2005-2014): pigment signatures, photophysiology and elemental ratios

Glaucia M. Fragoso[1], Alex J. Poulton[2], Igor M. Yashayaev[3], Erica J. H. Head[3], Duncan A. Purdie[1]

5  [1]Ocean and Earth Science, University of Southampton, National Oceanography Centre Southampton, Southampton UK.
[2]Ocean Biogeochemistry and Ecosystems, National Oceanography Centre; Southampton UK.
[3]Ocean and Ecosystem Science Division, Department of Fisheries and Oceans, Canada, Bedford Institute of Oceanography.

Correspondence to: Glaucia Fragoso, glaucia.fragoso@noc.soton.ac.uk



**Abstract.** The Labrador Sea is an ideal region to study the biogeographical, physiological and biogeochemical implications of phytoplankton communities due to sharp transitions of distinct water masses across its shelves and the central basin, intense nutrient delivery due to deep vertical mixing during winters and continual inflow of Arctic, Greenland melt and Atlantic waters. In this study, we provide a decadal assessment (2005-2014) of late spring/early summer phytoplankton communities from surface waters of the Labrador Sea based on pigment markers and CHEMTAX analysis, and their physiological and biogeochemical signatures. Diatoms were the most abundant group, blooming first in shallow mixed layers of haline-stratified Arctic shelf waters. Along with diatoms, chlorophytes co-dominated at the western end of the section (particularly in the polar waters of the Labrador Current (LC)), whilst *Phaeocystis* co-dominated in the east (modified polar waters of the West Greenland Current (WGC)). Pre-bloom conditions occurred in deeper mixed layers of the central Labrador Sea in May, where a mixed assemblage of flagellates (dinoflagellates, prasinophytes, prymnesiophytes, particularly coccolithophores, and chrysophytes/pelagophytes) occurred in low chlorophyll areas, succeeding to blooms of diatoms and dinoflagellates in thermally-stratified Atlantic waters in June. Light-saturated photosynthetic rates and saturation irradiance levels were higher at stations where diatoms were the dominant phytoplankton group (> 70 %), as opposed to stations where flagellates were more abundant (from 40 % up to 70 %). Phytoplankton communities from the WGC (*Phaeocystis* and diatoms) had lower light-limited photosynthetic rates, with little evidence of photo-inhibition, indicating greater tolerance to a high light environment. By contrast, communities from the central Labrador Sea (dinoflagellates and diatoms), which bloomed later in the season (June), appeared to be more sensitive to high light levels. Ratios of accessory pigments (AP) to total chlorophyll *a* (TChl*a*) varied according to phytoplankton community composition, with polar phytoplankton (cold-water related) having lower AP to TChl*a* ratios. Phytoplankton communities associated with polar waters (LC and WGC) also had higher and more variable particulate organic carbon (POC) to particulate organic nitrogen (PON) ratios, suggesting the influence of detritus from freshwater input, derived from riverine, glacial and/or sea-ice meltwater. Long-term observational shifts in phytoplankton communities were not assessed in this study due to the short temporal frame (May to June) of the data. Nevertheless, these results have provided a baseline of current distributions and an evaluation of the biogeochemical role of spring phytoplankton communities in the Labrador Sea, which will improve our understanding of potential long-term responses of phytoplankton communities in high-latitude oceans to a changing climate.

**Keywords**

Phytoplankton communities, CHEMTAX, hydrography, photophysiology, biogeochemistry, Labrador Sea



## 1. Introduction

Marine phytoplankton form a taxonomically and functionally diverse group, with different requirements and modes of acquisition of light and nutrients, as well as strategies for resource competition and predation defence (Acevedo-Trejos et al., 2015; Bonachela et al., 2015; Falkowski, 2004). Thus, marine phytoplankton communities are structured by the overall
fitness of individuals within species assemblages with respect to a variety of factors, including the physical setting, nutrient and light availability, dispersal, predation and competition for resources (Litchman and Klausmeier, 2008). Over large scales, environmental heterogeneity selects for phytoplankton assemblages, which creates biogeographical patterns of abundance, composition, traits and diversity distributions of phytoplankton communities in the global ocean (Barton et al., 2013; Follows et al., 2007; Hays et al., 2005).


Phytoplankton communities also impact the structuring of marine ecosystems due to their functional community role in biogeochemical cycling, efficiency of carbon transport to deeper waters, palatability and transfer of energy to higher trophic levels. Diatoms, for example, are assumed to be the major contributor to the biological pump (Smetacek et al., 2004), large *Phaeocystis* spp. (> 100 µm) colonies are apparently not grazed as efficiently as diatoms due to the exudation of mucilage
(Haberman et al., 2003) and some cyanobacteria are able to fix nitrogen, which can provide a significant amount of nitrogen to the oligotrophic regions of the ocean (Barton et al., 2013; Tyrrell, 1999). Distinct phytoplankton assemblages have been reported to influence differently particulate (Martiny et al., 2013a, 2013b; Smith and Asper, 2001) and dissolved elemental stoichiometry (C:N:P)(Weber and Deutsch, 2010), the drawdown of gases (Arrigo, 1999; Tortell et al., 2002) and the efficiency of carbon export (Guidi et al., 2009; Le Moigne et al., 2015). Patterns of phytoplankton stoichiometry appear to be
consistent phylogenetically and within higher taxonomic levels (Ho et al., 2003; Quigg et al., 2003), although they may vary according to nutrient supply ratios (Bertilsson et al., 2003; Rhee, 1978), as well as phenotypically within species across the same population (Finkel et al., 2006). However, detritus and dead plankton material also influence overall particulate C:N:P ratios in the ocean, which complicates the interpretation of *in situ* observations of phytoplankton elemental stoichiometry (Martiny et al., 2013a).


The sub-Arctic North Atlantic is a complex system with contrasting hydrography that structures plankton communities within distinct biogeographical provinces (Fragoso et al., 2016; Head et al., 2003; Li and Harrison, 2001; Platt et al., 2005; Sathyendranath et al., 1995, 2009). The Labrador Sea is a particularly interesting region to study the biogeographical and biogeochemical implications of phytoplankton communities due to the sharp transitions of distinct water masses across its
shelves and basin (Yashayaev, 2007). Biogeographical regions of the Labrador Sea shape zooplankton (Head et al., 2000, 2003) and phytoplankton community composition (Fragoso et al., 2016), cell size (Platt et al., 2005) and bio-optical properties (Cota, 2003; Lutz et al., 2003; Platt et al., 2005; Sathyendranath et al., 2004; Stuart et al., 2000), as well as the seasonality of phytoplankton blooms (Frajka-Williams and Rhines, 2010; Lacour et al., 2015; Wu et al., 2007, 2008). More



recently, Fragoso et al. (2016) showed that the biogeography of phytoplankton communities in the Labrador Sea is shaped by
specific species as indicators of Atlantic or Arctic waters, emphasising the potential importance of using phytoplankton composition as indicators of water masses.

Phytoplankton communities within a biogeographical region are subject to similar environmental conditions, such as temperature (Bouman et al., 2003), nutrient concentration (Browning et al., 2014) and irradiance (Arrigo et al., 2010), and
these, along with community composition (Bouman et al., 2005), affect their overall  photophysiological response. It has been suggested that irradiance (light levels and day length) and temperature are the primary factors that influence phytoplankton photophysiology in high-latitude Arctic/Atlantic waters, including the Labrador and Barents seas and Baffin Bay (Platt et al., 1982; Rey, 1991; Subba Rao and Platt, 1984), while the influence of phytoplankton composition on photophysiological patterns has not been investigated thoroughly.


Quantification of marine phytoplankton community composition, for large numbers of samples, is challenging if the time-consuming methods of microscopic identification and enumeration are employed. Moreover, small cells ($< 5\mu m$) are difficult to identify and count using light microscopy. To overcome these problems, quantification and analyses of phytoplankton pigments by high performance liquid chromatography (HPLC) has been widely used to monitor phytoplankton community
distributions over large temporal and spatial scales (e.g., Aiken et al., 2009; Peloquin et al., 2013; Platt et al., 2005). Photosynthetic pigments (chlorophylls and carotenoids) are frequently used to identify taxonomic and functional groups. Pigment-based chemotaxonomy can be used to determine phytoplankton classes (Coupel et al., 2012, 2015; Gonçalves-Araujo et al., 2012), functional cell sizes (Aiken et al., 2009; Platt et al., 2005; Poulton et al., 2006) and assemblage dominance using accessory pigment ratios (Fragoso and Smith, 2012). The  interpretation of the pigment data is not always
straightforward, however, since some pigments are shared by several algal groups and can change according to local nutrient and light conditions (DiTullio et al., 2007; van Leeuwe and Stefels, 1998, 2007). The chemotaxonomic tool, CHEMTAX (CHEMical TAXonomy), provides a valuable approach to estimate phytoplankton class abundances when used in conjunction with microscopic information (Irigoien et al., 2004; Mackey et al., 1996; Wright et al., 1996). CHEMTAX has the advantage of providing more information about phytoplankton classes than individual diagnostic pigments or ratios, and
has been used widely to investigate phytoplankton biogeography on regional scales (Muylaert et al., 2006; Wright and Van den Enden, 2000) and globally (Swan et al., 2015).

In this study, we investigate the multi-year (2005-2014) distributions of late spring and early summer (May to June) phytoplankton communities in the various hydrographic settings across the shelves, slopes and deep basin of the Labrador
Sea based on phytoplankton pigments.  In addition, we examine the overall photophysiological and biogeochemical traits associated with these phytoplankton communities. Long-term analyses of phytoplankton communities and their potential biogeochemical and physiological signatures are needed to comprehensively understand current conditions and to project





possible responses of these communities to climate change. The results presented here will improve our understanding of potential long-term changes of phytoplankton communities in high-latitude oceans and provide a baseline description of the

current distributions and biogeochemical traits of phytoplankton communities in the Labrador Sea with which future observations can be compared.

## 2. Methods

### 2.1 Study area

The Labrador Sea is a high latitude marginal sea located in the northwestern part of the Atlantic Ocean and is a transition

zone of the Arctic and sub-Arctic ecosystems. It is bounded by Davis Strait to the north, Newfoundland to the south, the Labrador Coast of Canada to the west and Greenland to the east (Fig 1). The bathymetry of the Labrador Sea can be subdivided into the wide continental shelf and relatively gentle continental slope on its western side (the Labrador Shelf, > 500 km and < 250 m deep), narrow shelf and very steep continental slope on the eastern side (the Greenland Shelf, < 100 km and < 2500 m deep) and the deep basin (> 3000 m deep) confined by the continental slopes (Fragoso et al., 2016).


The central deep basin (> 3000 m) of the Labrador Sea contains a counter-clockwise flow and is comprised of a mixture of, mostly, relatively warm and salty waters originating from the Atlantic, which is mainly identified as the Irminger Current (IC) and cold fresh water, originating from the Arctic *via* the surrounding shelves (Fig 1). The inshore branch of the Labrador Current (LC) overlies the Labrador Shelf and includes Arctic water originating from Baffin Bay and the Canadian

Arctic Archipelago *via* Davis Strait and from Hudson Bay *via* Hudson Strait, together with inputs of melting sea ice, which originate locally or from farther north. The main branch of the Labrador Current flows along the Labrador slope from north to south and is centred at the 1000 m contour.  It is composed of a mixture of Arctic water from Baffin Bay *via* Davis Strait and the branch of the West Greenland Current that flows west across the mouth of Davis Strait. The West Greenland Current (WGC), which flows from south to north on the Greenland shelf and along the adjacent slope, is a mixture of  cold, low

salinity Arctic water exiting the Nordic Seas with the East Greenland Current (EGC) (Yashayaev, 2007),  together with sea ice and glacial melt waters (Fig 1). More detailed descriptions of the hydrography of the Labrador Sea can be found elsewhere (Fragoso et al. 2016, Head et al.2013, Yashayaev and Seidov 2015, Yashayaev 2007).

### 2.2 Sampling

Data for this study were obtained from stations along the AR7W Labrador Sea repeat hydrography line (World Ocean

Circulation Experiment Atlantic Repeat 7-West section, for details see Fragoso et al., 2016), which runs between Misery Point on the Labrador coast (through Hamilton Bank on the Labrador Shelf) and Cape Desolation on the Greenland coast. Stations were sampled for over a decade (2005-2014) by scientists from the Canadian Department of Fisheries and Oceans




during late spring and/or early summer (Table 1). Fixed stations were sampled on the AR7W section, across shelves and in the deep central basin, as well as some additional non-standard stations (Fig. 1).


Vertical profiles of temperature and salinity were measured with a Seabird CTD system. Water samples were collected using 10-L Niskin bottles mounted on the rosette frame. Mixed layer depths were calculated from the vertical density ($\sigma_\Theta$) distribution and defined as the depth where $\sigma_\Theta$ changes by 0.03 kg m$^{-3}$ from a stable surface value (~10 m) (Weller and Plueddemann, 1996). A stratification index (SI) was calculated as the seawater density difference (between 10 m to 60 m)

normalised to the equivalent difference in depth.

Water samples from the surface (near-surface) layer (< 10 m) were collected (0.5 L–1.5 L) for the determination of chlorophyll *a*, accessory pigments, nutrients, particulate organic carbon (POC) and nitrogen (PON) analysis, and for primary production measurements. Bulk chlorophyll *a* concentration was measured after extraction from filters in 90 % acetone at -

20°C for approximately 24 hours and fluorescence was determined using a Turner Designs fluorometer (Holm-Hansen et al., 1965). Samples for detailed pigment analysis were filtered onto 25 mm glass fibre filters (GF/F Whatman Inc., Clifton, New Jersey) and immediately flash frozen in liquid nitrogen and kept frozen in a freezer (at -80° C) until analysis in the laboratory. Nutrient samples were analysed at sea (within 12 h of collection) on the SEAL AutoAnalyser III.  Samples for POC and PON were filtered (0.25–1 L) onto 25 mm pre-combusted GF/F filters and frozen (-20° C) and returned to the

laboratory for later analysis. In the laboratory samples were oven-dried (60 ℃) for 8-12 hours, stored in a dessicator, pelletelised in pre-combusted tin foil cups and analysed using a Perkin Elmer 2400 Series CHNS/O analyser  as described in Pepin and Head (2009).

## 2.3 Pigment analysis

Pigments (chlorophyll *a* and accessory pigments) were quantified using reverse-phase (Beckman C18, 3 μm Ultrasphere column), High-Performance Liquid Chromatography (HPLC) according to the procedure described in Stuart and Head (2005), known as the "BIO method". Prior to analysis, pigments were extracted by homogenizing the frozen filters in 1.5 mL 95 % acetone, grinding the filters using a motorized grinder, centrifuging to remove the solids and taking an aliquot of the supernatant, which was buffered by dilution with 0.5 M aqueous ammonium acetate at a ratio of 2:1 before injection on to

the column. The samples were run using a gradient elution method, with methanol, aqueous ammonium acetate and ethyl acetate as solvents (Stuart and Head 2005). Pigment peaks were identified and quantified by their retention times and absorbance or fluorescence signals, by comparison with those of pure pigments (Stuart and Head, 2005). A list of pigments identified and quantified for this study is included in Table 2.



### 2.4 Pigment interpretation

The CHEMTAX software (version 1.95, Mackey et al., 1996) was used to estimate ratios of abundance of distinct micro-algal classes to total chlorophyll *a* from *in situ* pigment measurements. The software utilises a factorization program that uses "best guess" ratios of accessory pigments to chlorophyll *a* that are derived for different classes from the literature and marker pigment concentrations of algal groups that are known to be present in the study area. This program uses the steepest descent algorithm to obtain the best fit to the data based on assumed pigment to chlorophyll *a* ratios. The initial matrices are optimized by generating 60 further pigment ratio tables using a random function (RAND in Microsoft Excel) as described in Coupel et al. (2015). The results of the six best output matrices (with the smallest residuals, equivalent to 10 % of all matrices) were used to calculate the averages of the abundance estimates and final pigment ratios. The following pigment chosen for CHEMTAX analysis were: 19-butanoyloxyfucoxanthin (BUT19), 19-hexanoyloxyfucoxanthin (HEX19), alloxanthin (ALLOX), chlorophyll *a* (CHLA), chlorophyll *b* (CHLB), chlorophyll *c*3 (CHLC3), fucoxanthin (FUCOX), peridinin (PERID), prasinoxanthin (PRASINO) and zeaxanthin (ZEA).

One of the main assumptions of the CHEMTAX method is that pigment ratios remain constant across the subset of samples that are being analysed (Swan et al., 2015). To satisfy this assumption, *a priori* analysis was performed, where pigment data were sub-divided into groups using cluster analysis (Bray-Curtis similarity) and each group was processed separately by the CHEMTAX program (Table 3). This approach was used because distinct phytoplankton communities have been observed in the Labrador Sea (Fragoso et al., 2016) so that the ratio of accessory pigment to chlorophyll *a* probably varies within different water masses across the Labrador Sea (LC, IC and WGC). Pigment concentration data (BUT19, HEX19, ALLOX, CHLA, CHLB, CHLC3, FUCOX, PERID, PRASINO and ZEA) were standardized and fourth-root transformed before being analysed.

A first cluster analysis on transformed pigment data identified five major groups having 60 % similarity between samples. Clusters included stations partially located: 1) on the shelves, where FUCOX dominated at a few stations (I), 2) in the eastern part of the Labrador Sea, where most stations had high relative concentrations of FUCOX and CHLC3 (II), 3) in the central Labrador Sea, where a few stations had higher proportions of FUCOX, HEX19 and PERID (III), 4) on the western part of the section, where CHLB and FUCOX were the main pigments at most stations (IV) and 5) in the central Labrador Sea, where most stations had a mixture of pigments (FUCOX, CHLC3, HEX19, CHLB, PERID and others) (V) (Fig 2). The other main requirement of the CHEMTAX method is that information about the phytoplankton taxonomy is used to assure that the pigment ratios are applied and interpreted correctly (Irigoien et al., 2004). To satisfy this requirement, initial pigment ratios were carefully selected and applied to each cluster to adjust the pigments to the appropriate classes according to microscopic observations (Fragoso et al., 2016) and literature information (see Table 3). Pigment ratio tables were based on the literature in waters having comparable characteristics to the Labrador Sea, such as Baffin Bay (Vidussi et al., 2004), the





Beaufort Sea (Coupel et al., 2015) and the North Sea (Antajan et al., 2004; Muylaert et al., 2006) or from surface (high light) field data (Higgins et al., 2011) (Table 3).


Prasinophytes were separated into "prasinophyte type 1", which contains PRASINO and "prasinophyte type 2", such as *Pyramimonas* and *Micromonas*, with the latter previously found lacking PRASINO but containing ZEA in North Water Polynya (Canadian Arctic) (see Vidussi et al., 2004). Both genera were observed in light microscope counts in Labrador Sea samples (Fragoso et al., 2016) and *M. pusilla* has been observed in the Beaufort Sea (Coupel et al., 2015), and was found to

be one of the main pico-eukaryotes in the North Water Polynya (Canadian Arctic) from April to July of 1998 (Lovejoy et al., 2002). In addition to prasinophytes type 2, ZEA is also the major accessory pigment of cyanobacteria, such as *Synechococcus* spp., which has been previously observed in the Labrador Sea, particularly in Atlantic waters (Li et al., 2006), and which is also a minor pigment in chlorophytes (Vidussi et al., 2004). Because of its association with the warmer Atlantic waters, it was assumed that cyanobacteria were absent from very cold waters, such as the Labrador Current

(Fragoso et al., 2016). Prasinophytes contain CHLB, but so do chlorophytes (Vidussi et al., 2004), which were observed in large numbers with the microscope. Dinoflagellates were separated into those species that contain PERID, such as *Heterocapsa* sp. and *Amphidium* (Coupel et al., 2015; Higgins et al., 2011) and those that do not, such as *Gymnodinium* spp. (here defined as Dino-2 class as in Higgins et al. (2011)), but which may contain CHLC3, BUT19, HEX19 and FUCOX. Dinoflagellates were observed in lower concentrations in the eastern Labrador Sea, so that Dino-2 was assumed absent from

this area (clusters I & II in Table 3). Cryptophytes (Cryptophycea in Table 3) are the only group to contain ALLOX.

Prymnesiophytes were divided into three groups: 1) *Phaeocystis pouchetii*, which was observed in high concentrations in the eastern Labrador Sea (Fragoso et al., 2016) (clusters I & II, Table 3); 2) Prymnesiophyte HAPTO-7 (as in Higgins et al. (2011)), associated with *Chrysocromulina* spp. previously observed in the western Labrador Sea (in the Labrador Current, this study) (cluster IV, Table 3) and HAPTO-6 (as in Higgins et al. (2011)), which included the coccolithophores,

particularly *E. huxleyi* associated with Atlantic waters (central-eastern region of the Labrador Sea) (clusters I, II, III and V, Table 3). *Phaeocystis pouchetii* occurred in waters having low HEX19 and BUT19 concentrations and high CHLC3 and FUCOX concentrations (cluster II, Fig. 2). Similar pigment compositions were found in *Phaeocystis globosa* blooms in Belgian Waters (Antajan et al., 2004; Muylaert et al., 2006) and high ratios of CHLC3 to CHLA were, previously, used to

identify *Phaeocystis pouchetii* in the Labrador Sea (Stuart et al., 2000). Thus, CHLC3 and FUCOX were the only pigments that could be used to represent this species. In addition to CHLC3 and FUCOX, HAPTO-7 included HEX19 and HAPTO-6 included HEX19 and BUT19 as in Higgins et al. (2011). Chrysophytes and pelagophytes, such as *Dictyocha speculum* have high ratios of BUT19 to CHLA (Coupel et al., 2015; Fragoso and Smith, 2012). Finally, diatoms were identified as containing high FUCOX:CHLA ratios (Vidussi et al., 2004) (Table 3).




## 2.5 Photosynthesis versus irradiance incubations

Water samples were spiked with $^{14}$C-bicarbonate and incubated in a light box under 30 different irradiance levels (from 5-2000 µmol quanta m$^{-2}$ s$^{-1}$) at *in situ* temperature for 2 to 3 hours to measure parameters derived from photosynthesis versus irradiance (P-E) curves as described by Stuart et al. (2000). Measurements were fitted to the equation of Harrison and Platt

(1986) to determine the initial slope of the P-E curve, also known as the photosynthetic efficiency ($\alpha^B$), the maximum photosynthetic rate normalized to chlorophyll biomass ($P_m{}^B$), the light intensity approximating the onset of saturation ($E_k$), the saturation irradiance ($E_s$) and the photo-inhibition parameter ($\beta$).

## 2.6 Statistical analysis

### 2.6.1 Phytoplankton-derived POC estimation

Fragoso et al. (2016) found a significant linear relationship between phytoplankton carbon calculated from phytoplankton cell counts and POC data using results from 2011-2014 surveys in the Labrador Sea (POC = 1.01POC$_{phyto}$ + 240.92; r$^2$ = 0.47; n = 44; p < 0.0001). To estimate phytoplankton-derived carbon (POC$_{phyto}$) concentration (as opposed to total POC, which includes detritus and heterotrophic organisms), regression analysis was performed (POC$_{phyto}$ = 38.9Chl$a$; r$^2$ = 0.9; n =

41; p < 0.0001) using the carbon calculated from cell counts (derived from Fragoso et al., 2016) and measurements of total chlorophyll $a$: this expression was then applied to estimate POC$_{phyto}$ for stations where phytoplankton cell counts were not available (2005-2010).

### 2.6.2 Phytoplankton community structure

Phytoplankton community structure derived from pigment concentrations was investigated using PRIMER-E (v7) software (Clarke and Warwick, 2001). Chlorophyll $a$ concentrations derived for each algal class resulting from CHEMTAX analysis were standardized (converted to percentage values) to obtain their relative proportions, which were fourth-root transformed to allow the least abundant groups to contribute to the analysis. Similarity matrices were generated from Bray-Curtis similarity for cluster analysis. A SIMPER (SIMilarity PERcentages) routine with a cut off of 90 % cumulative contribution

to the similarity was used to reveal the contributions of each class to the overall similarity within clusters. One-way ANOSIM was also applied to determine whether taxonomic compositions of the clusters were significantly different.

A redundancy analysis (RDA) using the CANOCO 4.5 software (CANOCO, Microcomputer Power, Ithaca, NY) was performed to analyse the effects of environmental factors on the Labrador Sea phytoplankton community structure as

described in Fragoso et al. (2016). Data were log-transformed and forward-selection (*a posteriori* analysis) identified the subset of environmental variables that significantly explained the taxonomic distribution and community structure when





analysed individually ($\lambda_1$, marginal effects) or when included in a model where other forward-selected variables were analysed together ($\lambda a$, conditional effects). A Monte Carlo permutation test (n=999, reduced model) was applied to test the statistical significance ($p < 0.05$) of each of the forward-selected variables.

## 3. Results

### 3.1 Environmental variables

Environmental parameters, as well as chlorophyll *a* (Chl*a*) concentrations varied noticeably along the southwest-northeast section of the Labrador Sea (Fig. 3). The shelf and slope regions (LSh, LSl, GSl, GSh) had colder and fresher waters (< 3 °C and < 33.5, respectively) compared to the central basin (CB), where surface waters were saltier (> 33.5) and warmer (> 3 °C), particularly in 2005, 2006, 2012 and 2014 (> 5 °C) (Fig. 3b, c). Shelf waters that were colder and fresher were the most highly stratified (> $5 \times 10^{-3}$ kg m$^{-4}$), particularly on the Labrador Shelf (> $15 \times 10^{-3}$ kg m$^{-4}$), whereas waters from the CB were less well stratified (< $5 \times 10^{-3}$ kg m$^{-4}$), except at those stations where waters were slightly warmer than usual (> 5°C) during 2005, 2012 and 2014 (Fig. 3d). Chl*a* concentrations were higher (> 4 mg Chl*a* m$^{-3}$) at stations where waters were more highly stratified, particularly on the shelves (Fig. 3e). Nitrate, phosphate and silicate concentrations were inversely related to Chl*a* concentration, being lower (< 5, 0.5, 3 µmol L$^{-1}$, respectively) on the shelves and, during some years, in the CB (e.g. 2012), where blooms formed (Fig. 3e-h). POC:PON ratios were also higher (> 8) at most stations in shelf and slope waters and at a few stations in the CB during 2009 and 2011 (Fig. 3i). Shelf waters mostly had higher silicate:nitrate (Si(OH)$_4$:NO$_3^-$) ratios (> 1) than the CB, particularly the LSh (Fig. 3j). Labrador Sea surface waters usually had nitrate:phosphate (NO$_3^-$:PO$_4^{3-}$) < 16, although NO$_3^-$:PO$_4^{3-}$ were higher in the CB than in shelf regions (> 10) (Fig. 3k).

### 3.2 CHEMTAX interpretation and group distributions

A cluster analysis of algal classes derived from CHEMTAX results revealed clusters of stations at various similarity levels (Fig. 4). ANOSIM one-way pairwise analysis between clusters suggested that they were significantly different in pigment algal composition (p = 0.001), although pairwise analysis of clusters C3a and C3b showed that these groups overlapped (more similar composition, R statistic = 0.33) than other clusters, which were clearly separated (R statistic values approached 1) (see Clarke and Warwick, 2001). The first division occurred at 61 %, separating three main clusters (A, B and C) (Fig. 4a). Cluster C was subdivided at 65 % resulting in clusters C1, C2 and C3 (Fig. 4a). A third division (similarity of 73 %) occurred at cluster C3 resulting in two other clusters C3a and C3b (Fig. 4a). Overall, six functional clusters (A, B, C1, C2, C3a and C3b) represented the distinct phytoplankton communities occurring in the Labrador Sea (Fig. 4a). These communities generally occupied different regions of the Labrador Sea, namely the Labrador Shelf/Slope (west, mainly Cluster C3a), Central Basin (middle, mainly Clusters C2 or C3b) and the Greenland Shelf/Slope (east, mainly Clusters C3a, A, B) (Fig. 4b,c).



Chl*a* concentrations were higher at stations where diatoms were especially dominant (Fig. 4b,c). Diatoms were the most abundant phytoplankton group in Labrador Sea waters, particularly at stations on the shelves, where communities were

sometimes composed of almost 100 %  diatoms (clusters A and C1) (Fig. 4b,c). Diatoms were also abundant at (or near to) the Greenland Shelf, where *Phaeocystis* was co-dominant (cluster B) and at (or near to) the Labrador Shelf in the west section, where chlorophytes were the second most abundant group (cluster C3a). Likewise, diatoms were dominant in the central Labrador Sea in some years (2008, 2012 and 2014, cluster C2), where dinoflagellates were also dominant (Fig. 4b,c). Most stations in the central basin had low Chl*a* concentrations and high diversity of algal groups (cluster C3b), with mixed

assemblages of diatoms, dinoflagellates and other flagellates (Fig. 4b,c). The positions of fronts, usually characterised by sharp transitions in phytoplankton communities varied from year to year, but were generally located near the continental slopes (Fig. 4c).

### 3.3 Phytoplankton distributions and environmental controls

Distributions of surface phytoplankton communities in the Labrador Sea during spring and early summer (2005-2014) varied

according to the water mass distributions across the shelves and central basin of the Labrador Sea. Potential temperatures and salinities also varied among these water masses (Fig. 5a). In general, chlorophytes and diatoms (cluster C3a) were associated with the inshore branch of the Labrador Current (LC), on the Labrador Shelf, where the surface waters were fresher (salinity < 33.5), colder (temperature < 2°C) and least dense ($\sigma_\Theta$ of most stations approximately < 26.5 kg m$^{-3}$) (Fig. 5a). Mixed assemblages (cluster C3b), as well as blooms (chlorophyll average = 4 mg Chl*a* m$^{-3}$) of dinoflagellates and

diatoms (cluster C2) were associated with the warmer (temperature > 3°C), saltier (salinity > 34) and denser ($\sigma_\Theta$ of most stations < 27 kg m$^{-3}$) Atlantic water mass, and the Irminger Current (IC) (Fig. 5a). *Phaeocystis* (cluster B) dominated in waters of the West Greenland Current (WGC), which had intermediate temperatures (mostly 0-4°C) and salinities (33-34.5) when compared to those of the LC and IC (Fig. 5a).

Redundancy analysis (RDA) was used to investigate the hydrographic variables that explained the variance (explanatory variables) in the phytoplankton communities based on pigment analyses. The ordination diagram (Fig. 5b) revealed that most stations from distinct clusters were concentrated within one quadrant, where arrows representing environmental variables in the same or opposite directions of the clusters of stations suggest positive or negative correlations proportional to the length of the arrow. The first axis (x-axis) of the analysis, which explained most of the variance (eigen-value = 25.7 % of species

data and 83.5 % of species-environment relation, Table 4), clearly shows that the phytoplankton communities respond strongly to spatial aspects of the data (Fig. 5b). Thus, stations in Arctic waters were to the left of the y-axis (low nutrient concentrations, temperature and salinity values), while stations located in Atlantic waters were to the right (opposite trend, Fig. 5b). Diatoms and chlorophytes (cluster C3a, upper left quadrant of Fig. 5b) were associated with lower salinities and temperatures and highly stratified waters. *Phaeocystis* and diatoms (cluster B, lower left quadrant of Fig. 5b) were associated

with waters where nutrient concentrations (mainly nitrate, but also phosphate and silicate) were relatively low (average



nitrate concentration from cluster B < 3 μM, Table 5). In Atlantic waters, temporal aspects of the data were also observed (upper and lower right quadrants (Fig. 5b)). Thus, mixed assemblages (cluster C3b) were associated with higher nutrient concentrations (pre-bloom conditions in Atlantic waters, upper right quadrant), whereas dinoflagellates and diatoms (cluster C2) were associated with warmer and saltier waters, resembling blooming conditions in Atlantic waters induced by thermal

stratification (lower right quadrant of Figure 5b). Summed, the canonical axes explained 99.8 % of the variance (axis 1, p = 0.002; all axes, p = 0.002) (Table 4), which means that the environmental variables included in this analysis explained almost 100 % of the variability.

Forward selection showed that five of the six environmental factors (silicate, temperature, salinity, nitrate and phosphate)
included in the analysis best explained the variance in the phytoplankton community distributions when analysed together (Table 4). When all variables were analysed together (conditional effects, referred to as $\lambda_a$ in Table 4), silicate was the most significant explanatory variable ($\lambda_a$ = 0.2, p = 0.001), followed by temperature ($\lambda_a$ = 0.05, p = 0.001), salinity ($\lambda_a$ = 0.02, p = 0.002), nitrate concentration ($\lambda_a$ = 0.01, p = 0.016) and phosphate ($\lambda_a$ = 0.02, p = 0.002) (Table 4). SI was the only explanatory variable that had no significance in explaining the distribution of phytoplankton communities (Table 4).

**3.4 Phytoplankton distributions and environmental controls**

Particulate organic carbon (POC) collected on filters can include organic carbon from a variety of sources, such as phytoplankton, bacteria, zooplankton, viruses and detritus (Sathyendranath et al., 2009). Assuming that phytoplankton associated organic carbon, as estimated from phytoplankton cell volumes ($POC_{phyto}$) is strongly correlated with Chl$a$ values, the proportion of $POC_{phyto}$ should increase in eutrophic waters, which usually occurs with high Chl$a$ and POC
concentrations, and that it should be lower in oligotrophic waters. Indeed, our results showed higher proportions of $POC_{phyto}$ (> 60 %) in waters with higher POC concentrations (Fig. 6a). However, there were stations where POC levels were high and where the contribution of $POC_{phyto}$ was low, suggesting that there may have been other sources of POC (e.g. detritus).

The relationships between $POC_{phyto}$ and POC:PON also varied among the different phytoplankton community types (Fig. 6).
In general, stations in shelf regions, which have higher inputs of Arctic and glacial melt waters (lower salinity values), where diatoms co-dominated with chlorophytes in the west and east (cluster C3a) or with *Phaeocystis* in the east (cluster B), had higher and more variable values for POC:PON ratios than did stations influenced by Atlantic water (Fig. 6b). Some  shelf stations had relatively high proportions of $POC_{phyto}$ to total POC, suggesting that phytoplankton community growth dominated by diatoms and chlorophytes (cluster C3a) contributed more to the total POC (most stations from cluster C3a had
$POC_{phyto}$ > 50 %) (Fig. 6b). On the other hand, some shelf stations, particularly the one dominated by diatoms and *Phaeocystis* (cluster B) had high POC:PON ratios (> 10), with low $POC_{phyto}$ contributions, suggesting an increased contribution of detritus to the total POC (Fig. 6c). Stations influenced by Atlantic waters had generally lower contributions of $POC_{phyto}$, with most stations having POC:PON ratios < 6.6 (Fig. 6c).





### 3.5 Physiological patterns

Accessory pigments (AP) versus total chlorophyll *a* (TChl*a*) scatterplot from surface waters of the Labrador Sea showed a log-log linear relationship (Fig. 7). The slopes of these relationships varied within temperature (Fig. 7a) and among the distinct phytoplankton communities (Fig. 7b). Phytoplankton communities in cold waters (of Arctic origin), such as those co-dominated by diatoms and *Phaeocystis* in the east and diatoms and chlorophytes in the west, had a lower accessory pigments to TChl*a* ratio (logAP:logTChl*a*) (slope = 0.86 and 0.89, respectively) (Fig. 7b). Furthermore, communities from

warmer waters (Irminger Current from Atlantic origin), particularly those co-dominated by diatoms and dinoflagellates had higher ratios of logAP:logChl*a* (slope = 1.03) (Fig. 7b). Slopes of the logAP to logTChl*a* relationships were not statistically different among the different communities (ANCOVA, p > 0.05), except for those communities co-dominated by diatoms and *Phaeocystis* (cluster B), which had a slope that was statistically different from the others (ANCOVA, p= 0.016).

Photosynthetic parameters differed among the different phytoplankton communities. *Phaeocystis* and diatom communities near Greenland (cluster B) had the lowest photosynthetic efficiencies (average $\alpha^B$= $6.8 \times 10^{-2}$ µg C µg Chl*a* h$^{-1}$ W m$^{-2}$) with relatively high saturation irradiances (average $E_s$= 62 ± 11 W m$^{-2}$) and little photo-inhibition ($\beta = 4 \times 10^{-4}$ µg C µg Chl*a* h$^{-1}$ W m$^{-2}$) (Table 5). By contrast, phytoplankton communities dominated by diatoms and chlorophytes typically found in the Labrador Current (cluster C3a) were highly susceptible to photo-inhibition ($\beta = 16 \times 10^{-4}$ µg C µg Chl*a* h$^{-1}$ W m$^{-2}$), had

lower light saturation irradiances ($E_s$ = 35 W m$^{-2}$) and higher photosynthetic efficiencies ($\alpha^B$ = $9.2 \times 10^{-2}$ µg C µg Chl*a* h$^{-1}$ W m$^{-2}$) (Table 5). Phytoplankton communities in Atlantic waters (clusters C3b and C2) had the highest levels of photoprotective pigments, such as those used in the xanthophyll cycle (diadinoxanthin (DD) + diatoxanthin (DT)):Chl*a* > 0.07, particularly those communities co-dominated by diatoms and dinoflagellates (cluster C2) from stratified Atlantic waters (Table 5). These communities were the most susceptible to photo-inhibition ($\beta = 29 \times 10^{-4}$ µg C µg Chl*a* h$^{-1}$ W m$^{-2}$),

had the highest ratios of photoprotective pigments to Chl*a* ((DD+DT):Chl*a* = 0.12 ± 0.01), and the highest maximum photosynthetic rates ($P_m^B$ = 3.2 ± 0.4 µg C µg Chl*a* h$^{-1}$ W m$^{-2}$) (Table 5).

### 4. Discussion

### 4.1 Biogeography of phytoplankton communities in the Labrador Sea

In this study, our assessment of phytoplankton pigments from surface waters during spring/early summer of the Labrador Sea based on a decade of observations showed that the distribution of phytoplankton communities varied primarily with the distinct waters masses (Labrador, Irminger and Greenland Currents). There were three regions where major blooms (Chl*a* concentrations > 3 mg Chl*a* m$^{-3}$) occurred. For all three blooms, diatoms were predominant; however, they co-dominated with 1) chlorophytes in the west (mostly in the Labrador Current), 2) *Phaeocystis* in the east in the West Greenland Current




and 3) dinoflagellates in the central basin of the Labrador Sea, once waters were thermally-stratified. While diatoms bloomed in shallower mixed layers (< 33 m, Table 5), a more diverse community was found in most years in deeper mixed layers (> 59 m) in the central basin, resembling pre-bloom conditions. These patterns are similar to those seen in other shelf and basin regions of Arctic/subarctic waters (Coupel et al., 2015; Fujiwara et al., 2014; Hill et al., 2005). It is well known that diatoms tend to dominate in high-nutrient regions of the ocean due to their high growth rates, while their low surface

area to volume ratios mean that they do not do as well as nano- or picoplankton in low nutrient conditions  (Gregg et al., 2003; Sarthou et al., 2005). In the Labrador Sea, deep winter mixing (200 – 2300 m) provides nutrients to the near surface layers, which supports phytoplankton spring blooms, particularly of diatoms once light becomes available (Fragoso et al., 2016; Harrison et al., 2013; Yashayaev and Loder, 2009).

Chlorophytes were the second most abundant phytoplankton group, particularly in the central-western part of the Labrador Sea, but often occurring in the east as well. Chlorophytes are thought to contribute  1-13 % of total Chl$a$ in the global ocean (Swan et al., 2015) and to inhabit transitional regions, where nutrient concentrations become limiting for diatoms, but are not persistently low enough to prevent growth due to nutrient limitation, as occurs in the oligotrophic gyres  (Gregg and Casey, 2007; Gregg et al., 2003; Ondrusek et al., 1991). The Labrador Shelf is a dynamic region during springtime, where melting

sea ice in May provides a local freshwater input  (Head et al., 2003).  Melting sea ice provides intense stratification and shallow mixed layers for the phytoplankton and thus access to light, which promotes rapid growth of cold Arctic/ice-related phytoplankton near the sea ice shelf (Fragoso et al., 2016), and which likely stimulates the succession from large diatoms to smaller phytoplankton forms, such as chlorophytes, as nutrients become exhausted. Chlorophytes, as well as Prasinophytes such as *Pyramimonas*, a genus found in high abundances in surface Labrador Shelf waters, might also be associated with

melting sea ice, given that they have been found blooming (chlorophyll concentration ~ 30 mg Chl$a$ m$^{-3}$) in low salinity melt waters (salinity = 9.1) under the Arctic pack-ice (Gradinger, 1996).

Dinoflagellates were associated with the Irminger Current, where they were occasionally found blooming with diatoms in the warmer, stratified Atlantic waters of the central basin. These blooms dominated by dinoflagellates and Atlantic diatom

species, such as *Ephemera planamembranacea* and *Fragilariopsis atlantica*, start later in the season (end of May or June) as thermal stratification develops in the central Labrador Sea (Frajka-Williams and Rhines, 2010, Fragoso et al., 2016). Transition from diatoms to dinoflagellates has been well-documented in the North Atlantic between spring and summer, which occurs because dinoflagellates can use mixotrophic strategies to alleviate nutrient limitation as waters become warmer, highly stratified and nutrient-depleted (Barton et al., 2013; Head and Pepin, 2010; Head et al., 2000; Henson et al.,

2012; Leterme et al., 2005). The North Atlantic Oscillation index (NAO) and sea surface temperature (Zhai et al., 2013) appear to influence the relative proportions of diatoms and dinoflagellates as well as the variability in the start date of the North Atlantic bloom. A negative winter phase of NAO is associated with weaker northwest winds over the Labrador Sea and reductions in the depth of winter mixing and supply of nutrients to the upper layers (Drinkwater and Belgrano, 2003).



Vertical stability, thermal stratification and the initiation of the spring bloom tend to occur earlier under negative NAO conditions and the proportion of dinoflagellates in the warmer, more nutrient-limited waters may be higher (Zhai et al., 2013). Unfortunately, it was not possible to investigate the influence of NAO on the relative contribution of dinoflagellates and diatoms in the Labrador Sea section of the North Atlantic in this study, given that the sampling period varied from early/mid-May to mid/late-June. On the other hand, abundances of dinoflagellates appeared to be higher in warmer waters (> 5°C), suggesting that the communities were shifting from diatoms to dinoflagellates as the water became stratified and nutrient concentrations decreased.

*Phaeocystis* and diatoms bloom together in the eastern central Labrador Sea (Fragoso et al., 2016; Frajka-Williams and Rhines, 2010; Harrison et al., 2013; Head et al., 2000; Stuart et al., 2000; Wolfe et al., 2000). This is a region with high eddy kinetic energy during spring (Chanut et al., 2008; Frajka-Williams et al., 2009; Lacour et al., 2015), which causes the accumulation of low-salinity surface waters from the West Greenland Current and confines elevated levels of phytoplankton biomass, presumably of *Phaeocystis* and diatoms, in buoyant freshwater layers (Fragoso et al., 2016). Mesoscale eddies may stimulate growth of *Phaeocystis* and diatoms by inducing partial stratification at irradiance levels that are optimal for their growth, but too low for their competitors (blooms in these eddies usually start in April). Lacour et al. (2015) showed that irradiance levels estimated from satellite-derived photosynthetically active irradiance (PAR) and mixed layer depth climatologies are similar for thermally and haline-stratified spring blooms in the Labrador Sea. Nonetheless, these authors recognise the need for *in situ* measurements to confirm whether Labrador Sea spring blooms, presumably composed of distinctive phytoplankton communities, respond in the same manner to light-mixing regimes. The ability of *Phaeocystis* to grow under dynamic light irradiances explains why they are often found in deeper mixed layers, such as those found in Antarctic polynyas (Arrigo, 1999; Goffart et al., 2000), although this genus can also occur in shallow mixed layers, such as those found close to ice edges (Fragoso and Smith, 2012; Le Moigne et al., 2015).

Mesoscale eddies are also often associated with elevated zooplankton abundances (Frajka-Williams et al., 2009; Yebra et al., 2009). In the Labrador Sea, lower grazing rates have been observed in blooms dominated/co-dominated by colonial *Phaeocystis*, which are often located in these eddies and which may, in turn, explain why this species is dominant (Head and Harris, 1996; Wolfe et al., 2000). Although the exact mechanism that facilitates *Phaeocystis* growth in the north-eastern region of the Labrador Sea is not clear, it is evident that blooms of this species are tightly linked to mesoscale eddies, and that this relationship needs further investigation to better explain their regular reoccurrence in these waters.

### 4.2 Phytoplankton composition and related biogeochemistry

Particulate organic carbon (POC) and nitrogen (PON) concentrations, as well as the molar ratio of POC:PON varied within distinct hydrographic zones, indicating the presence of different biogeochemical provinces in the Labrador Sea. A canonical Redfield ratio of 6.6 for POC:PON appears to represent the global average (Redfield, 1958), although regional variations on





the order of 15 to 20 % have also been reported (Martiny et al., 2013a). The POC:PON appears to be closer to the Redfield ratio of 6.6 in productive sub-Arctic/Arctic waters, such as the northern Baffin Bay (Mei et al., 2005), the north-eastern Greenland shelf (Daly et al., 1999), and in Fram Strait and the Barents Sea (Tamelander et al., 2012). Crawford et al. (2015),

however, recently reported very low POC:PON ratios in oligotrophic Arctic waters of the Beaufort Sea and Canada Basin, where depth-integrated values of the POC:PON ratio were ~ 2.65, much lower than those in more productive domains, such as the sub-Arctic central Labrador Sea (POC:PON ~ 4).

In this study, highly productive surface waters of Arctic origin (near or on the shelves) had higher phytoplankton-derived

particulate organic carbon ($POC_{phyto}$ > 43 % from total POC), as well as higher and more variable POC:PON ratios (average > 6.9) compared with stations influenced by Atlantic water (average POC:PON < 6.3, $POC_{phyto}$ > 35 %). Diatoms have been suggested to contribute to larger phytoplankton-derived POC in Arctic/sub-Arctic waters (Crawford et al., 2015). The Labrador Shelf region, where blooms are generally dominated by large Arctic/ice-related diatoms (Fragoso et al., 2016), had relatively high contributions of $POC_{phyto}$ (> 50 %) to the total POC, even though smaller phytoplankton forms, such as

chlorophytes, were also abundant. Low POC:PON ratios, as well as low $POC_{phyto}$ concentrations were associated with Atlantic waters, which had greater contributions of flagellates (particularly before bloom initiation). Similar findings were reported by Crawford et al. (2015), where low $POC_{phyto}$ was associated with larger contributions of flagellates (< 8 μm) in oligotrophic Arctic waters, such as the Beaufort Sea and Canada Basin. Crawford et al. (2015) also considered that POC:PON ratios might have been reduced by the presence of heterotrophic microbes (bacteria, flagellates and ciliates) since

these microorganisms have POC:PON ratios lower than the canonical Redfield ratio of 6.6 (Lee and Fuhrman, 1987; Vrede et al., 2002). Bacteria and other heterotrophic organisms were not quantified in our study, although Li and Harrison (2001) showed that bacterial biomass from surface waters was 62 % greater (average from 1989 to 1998 =13.8 mg C m$^{-3}$) in the central region than in shelf areas of the Labrador Sea.

Changes in POC:PON may be related to the physiological status of phytoplankton and/or community structure. In the North Water Polynya (Baffin Bay), POC:PON ratios during phytoplankton blooms increased between spring (5.8) and summer (8.9) as phytoplankton responded to nitrate starvation by producing N-poor photo-protective pigments (Mei et al., 2005). Daly et al. (1999) also found high POC:PON ratios (~8.9) in Arctic surface waters dominated by diatoms on the north-eastern Greenland shelf, which were attributed to nutrient limitation. Atlantic waters appear to have an excess of nitrate

compared with Arctic waters (Harrison et al., 2013), which could explain why phytoplankton from Atlantic Waters had lower POC:PON ratios. Conversely, Arctic-influenced waters on or near the shelves had higher $Si(OH)_4:NO_3^-$ and lower $NO_3^-:PO_4^{3-}$ than those in the central basin in this study, which could also have contributed to the observed high POC:PON ratios.





A few stations in shelf waters of the Labrador Sea also had remarkably high POC:PON ratios (> 10), and low $POC_{phyto}$

contributions, suggesting high contributions of detritus. These waters probably receive higher inputs of Arctic and glacial ice

melt, which could introduce POC from external sources. Hood et al. (2015) showed that POC export from glaciers is large,

particularly from the Greenland Ice Sheet and it occurs in suspended sediments derived from glacier meltwater. High

POC:PON ratios (> 10), particularly in waters where *Phaeocystis* were abundant, may also be linked to the mucilaginous

matrix of the *Phaeocystis* colonies (Palmisano et al., 1986). The mucopolysaccharide  appears to contain excess carbon,

particularly when nutrients start to become depleted and colonies become senescent (Alderkamp et al., 2007; Wassmann et

al., 1990).

### 4.3 Physiological parameters of distinct phytoplankton communities

Accessories pigments (AP) are assumed to have a ubiquitous, global, log-log linear relationship with chlorophyll *a* in aquatic

environments (Trees et al., 2000). This linear relationship is often used as an index of quality-control in pigment analysis,

which are required due to uncertainties of the quantitative comparability of data among surveys, related to differences in

analytical procedures and sample storage methods used in different laboratories. In the current study, the slope of AP to total

chlorophyll *a* (TChl*a*) on a logarithm scale (Fig. 7) passed the quality control criteria of slopes ranging from 0.7 to 1.4 and $r^2$

> 0.90 as applied in previous studies (e.g., Aiken et al., 2009; Peloquin et al., 2013; Thompson et al., 2011) and were within

the range observed throughout worldwide aquatic systems (slope from 0.8 to 1.3 compared to 0.86 to 1.03 observed in our

study) (Trees et al., 2000). An interesting trend was found where phytoplankton pigment ratios varied clearly within distinct

communities in the Labrador Sea. According to our data, phytoplankton communities found in colder waters (of Arctic

origin) had lower accessory pigments ratios to total chlorophyll *a* ratio (logAP:logTChl*a*) (slope = 0.86) when compared to

communities from warmer waters (Irminger Current from Atlantic origin) (slope = 1.03). Changes in logAP:logTChl*a* as a

function of phytoplankton community composition has been observed before, when Stramska et al. (2006) related a higher

slope of logAP:logTChl*a* to dinoflagellates dominating during summer in northern polar Atlantic waters as opposed to lower

ratios of flagellates occurring in spring. Trees et al. (2000) and Aiken et al. (2009) also reported lower logAP:logTChl*a*

(slope < 1.00) in oligotrophic waters dominated by picoplankton as opposed to higher ratios in upwelling waters where

microplankton, particularly diatoms, were dominant.


Environmental parameters, such as nutrients and light conditions, have also been suggested to influence logAP:logTChl*a*

regardless of community composition (Trees et al., 2000). Nonetheless, in our study, these two parameters, analysed as

nitrate and silicate concentrations and Stratification Index, did not vary with logAP:logTChl*a* (data not shown) as opposed to

temperature. Phytoplankton community distributions varied clearly according to temperature with *Phaeocystis* occurring in

colder Arctic waters and dinoflagellates in warmer Atlantic waters. Although both communities were co-dominated by

diatoms (relative abundance > 70 %), logAP:logTChl*a* varied considerably, suggesting that either 1) diatom species from

both Arctic and Atlantic waters varied intrinsically in pigment composition, or 2) temperature had a physiological effect on





the logAP:logTChl$a$ ratio. Despite the observed trend of logAP:logTChl$a$ varying with temperature, a direct temperature-induced effect in logAP:logTChl$a$ is unknown.


The variation in photosynthetic parameters in the distinct phytoplankton biogeographical provinces demonstrated how each phytoplankton community responds to environmental conditions. Harrison and Platt (1986) found that the photophysiology of phytoplankton from the Labrador Sea is influenced by temperature and irradiance. Nonetheless, phytoplankton composition may also influence the values of the photosynthetic parameters. Light-saturated photosynthetic rates and

saturation irradiances, for instance, were higher at stations where diatoms were dominant (> 70 %), as opposed to stations where flagellates were more abundant (from 40 % up to 70 %). Similar findings were reported by (Huot et al., 2013), who observed that light-saturated photosynthetic rates in the Beaufort Sea (Arctic Ocean) were higher for communities composed of large cells, presumably diatoms, compared to smaller flagellates.

Polar phytoplankton communities from shelf waters (east *versus* west) observed in this study had distinctive photo-physiological characteristics. Diatom/*Phaeocystis* dominated communities from waters located near Greenland (east) had low photosynthetic efficiency (average $\alpha^B = 6.8 \times 10^{-2}$ µg C µg Chl$a$ h$^{-1}$ W m$^{-2}$) and high light-saturation irradiances ($E_s = 62$ W m$^{-2}$), while diatom/chlorophyte dominated communities on or near the Labrador Shelf (west) had the reverse ($E_s = 35$ W m$^{-2}$, $\alpha^B = 9.2 \times 10^{-2}$ µg C µg Chl$a$ h$^{-1}$ W m$^{-2}$). Low photosynthetic efficiency and high light-saturation irradiance in

diatom/*Phaeocystis* dominated communities mean that photosynthetic rates were relatively low at high light intensities, although photo-inhibition was low ($\beta = 4 \times 10^{-4}$ µg C µg Chl$a$ h$^{-1}$ W m$^{-2}$). *Phaeocystis antarctica*, widespread in Antarctic waters, relies heavily on photo-damage recovery, such as D1 protein repair (Kropuenske et al., 2009), which could explain how these communities overcome photo-inhibition. These results were inconsistent with those reported by Stuart et al. (2000); however, who found a higher photosynthetic efficiency ($\alpha^B$) for a population dominated by *Phaeocystis* near

Greenland compared with that of a diatom dominated population near the Labrador coast. Stuart et al. (2000) attributed the higher $\alpha^B$ to the smaller cell size of *Phaeocystis*. In the current study, however, chlorophytes were present in high concentrations on the Labrador Shelf, which may explain the discrepancy between these results.

Phytoplankton communities from Atlantic waters (co-dominated by diatoms and dinoflagellates) were highly susceptible to

photo-inhibition ($\beta = 29 \times 10^{-4}$ µg C µg Chl$a$ h$^{-1}$ W m$^{-2}$) compared with the other communities in the Labrador Sea. Days are longer and solar incidence is higher in June as compared to May at these latitudes (Harrison et al., 2013), which, in this study, was the time when dinoflagellates bloomed in the central Labrador Sea as a consequence of thermal stratification, which explains the sensitivity of this community to high-light levels. To cope with photo-damage, this phytoplankton community appeared to increase the levels of photoprotective pigments, such as those used in the xanthophyll cycle

(diadinoxanthin (DD) + diatoxanthin (DT)). These communities also had high diatoxanthin levels compared with the polar phytoplankton communities, suggesting that the community was experiencing higher light intensities (Moisan et al., 1998).





## 4.4 Phytoplankton communities assessed by HPLC and CHEMTAX methods

Phytoplankton pigments and CHEMTAX methods provide information about phytoplankton community structure, and are
especially powerful when used in conjunction with microscopic analysis (light and high resolution scanning electron
microscopy) (Coupel et al., 2012, 2015; Eker-develi et al., 2012; Muylaert et al., 2006), cytometry (Devilla et al., 2005;
Fujiki et al., 2009) and molecular techniques (Not et al., 2007; Piquet et al., 2014; Zhang et al., 2015). However, choosing
the pigment markers in the CHEMTAX analysis and interpreting the results are not always straightforward and, therefore,
conclusions need to be drawn with caution. Many environmental factors (primarily light and nutrients) (DiTullio et al., 2007;
Henriksen et al., 2002; van Leeuwe and Stefels, 1998, 2007), in addition to natural variability among species from the same
classes or even strains of the same species (Zapata et al., 2004) affect accessory pigment levels and ratios to chlorophyll *a*,
which could introduce some uncertainties when applying CHEMTAX. Thus, phytoplankton abundances determined using
CHEMTAX represent approximations based on pigment distributions. These limitations can, however, be lessened when this
technique is combined with existing knowledge of main phytoplankton groups occurring in the samples through microscopic
identification.

A number of studies have used CHEMTAX methods to determine phytoplankton community structure in Arctic/subarctic
waters (Coupel et al., 2012, 2015; Lovejoy et al., 2007; Piquet et al., 2014; Vidussi et al., 2004; Zhang et al., 2015). Spring
phytoplankton communities from the Labrador Sea have already been investigated in detail (Fragoso et al., 2016), although
the analysis did not include most nano- and pico-flagellates (except cryptophytes and *Phaeocystis pouchetii*) and were done
over only four years (2011-2014) at selected stations along the L3 (=AR7W) transect. Here, we have combined
phytoplankton information from Fragoso et al. (2016) with additional pigment analyses. Although cross comparison among
these two techniques (carbon biomass estimated from microscopic counts *versu*s algal class chlorophyll *a* estimated from
CHEMTAX) should not be expected to give exactly equivalent results, given that most flagellates observed in the pigment
analysis were not counted under the microscope, some comparability should be possible, at least for the larger cells (e.g.
diatoms).

*Phaeocystis* (r$^2$ = 0.79) and diatom (r$^2$ = 0.74) biomasses were well correlated when carbon biomasses estimated from
microscopic counts were compared with CHEMTAX-derived algal chlorophyll *a* biomass (data not shown). This confirms
that using chlorophyll *c3* was appropriate for detecting and quantifying *Phaeocystis* biomass in the Labrador Sea. Similar
associations have been observed in *Phaeocystis* from boreal waters (e.g. *P. pouchetii* but *P. globosa* as well; Antajan et al.,
2004; Muylaert et al., 2006; Stuart et al., 2000; Wassmann et al., 1990), while other pigment markers have been used
elsewhere, e.g. 19- hexanoyloxyfucoxanthin, which is characteristic of *Phaeocystis antarctica* in austral polar waters (Arrigo
et al., 2010, 2014; Fragoso and Smith, 2012; Fragoso, 2009). Dinoflagellates gave a poor correlation between biomass



estimates made using the two methods ($r^2$ = 0.12, data not shown) possibly because some heterotrophic dinoflagellates, which lack photosynthetic pigments, might have been included in the microscopic counts from Fragoso et al. (2016). Cryptophyte biomass estimates were not related (data not shown), likely because their biomass was underestimated in microscopic counts.

## 5. Conclusions

In this study, we have provided a geographical description of phytoplankton community structure in spring and early summer surface waters of the Labrador Sea based on pigment data from over a decade of sampling (2005-2014). Phytoplankton communities and their photophysiological and biogeochemical signatures were assessed using CHEMTAX, so that a geographical baseline of the major phytoplankton groups has been provided for the central Labrador Sea and its adjacent continental shelves. In spite of interannual variability (due to differences in survey dates and natural variability),

spring phytoplankton communities showed distinct spatial variations from east to west and there were clear temporal differences between May and June. The main conclusions of our study are that: 1) diatoms contributed most to the chlorophyll *a* in waters where phytoplankton blooms were observed (> 3 mg Chl*a* m$^{-3}$); while other groups (chlorophytes, dinoflagellates and *Phaeocystis*) were geographically segregated within distinct hydrographical zones; 2) a diverse mixed assemblage dominated by flagellates from several groups occurred in low chlorophyll, pre-bloom conditions in the central

Labrador Sea; and 3) different phytoplankton communities had different ratios of accessory pigments to total chlorophyll *a*; and 4) POC:PON ratios were influenced by phytoplankton community composition, as well as freshwater input of allochthonous carbon in shelf waters which have nearby sources (e.g. melting glacial and sea- ice and river outflows).

Marine phytoplankton respond rapidly to changes in the ocean, and their responses directly impact local marine food webs

and global biogeochemical cycles. Climate-driven processes modify the factors, including light availability, nutrient input and grazing pressure that shape phytoplankton physiological traits and alter community structure (Litchman et al., 2012; Montes-Hugo et al., 2009). High latitude seas, particularly the Labrador Sea, are regions that are extremely vulnerable to climate change and often show similar patterns of variability on interannual and decadal scales across the entire domain (Yashayaev and Seidov, 2015; Yashayaev et al., 2015) and they could, therefore, be subject to rapid shifts in phytoplankton

biomass, size and species composition. Although climate-induced responses of phytoplankton communities in vulnerable regions are difficult to predict, the long-term observations of these communities reported here and the analysis of their biogeochemical and physiological signatures are important in order to create a baseline for evaluation of changes that will occur in the future, as greenhouse gas-driven warming continues in this and other regions of the global ocean.



**Acknowledgements**

We would like to thank Sinhue Torres-Valdes, Mark Stinchcombe and Brian King (National Oceanography Centre) for collecting the samples and providing the nutrient and hydrographic data from JR302 cruise. Many thanks to Carol Anstey, Jeff Anning and Tim Perry (Bedford Institute of Oceanography) for collecting and analysing nutrients concentrations, phytoplankton pigments and photosynthetic measurements. The officers and crew of the *CCGS Hudson* and *RSS James*
*Clark Ross* and the support of technicians and scientists from all cruises in analysing and providing the chlorophyll, particulate organic carbon and nitrogen and hydrographic data are also acknowledged. We would like to thank Simon Wright (Australian Antarctic Division) for providing us with a copy of the CHEMTAX software v.1.95. We are grateful to the reviewers who offered useful suggestions to improve the manuscript. G.M.F. was funded by a Brazilian PhD studentship, Science without Borders (CNPq, 201449/2012-9). This research was also partially funded by UK Ocean Acidification, a
National Environment Research Council grant (NE/H017097/1) through an added value award to AJP.

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



**Table 1. Research cruises, sampling dates and number of samples per cruise (n) where pigment data were collected in the Labrador Sea during early spring and late summer (2005-2014).**

| Cruise | Dates | Year | n |
|---|---|---|---|
| HUD-2005-16 | 29 May - 3 June | 2005 | 25 |
| HUD-2006-019 | 23 May - 31 May | 2006 | 12 |
| HUD-2007-011 | 11 May - 21 May | 2007 | 32 |
| HUD-2008-009 | 22 May - 29 May | 2008 | 25 |
| HUD-2009-015 | 18 May - 23 May | 2009 | 26 |
| HUD-2010-014 | 14 May - 24 May | 2010 | 27 |
| HUD-2011-009 | 11 May - 17 May | 2011 | 33 |
| HUD-2012-001 | 3 June - 11 June | 2012 | 30 |
| HUD-2013-008 | 9 May - 21 May | 2013 | 27 |
| JR302 | 10 June - 24 June | 2014 | 16 |




**Table 2. List of phytoplankton pigments and their distributions in algae classes, abbreviations and formulas.**

| Abbreviation | Name | Characteristic of the pigment | Present in/ Index of/Formula | Ref. |
|---|---|---|---|---|
| PSC | Photosynthetic carotenoid | Light harvesting | All algae | |
| PPC | Photoprotective carotenoid | Photoprotection | All algae | |
| PPP | Photosynthetic pigment | Light harvesting | All algae | |
| BUT19 | 19'-butanoyloxyfucoxanthin | PSC | Prymnesiophytes and crysophytes | 1 |
| HEX19 | 19'-hexanoyloxyfucoxanthin | PSC | Diatoms, prymnesiophytes and some dinoflagellates | 2 |
| ALLOX | Alloxanthin | PPC | Cryptophytes | 1 |
| ACAROT | $\alpha$-carotene | PPC | Various | 1 |
| BCAROT | $\beta$-carotene | PPC | Various | 1 |
| CHLB | Chlorophyll $b$ | PPP | Chlorophytes, prasinophytes, euglenophytes | 1 |
| CHLC12 | Chlorophyll $c_1 + c_2$ | PPP | Diatoms, prymnesiophytes, dinoflagellates, chrysophytes and raphidophytes | 1 |
| CHLC3 | Chlorophyll $c_3$ | PPP | Prymnesiophytes, chrysophytes | 1 |
| CHLIDEA | Chlorophyllide $a$ | Degradation product of CHLA | Senescent phytoplankton | |
| DIADINOX | Diadinoxanthin | PPC | Diatoms, prymnesiophytes, dinoflagellates, chrysophytes and raphidophytes | 1 |
| DIATOX | Diatoxanthin | PPC | Diatoms, prymnesiophytes, dinoflagellates, chrysophytes and raphidophytes | 1 |
| FUCOX | Fucoxanthin | PSC | Diatoms, prymnesiophytes, raphidophytes and some dinoflagellates | 1 |
| CHLA | Chlorophyll $a$ | PPP | All phytoplankton except *Prochlorococcus* | 1 |
| PERID | Peridinin | PSC | Some dinoflagellates | 1 |
| PRASINO | Prasinoxanthin | PPC | Some prasinophytes | 1 |
| VIOLAX | Violaxanthin | PPC | Chlorophytes, prasinophytes and eustigmatophytes | 1 |
| ZEA | Zeaxanthin | PPC | Cyanobacteria, *Prochlorococcus*, chlorophytes | 1 |
| TChl$a$ | Total chlorophyll $a$ | | CHLA + CHLIDEA | |
| TC | Total carotenoids | Include all carotenoids | BUT19 + HEX19 + ALLOX + ACAROT + BCAROT + DIADINOX + DIATOX + FUCOX + PERID + PRASINO + VIOLAX + ZEA | |
| AP | Accessory pigments | Include all pigments except CHLA | TC + CHLB + CHLC12 + CHLC3 | |
| FUCOX/AP | Fucoxanthin to accessory pigments ratio | | FUCOX/AP | |

[1](Jeffrey et al., 1997), [2](Higgins et al., 2011).



**Table 3. Final ratio matrix of accessory pigment to chlorophyll *a* for distinct algal classes for each cluster group.**

| Region<br>Class / Pigment | I & II (Eastern Labrador Sea)<br>CHLB | CHLC3 | FUCOX | PERID | ZEA | ALLOX | BUT19 | HEX19 | PRASINO | CHLA | Ref |
|---|---|---|---|---|---|---|---|---|---|---|---|
| Prasinophyte 1 | 0.459 | 0 | 0 | 0 | 0 | 0 | 0 | 0 | 0.075 | 1 | 2 |
| Prasinophyte 2 | 0.650 | 0 | 0 | 0 | 0.008 | 0 | 0 | 0 | 0 | 1 | 2 |
| Chlorophyte | 0.168 | 0 | 0 | 0 | 0.040 | 0 | 0 | 0 | 0 | 1 | 2 |
| Dinoflagellates | 0 | 0 | 0 | 0.609 | 0 | 0 | 0 | 0 | 0 | 1 | 2,5 |
| Cryptophycea | 0 | 0 | 0 | 0 | 0 | 0.785 | 0 | 0 | 0 | 1 | 2 |
| *Phaeocystis* | 0 | 0.167 | 0.188 | 0 | 0 | 0 | 0 | 0 | 0 | 1 | 1 |
| HAPTO-6 | 0 | 0.199 | 0.270 | 0 | 0 | 0 | 0.021 | 1.261 | 0 | 1 | 4 |
| Chryso/Pelagophyte | 0 | 0.120 | 0.454 | 0 | 0 | 0 | 0.589 | 0 | 0 | 1 | 2 |
| Cyanobateria | 0 | 0 | 0 | 0 | 0.262 | 0 | 0 | 0 | 0 | 1 | 3 |
| Diatoms | 0 | 0 | 0.328 | 0 | 0 | 0 | 0 | 0 | 0 | 1 | 2 |
| **Region**<br>**Class / Pigment** | **III & V (Central Labrador Sea)**<br>CHLB | CHLC3 | FUCOX | PERID | ZEA | ALLOX | BUT19 | HEX19 | PRASINO | CHLA | Ref |
| Prasinophyte 1 | 0.316 | 0 | 0 | 0 | 0 | 0 | 0 | 0 | 0.129 | 1 | 2 |
| Prasinophyte 2 | 0.716 | 0 | 0 | 0 | 0.008 | 0 | 0 | 0 | 0 | 1 | 2 |
| Chlorophyte | 0.171 | 0 | 0 | 0 | 0.025 | 0 | 0 | 0 | 0 | 1 | 2 |
| Dinoflagellates | 0 | 0 | 0 | 0.681 | 0 | 0 | 0 | 0 | 0 | 1 | 2,5 |
| Dino-2 | 0 | 0.290 | 0.348 | 0 | 0 | 0 | 0.060 | 0.168 | 0 | 1 | 4 |
| Cryptophycea | 0 | 0 | 0 | 0 | 0 | 0.674 | 0 | 0 | 0 | 1 | 2 |
| HAPTO-6 | 0 | 0.081 | 0.202 | 0 | 0 | 0 | 0.018 | 1.549 | 0 | 1 | 4 |
| Chryso/Pelagophyte | 0 | 0.049 | 0.184 | 0 | 0 | 0 | 0.264 | 0 | 0 | 1 | 2 |
| Cyanobateria | 0 | 0 | 0 | 0 | 0.142 | 0 | 0 | 0 | 0 | 1 | 3 |
| Diatoms | 0 | 0 | 0.512 | 0 | 0 | 0 | 0 | 0 | 0 | 1 | 2 |
| **Region**<br>**Class / Pigment** | **IV (Western Labrador Sea)**<br>CHLB | CHLC3 | FUCOX | PERID | ZEA | ALLOX | BUT19 | HEX19 | PRASINO | CHLA | Ref |
| Prasinophyte 1 | 0.216 | 0 | 0 | 0 | 0 | 0 | 0 | 0 | 0.078 | 1 | 2 |
| Prasinophyte 2 | 1.081 | 0 | 0 | 0 | 0.012 | 0 | 0 | 0 | 0 | 1 | 2 |
| Chlorophyte | 0.113 | 0 | 0 | 0 | 0.045 | 0 | 0 | 0 | 0 | 1 | 2 |
| Dinoflagellates | 0 | 0 | 0 | 0.785 | 0 | 0 | 0 | 0 | 0 | 1 | 2,5 |
| Dino-2 | 0 | 0.028 | 0.049 | 0 | 0 | 0 | 0.018 | 0.040 | 0 | 1 | 4 |
| Cryptophycea | 0 | 0 | 0 | 0 | 0 | 0.703 | 0 | 0 | 0 | 1 | 2 |
| HAPTO-7 | 0 | 0.030 | 0.389 | 0 | 0 | 0 | 0 | 1.218 | 0 | 1 | 4 |
| Chryso/Pelagophyte | 0 | 0.056 | 0.470 | 0 | 0 | 0 | 0.613 | 0 | 0 | 1 | 2 |
| Diatoms | 0 | 0 | 0.343 | 0 | 0 | 0 | 0 | 0 | 0 | 1 | 2 |

[1](Antajan et al., 2004), [2](Vidussi et al., 2004), [3](Muylaert et al., 2006), [4](Higgins et al., 2011), [5](Coupel et al., 2015)



**Table 4 – Results of the Redundancy Analyses (RDA) with the effects, eigenvalues and percentages of variance explained used in the analysis. Marginal ($\lambda_1$) and conditional effects ($\lambda_a$) refers to the absolute and additional effects, respectively, of the environmental variable (s) used in the RDA analysis after the automatic forward selection. Explanatory variables are temperature (°C), salinity, nitrate ($NO_3^-$), phosphate ($PO_4^{3-}$), silicate ($Si(OH)_4$) ($\mu mol\ L^{-1}$) and Stratification Index (SI) (kg m$^{-4}$). Significant p-values (p < 0.05) represents the variables that significantly explains the variation in the analyses.**

| Marginal Effects | | Conditional Effects | | | |
|---|---|---|---|---|---|
| **Variable** | $\lambda_1$ | **Variable** | $\lambda_a$ | **P** | **F** |
| $Si(OH)_4$ | 0.2 | $Si(OH)_4$ | 0.2 | 0.001 | 61.65 |
| $NO_3^-$ | 0.19 | Temperature | 0.05 | 0.001 | 17.3 |
| $PO_4^{3-}$ | 0.17 | Salinity | 0.02 | 0.002 | 6.94 |
| Salinity | 0.09 | $NO_3^-$ | 0.01 | 0.016 | 4.31 |
| Temperature | 0.07 | $PO_4^{3-}$ | 0.02 | 0.002 | 7.22 |
| SI | 0.06 | SI | 0.01 | 0.153 | 1.72 |

| Axes | 1 | 2 | 3 | 4 | Total variance |
|---|---|---|---|---|---|
| **Eigen-values** | 0.257 | 0.042 | 0.005 | 0.003 | 1 |
| **Taxa-environment correlations** | 0.676 | 0.404 | 0.321 | 0.245 | |
| **Cumulative percentage variance** | | | | | |
| of species data | 25.7 | 29.9 | 30.3 | 30.7 | |
| of species-environment relation | 83.5 | 97.2 | 98.8 | 99.8 | |

| | | |
|---|---|---|
| **Sum of all eigenvalues** | | 1 |
| **Sum of all canonical eigenvalues** | | 0.307 |

Test of significance of first canonical axis: eigen-value = 0.257; F-ratio = 84.938; P-value = 0.002.

Test of significance of all canonical axis: trace = 0.307; F-ratio = 18.184; P-value = 0.002.





**Table 5 – Average, standard errors and number of observations (in parenthesis) of environmental and biological variables of each cluster group. MLD = mixed layer depth, SI= Stratification index, NO₃⁻ = nitrate, PO₄³⁻ = phosphate, Si(OH)₄ = silicate, DT= diatoxanthin, DD= diadinoxanthin, POC= particulate organic carbon, PON= particulate organic nitrogen, POC_phyto = phytoplankton-derived particulate organic carbon, $\alpha^B$ = initial slope of the photosynthesis-irradiance curve, $P_m^B$ = maximum normalised photosynthesis, $E_k$ = half-saturation irradiance, $E_s$ = saturation irradiance.**


| | Cluster A | | Cluster B | | Cluster C3a | | Cluster C3b | | Cluster C2 | | Cluster C1 | |
|---|---|---|---|---|---|---|---|---|---|---|---|---|
| Temperature (°C) | 2.8 ± 0.6 | (17) | 2.0 ± 0.3 | (46) | 1.6 ± 0.2 | (62) | 3.4 ± 0.2 | (92) | 4.8 ± 0.3 | (32) | 1.4 ± 0.9 | (4) |
| Salinity | 33.4 ± 0.4 | (17) | 33.7 ± 0.1 | (46) | 33.1 ± 0.2 | (62) | 34.1 ± 0.1 | (92) | 34.4 ± 0.1 | (32) | 33.0 ± 0.8 | (4) |
| MLD (m) | 32.2 ± 10.6 | (17) | 32.6 ± 3.4 | (46) | 31.2 ± 3.6 | (62) | 59 ± 7.4 | (92) | 29.8 ± 3.0 | (32) | 16.0 ± 2.1 | (4) |
| SI × 10⁻³ (kg m⁻⁴) | 9.1 ± 1.5 | (17) | 6.3 ± 0.8 | (46) | 10.7 ± 1.1 | (62) | 5.0 ± 0.7 | (92) | 6.1 ± 0.8 | (31) | 6.6 ± 4.3 | (4) |
| NO₃⁻ (µmol L⁻¹) | 2.9 ± 1.1 | (17) | 2.7 ± 0.5 | (46) | 3.4 ± 0.6 | (58) | 8.4 ± 0.5 | (83) | 3.7 ± 0.7 | (32) | 3.8 ± 3.4 | (4) |
| Si(OH)₄ (µmol L⁻¹) | 2.2 ± 0.7 | (17) | 2.8 ± 0.3 | (46) | 3.5 ± 0.3 | (58) | 5.4 ± 0.2 | (83) | 3.0 ± 0.4 | (32) | 2.3 ± 1.7 | (4) |
| PO₄³⁻ (µmol L⁻¹) | 0.3 ± 0.1 | (17) | 0.3 ± 0 | (45) | 0.4 ± 0 | (55) | 0.7 ± 0 | (79) | 0.3 ± 0 | (32) | 0.4 ± 0.2 | (4) |
| Si(OH)₄:NO₃⁻ | 6.0 ± 3.2 | (14) | 3.6 ± 1.3 | (37) | 8.5 ± 2.5 | (54) | 1.1 ± 0.2 | (82) | 1.6 ± 0.3 | (32) | 3.9 ± 2.2 | (4) |
| NO₃⁻:PO₄³⁻ | 8.2 ± 2.0 | (11) | 5.2 ± 0.7 | (45) | 5.9 ± 0.8 | (55) | 11.4 ± 0.5 | (79) | 8.7 ± 0.8 | (32) | 5.5 ± 3.5 | (4) |
| Chlorophyll *a* (mg Chl*a* m⁻³) | 3.8 ± 1.1 | (17) | 5.5 ± 0.7 | (45) | 7.7 ± 0.7 | (59) | 2.0 ± 0.2 | (91) | 4.0 ± 0.3 | (31) | 8.8 ± 4.8 | (4) |
| DT:(DT+DD) | 0.01±0.006 | (16) | 0.02±0.01 | (44) | 0.04±0.01 | (62) | 0.10±0.01 | (92) | 0.08±0.01 | (32) | 0.02±0.02 | (4) |
| (DD+DT):Chl*a* | 0.08±0.02 | (17) | 0.03±0.004 | (46) | 0.04±0.003 | (62) | 0.07±0.004 | (92) | 0.12±0.01 | (32) | 0.07±0.02 | (4) |
| POC (mg C m⁻³) | 245 ± 45 | (4) | 498 ± 38 | (27) | 533 ± 30 | (45) | 234 ± 18 | (63) | 512 ± 46 | (15) | 393 ± 296 | (2) |
| PON (mg N m⁻³) | 39 ± 8 | (4) | 65 ± 4 | (27) | 74 ± 4 | (45) | 38 ± 3 | (64) | 83 ± 9 | (15) | 42 ± 29 | (2) |
| POC_phyto (%) | 23.0 ± 2.6 | (4) | 49.2 ± 5.8 | (26) | 60.9 ± 3.9 | (44) | 33.3 ± 1.3 | (64) | 36.0 ± 3.0 | (15) | 37.8 ± 0.9 | (2) |
| POC:PON | 6.5 ± 0.6 | (4) | 7.8 ± 0.4 | (27) | 7.5 ± 0.3 | (45) | 6.6 ± 0.2 | (64) | 6.2 ± 2.0 | (15) | 8.6 ± 1.1 | (2) |
| $\alpha^B$ × 10⁻²(µg C µg Chl*a* h⁻¹ W m⁻²) | - | | 6.8 ± 2 | (9) | 9.2 ± 2 | (10) | 7.1 ± 1 | (18) | 7.1 ± 1 | (4) | - | |
| $P_m^B$ (µg C µg Chl*a* h⁻¹ W m⁻²) | - | | 2.9 ± 0.4 | (9) | 2.3 ± 0.3 | (10) | 2.3 ± 0.1 | (18) | 3.2 ± 0.4 | (4) | - | |
| $E_k$ (W m⁻²) | - | | 60 ± 11 | (9) | 29 ± 4 | (10) | 38 ± 3 | (18) | 46 ± 3 | (4) | - | |
| $E_s$ (W m⁻²) | - | | 62 ± 11 | (9) | 35 ± 6 | (10) | 43 ± 4 | (18) | 56 ± 4 | (4) | - | |
| $\beta$ × 10⁻⁴(µg C µg Chl*a* h⁻¹ W m⁻²) | - | | 4 ± 2 | (9) | 16 ± 7 | (10) | 10 ± 4 | (18) | 29 ± 10 | (4) | - | |



**FIGURE LEGENDS**

**Figure 1- Map showing stations along the AR7W transect and additional stations sampled during late spring and early summer (2005–2014). The station positions are superimposed on a composite image of sea surface temperature for the last three weeks of May 2006 collected by the NOAA satellite (AVHRR). White patches represent ice (Labrador and Greenland coasts).**

**Figure 2. Percentage contribution of each pigment to the similarity of sampled stations in different clusters (I-V). Pigment abbreviations are described in Table 2.**

  **Figure 3 – Values for environmental variables (temperature, salinity, stratification index (SI)), concentrations of nutrients (nitrate ($NO_3^-$), silicate ($Si(OH)_4$), phosphate ($PO_4^{3-}$)), chlorophyll *a* and ratios between nutrients and for**
**particulate organic carbon (POC) to particulate organic nitrogen (PON) at individual stations sampled between 2005 and 2014 (y-axis) and distances from a fixed reference position in the Northeast Gulf of St Lawrence shown by the star in Figure 3a (x-axis). LSh = Labrador Shelf, LSl = Labrador Slope, CB = Central Basin, GSl = Greenland Slope, GSh = Greenland Shelf.**

**Figure 4. Dendrogram showing clustering of samples (a) and the proportion of chlorophyll *a* contributed by each phytoplankton class for each cluster (b). Spatial distribution of distinct phytoplankton communities (cluster groups) along the section, showing the distance from the star in Fig 3a) (c). Bubble size in (c) represents total chlorophyll *a* biomass (minimum = 0.3 mg Chl*a* m$^{-3}$ and maximum = 25 mg Chl*a* m$^{-3}$).**

**Figure 5- Positions of individual stations in relation to temperature (°C) and salinity (a) and redundancy analysis (RDA) ordination plot (b). The stations are colour-coded according to the cluster groups (see details in Figure 4). The TS plot (a) shows the approximate ranges of potential temperature (°C) and salinity of the Labrador Current (LC), the West Greenland Current (WGC) and the Irminger Current (IC). Arrows in (b) show the explanatory (environmental) variables used in the analysis.**


  **Figure 6- Relationship between particulate organic carbon (POC) and particulate organic nitrogen (PON) in a logarithmic scale, with the points (stations) as a function of phytoplankton-derived organic carbon content (POC$_{phyto}$/POC, %) (a), POC:PON *versus* salinity (b), phytoplankton-derived organic carbon content (POC$_{phyto}$/POC, %) *versus* the POC:PON ratio (c). The points (stations) in (b) and (c) are colour-coded according to the cluster groups**
**(see details in Figure 4). Solid lines in (b) and (c) show the C:N Redfield ratio of 6.6 and the dashed line in (c) shows where POC$_{phyto}$ contributes 50 % of the total POC.**

  **Figure 7. Relationship between total accessory pigments (mg AP m$^{-3}$) and total chlorophyll (mg TChl*a* m$^{-3}$) on a logarithmic scale, with the points (stations) according to temperature (a) and colour-coded according to**
**phytoplankton community cluster group (see details in Figure 4) (b).**





**Figure 1**

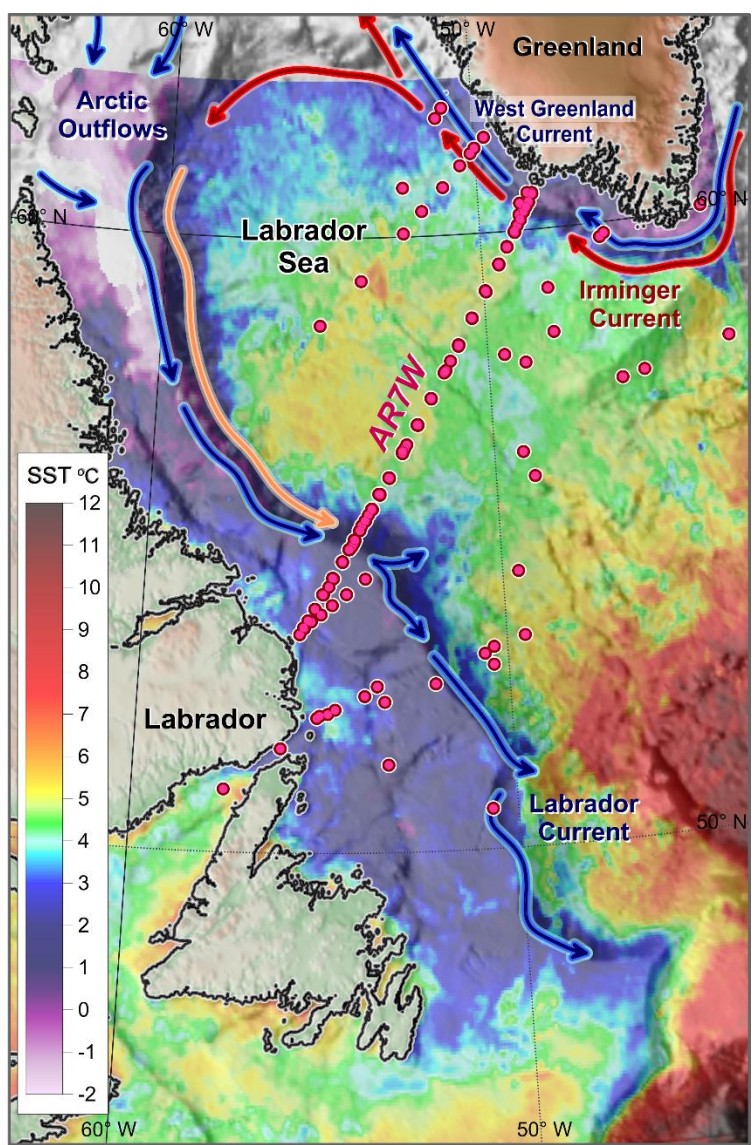



**Figure 2**

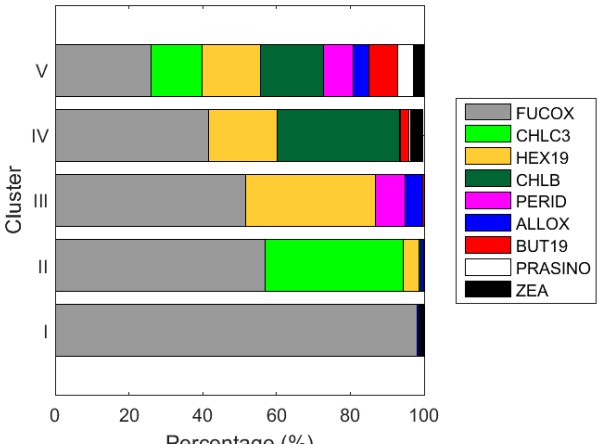





**Figure 3**

Distance from fixed reference position (km)




**Figure 4**

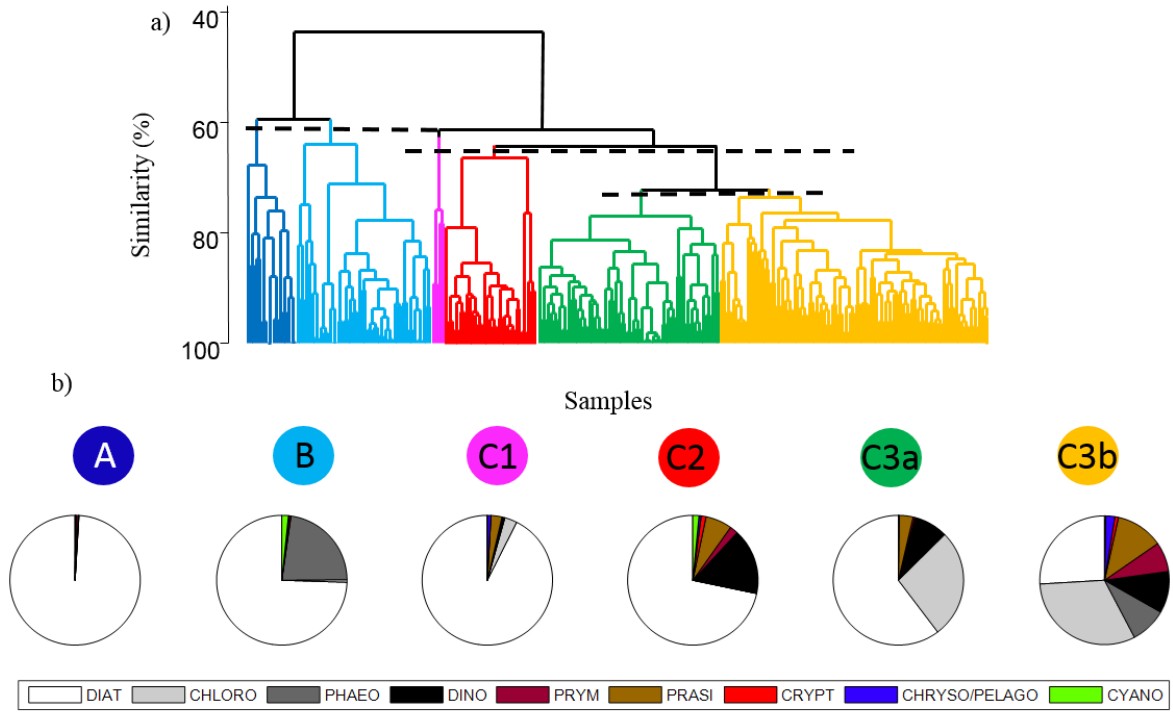

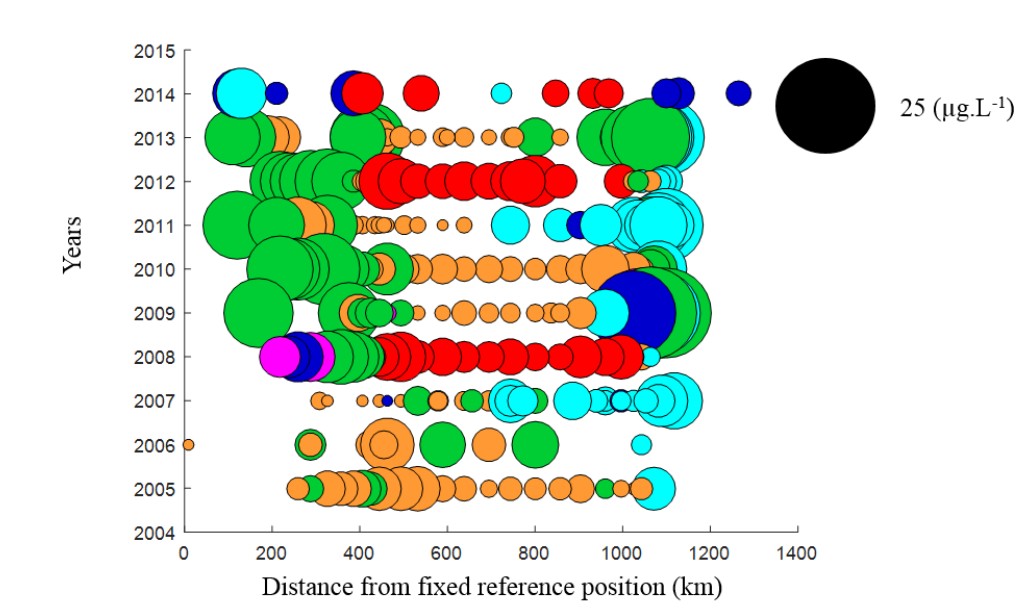






**Figure 5**

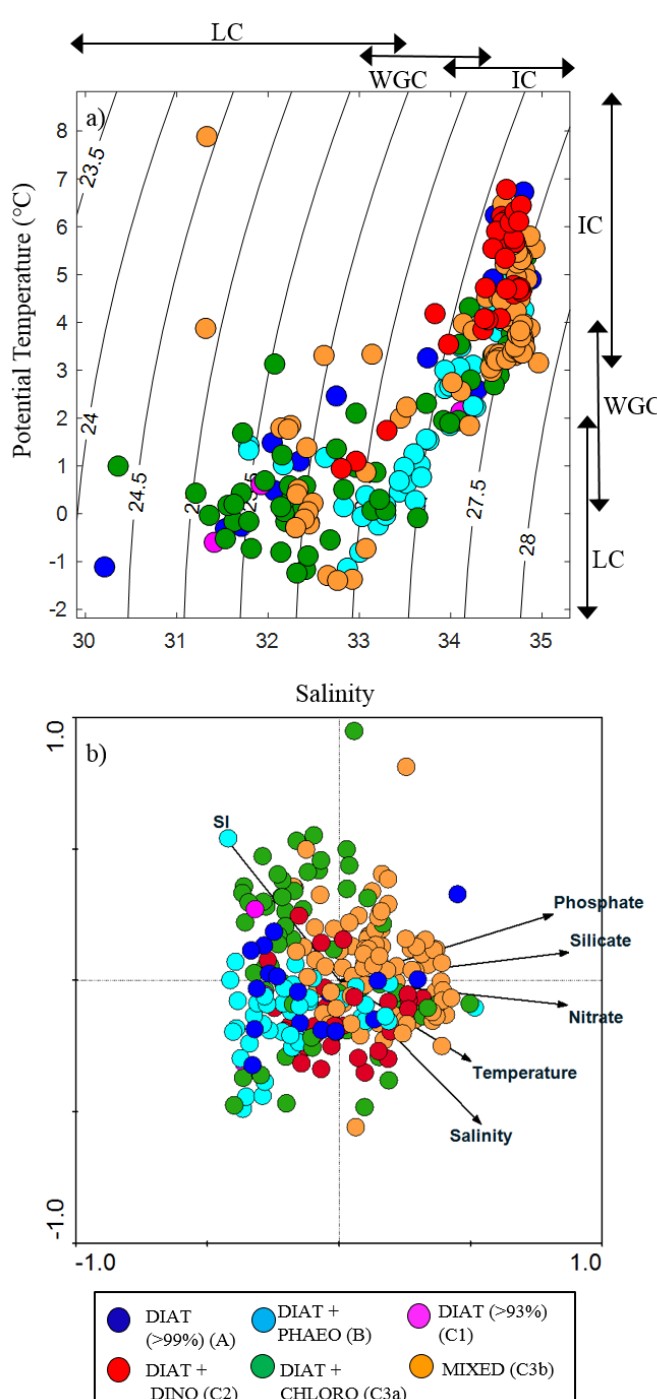



**Figure 6**

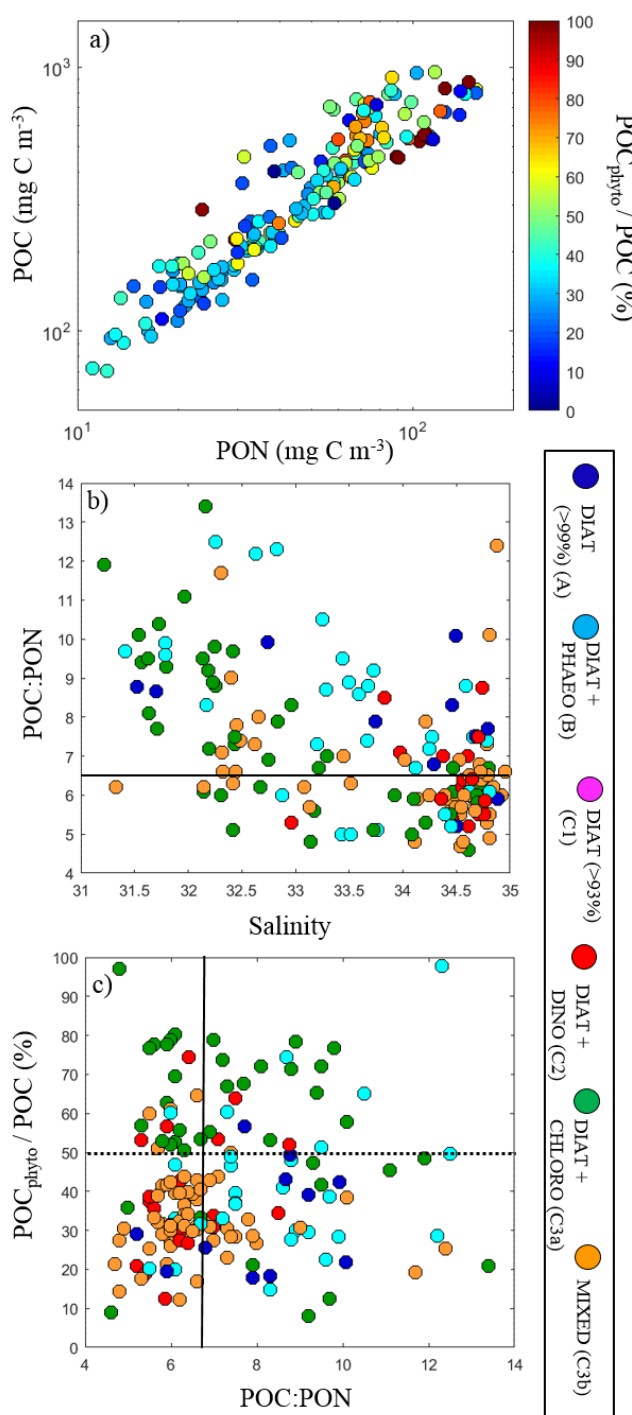


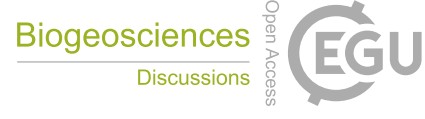

**Figure 7**

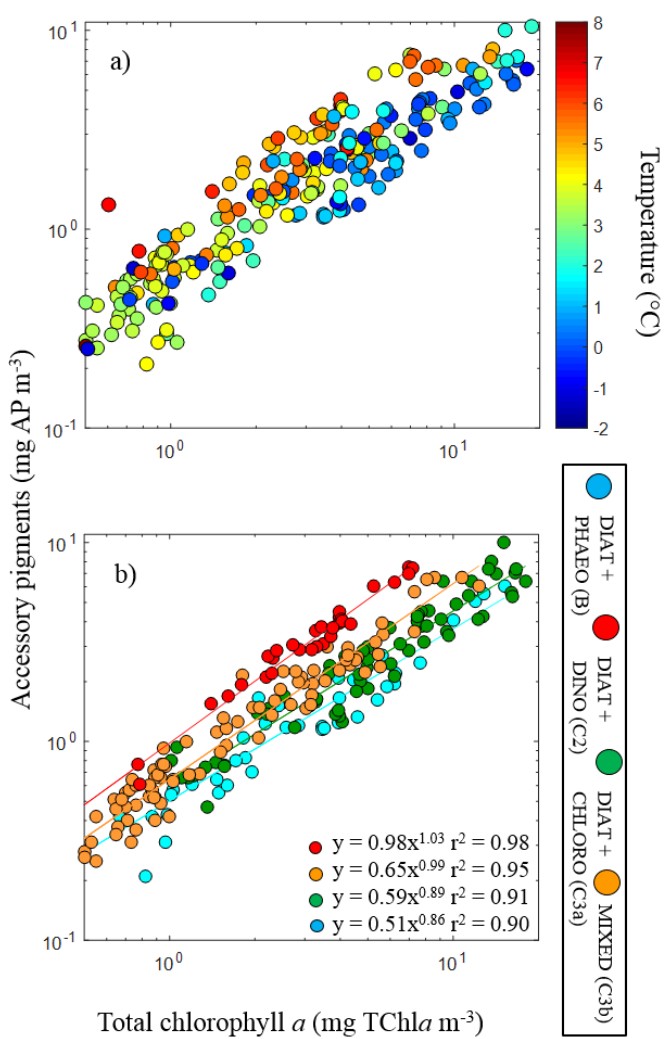