# Peer review of "Spring phytoplankton communities of the Labrador Sea (2005-2014): pigment signatures, photophysiology and elemental ratios"

_Biogeosciences, 2016_

## Referee Comment (RC1) · Anonymous Referee #1 · 22 Aug 2016

General comments This work provides information on the phytoplankton groups found in the surface waters of the Labrador Sea. Pigment signatures determined with HPLC were analyzed with CHEMTAX to obtain the contribution of the various algal groups to the total chlorophyll a concentration. The authors also related the phytoplankton biogeographic distribution to the properties of the various water masses and the photophysiology of cells during the late spring /early summer over a 10 year period. The use of CHEMTAX for this data set is a novel application, however, a previous publication by Fragoso et al. in 2016 described the phytoplankton communities linked to the various hydrographical areas of the Labrador Sea at depths less than 50 meters using microscopy. Although both microscopy and CHEMTAX analytical methods are critical

to any biogeographic examination of phytoplankton, I feel the two methods should have been combined into a single manuscript as they complement one another. Therefore, although I consider this work to be of value in its contribution to our understanding of the dynamics of the biogeochemical characteristics of the Labrador Sea, I feel its content fails to merit publication in present form.

Key problems that I feel need to be addressed include: 1) the absence of the initial CHEMTAX matrices and RMS errors 2) the organization of the methodology section; it is not well structured, it includes CHEMTAX results and lacks information (see specific comments) 3) the amount of information presented regarding taxonomy; species-specific information for the encountered groups of diatoms would have helped to understand differences on the photoprotective responses observed 4) of all the identified pigments (presented in Table 2) only the (DD+DT): chl a and DT:(DT+DD) are included for discussion on cell physiology. The authors should at least have included why they did not use the PPC:PSC, PPC:chl a or the pigment chlorophyllide a 6) the use of accepted and standardized abbreviations for the marker pigments and the phytoplankton groups in Tables 2 and 3 and throughout the text and finally 7) the correction of any incorrectly assigned references.

Specific comments

Introduction Line 54: change for Phaeocystis spp. colonies (> 100 $\mu$m). Line 83: update references. Line 84 what do you mean by "while the influence of phytoplankton composition on photophysiological patterns has not been investigated thoroughly?" please explain further.

Methods In general this section is not well structured and needs clarification and more detail.

Sampling and analysis are combined throughout this section and need to be presented with more organization. I recommend organizing this section into separate Study Area, Sampling and Biogeochemical Analyses sub-sections and limiting relevant data to relevant sub-sections.

Line 138: please include the number of stations sampled before fixed stations (was it 28 as in the previous work?). The number of depths sampled at each station should appear in the text as well.

Line 141: please write the specifications of the Seabird CTD system.

Lines 148-149: the description of how the total chl a was analyzed is presented before explaining how the collected samples for pigment analyses were filtered (probably on board?). Were samples for chl a fluorometric determination kept frozen at -20°C until analyses or at -80°C is a bit confusing. Was the extraction (90% acetone) performed by keeping the filters at -20°C for 24 h? or rather the filters were kept at -20°C until analysis (extraction for 24 h with 90% acetone)? Was acidification of the samples performed?

Line 151: I recommend changing this line to "samples for detailed pigment analysis were filtered onto 25 mm Whatman GF/F filters".

Lines 151-153 How much time passed between storage and analysis for the samples? Were the samples always analyzed in the same laboratory for every cruise over the 10-year period? Information on the maximum time of filtration is not provided and is very important for xanthophyll measurements. If too much passed while doing the filtration, the measurements of diatoxanthin are likely to be meaningless. This is also important for degradation pigment information, however the later data are not presented.

Line 153: were the nutrient samples kept frozen or refrigerated until analysis?

Pigment analysis Line 166: Was calibration done with external pigment standards obtained from DHI? Was the precision of the instrument tested? Is there a variation coefficient? Do you have limits of detection? Please at least provide the limits of detection and quantification and how were they estimated and if the pigments with concentrations below this limit were reported or not. All this information is relevant and missing.

Table 2: In this table and throughout the manuscript the authors should follow the abbreviations for phytoplankton pigments and pigment formulae suggested in the Scientific Council for Oceanic Research (SCOR), Jeffrey et al. 1997 or in Higgins et al. 2011 In: Roy S, Llewellyn CA, Egeland ES, Johnsen G (eds) Phyto- plankton pigments: characterization, chemotaxonomy and applications in oceanography. Cambridge University Press, Cambridge, p 257−301. Please, don't use capital letters or other abbreviations that are not standardized!

This table should summarize the distribution of major taxonomically significant pigments found in the various algal groups during the study. This is poorly done in its current form. The authors should avoid ambiguity. For example when referring to 19'-hexanoyloxyfucocanthin (Hex-fuco), it should be mentioned that is a major pigment in haptophytes and dinoflagellates (Type-2, lacking peridinin), instead of "some dinoflagellates" or "various". This information –if provided here-would improve significantly the reading of the few next sections dealing with the marker pigments used for the CHEMTAX analysis. Only if the authors are more specific, the use of the references Jeffrey et al. 1997 or Higgins et al. 2011 make sense. Please delete the reference column of this table unless is useful (not the case in its present form).

Chlorophyll c1 + c2 should stay as Chlorophyll c1 + c2. Please avoid the use of CHLC12.

Zeaxanthin is a minor pigment present in various groups as cyanobacteria, however this group is supposed to be practically absent in polar waters. Although Blais et al. 2012 showed that cyanobacteria may be underestimated in polar regions (Beaufort Sea & Baffin Bay). Did the authors find presence of cyanobacteria using epifluorescence microscopy? Also did the authors perform any correlation analyses between prasinoxanthin and zeaxanthin to prove that the zeaxanthin encountered did or did not correspond to a group of prasinophytes-containing zeaxanthin? Please provide this information.

Pigment interpretation There are major problems with this section. The title itself is more like the title of a results section. Actually the authors use the title "CHEMTAX interpretation" as a section included in the results. I suggest the authors change the title of the pigment interpretation section to HPLC pigment data or Clustering of HPLC data for CHEMTAX or CHEMTAX analysis or something similar-

This section is not well structured and difficult to follow partially because the authors explain the use of the selected initial pigment ratios while presenting the output matrices after the CHEMTAX analyses (Table 3). This is confusing for the reader. The initial ratio matrices used to seed CHEMTAX are not presented or explained with detail. Instead ambiguous information is presented e.g. "diatoms were identified as containing high fucoxanthin to chl a ratios"

Line 171: change it for Mackey et al., 1996, version 1.95. The following paragraph is not straightforward. The information on how CHEMTAX works in general and how version 1.95 works lacks clarity. This later version is a significant improvement on CHEMTAX application since the software sets up the multiple (60) initial pigment ratio matrices to obtain the more stable final values (as was recommended for example by Latasa 2007) and was actually used and described by Wright et al. 2009 and other authors before Coupel et al. 2015! Please add the references.

Line 179 to the end of the paragraph: please use the standardized abbreviations and you should at least explain why you decided to choose these particular marker pigments for the CHEMTAX analysis. Your microscopy results from the previous work should help here in a more detailed way.

Line 183: Again, please refer to Mackey et al. 1996 before more recent studies.

Line 191 to 197: Is figure 2 referring to the mean relative concentration of the main marker pigments to total accessory pigments (wt:wt) encountered or to chl a or total chl a or is based on the pigments absolute values? Unclear. It would have been helpful to include in this figure the biogeographical region linked with each cluster (as in figure

3).

Line 198: you already explained this earlier (lines 173-74). I think this is not very well explained and this may be the reason why you mentioned it again here. Line 199-200: "To satisfy this requirement, initial pigment ratios were carefully selected and applied to each cluster". This should actually be mentioned earlier in this section when you explain and justify why you use the selected pigment markers that best describe the phytoplankton community of your study area.

Line 204: The authors should justify why they have used the "high light" field ratios from Higgins et al. 2011. Moreover, considering the importance on the photo-physiological results obtained in this study why is there not more information beside the irradiance of the experimental incubations? Was the PAR incident irradiance measured at the sampling sites?

Line 205: "Prasinophytes were separated into type 1 (containing prasinoxanthin) and type 2 (lacking prasinoxanthin)". Both genera were observed in light microscope counts (Fragoso et al. 2016)" What do you mean? Fragoso et al. 2016 enumerated pico-phytoplankton (M. pusilla < 2 um)?

Line 209: Did the authors detect by HPLC the unknown carotenoid that characterizes the unique pigment signature of M. pusilla? Did they detect the pigment micromonal in their samples? or micromonol?

Line 211: "In addition to prasinophytes –type 2 (type 2A in Higgins et al. 2011- I assume), zea is also the major accessory pigment of cyanobacteria etc.. unclear paragraph.

Line 215: "Prasinophytes (type-1, Higgins et al. 2011) indeed contain chl b so do chlorophytes and they can be distinguished by their relative ratios of lutein to chl b (Higgins et al. 2011). Was lutein detected with the HPLC analyses? Again correlations would have helped here.

Line 218: I suggest the authors change Dino-2 class for Dino2 (dinoflagellates type-2). Avoid the use of class, use what is suggested by Higgins et al. 2011. As mentioned before, this could have been nicely done in Table 2.

Line 220: Why did the authors use the term Cryptophycea instead of cryptophytes?

Line 256: Please refer to algal groups or phytoplankton groups based on pigment composition instead of "class".

Results Line 294-296: Where is cluster C1 mentioned in this section to explain Figure 4?

Line 380: Why do you present saturation irradiances here as Wm-2 when in the methodology (line 237) you mentioned the 30 different irradiance levels is expressed as $\mu$mol quanta m-2s-1. Please use same units everywhere.

Line 382: What was the % contribution of DD, DT and $\beta,\beta$-carotene to the total PPC for clusters C3b and C2?

Line 381: DD+DT/Chl a; clusters C3b and C2 have also the lowest chl a concentration. However the level of deepoxidation is higher for these two cluster. How do your DDDT/chla and PPC/PSC ratios compare with other studies for the Arctic during spring/summer transition? Actually you don't present PPC/PSC, why?

Legend of figure 3: would be better if each variable and parameter is related to the corresponding panel.

Discussion Very little information is discussed about spatial and temporal incident PAR irradiance variation.

Line 405: Chlorophytes have also been associated with land-fast ice in the Arctic (e.g. Palmer et al. 2011).

Lines 524-529: I think this is a very interesting result and a interesting point for discussion. Here is where species identification for the diatom groups of Arctic and Atlantic

waters would have been helpful. How do these results compare to other Arctic studies?

Lines 540 to 550: This paragraph deserves a better explanation with at least details on the microscopic most abundant genera for diatoms.

Lines 564 to 575: is more a repeated line of the introduction.

Lines 564 to the end: The resulting ratios of the final CHEMTAX analysis should have been discussed here, at least accordance/discrepancies with past studies in the polar environment. The interesting comparison among the carbon biomass- estimated from CHEMTAX and the estimated by microscopic observations- should have been better structured and compared with other studies.

Lines 987 to 993: please relate each variable to the corresponding panel.

References need further formatting review.

Latasa M (2007) Improving estimations of phytoplankton class abundances using CHEMTAX. Mar Ecol Prog Ser 329:13−21

Wright SW, Ishikawa A, Marchant HJ, Davidson AT, van den Enden RL, Nash G (2009) Composition and significance of picophytoplankton in Antarctic waters. Polar Biol 32:797−808

---

## Referee Comment (RC2) · S. W. Wright (Referee) · 24 Aug 2016

GENERAL COMMENTS:

This paper provides a decadal assessment of phytoplankton communities of the Labrador Sea using pigment markers and CHEMTAX analysis, as well as environmental parameters (T, S, nutrients, MLD, etc) and photosynthetic parameters. A single transect was sampled during each late spring – early summer for 10 years with high geographic resolution. The comprehensive suite of measurements makes this a valuable data set that should provide a useful reference for future cruises. I believe it is appropriate for Biogeosciences.

The analyses appear to have been competently performed and I have no worries about the data.

Although the text itself is generally well written, at the broader level the manuscript itself unfortunately has two serious problems. First, it is not well structured – in particular, it lacks a clear Aim. Secondly, and perhaps as a consequence, the authors have attempted to cover too much data in a single publication. They describe the entire data set rather than derive a clear story from it. As a result, key parts of the story are insufficiently described despite a huge volume of complex text, and the overall story is confusing. Three subplots are introduced (Accessory pigment:Chl_a ratios, POC:PON ratios, and photosynthetic parameters) that add little to (what I consider to be) the main story but add considerable verbiage and unnecessary confusion. There is possibly sufficient data here for a thesis, in which each of these subplots would warrant a separate chapter. Here they would be better relegated to separate publications, possibly followed by a review paper that integrates this study with previous work in the region.

Due to lack of a coherent focus, the data and discussion are not well integrated.

Despite these problems, this is a very useful study that should be published, but the manuscript requires substantial revision.

STRUCTURAL COMMENTS:

Introduction:

This paper desperately needs a clear Aim to provide a basis for a narrative, to dictate what is included in (or excluded from) the paper, to provide a focus for the Results, Discussion and Conclusions, and by which to judge the success of the project. There is an implicit aim in the sampling regime – "What are the major determinants of phytoplankton composition and abundance in the Labrador Sea?" My comments hereafter will address this aim, and I leave the authors to judge how appropriate they are to the

revised paper.

The Introduction must provide sufficient information to provide the context for the Aim and to allow the reader to understand the significance of the results as they are presented. It must introduce all of the major topics covered in the paper , but nothing else. Thus, the first two paragraphs (lines 42-65) are unnecessary; as is the paragraph on CHEMTAX starting line 86 (which should be replaced by a brief outline on the approach taken to address the Aim).

The description of the study region is currently split between the Introduction (lines 66-84), Methods (lines 114-132), and Discussion (lines 409 – 413). Given that the notional paper is now about the Labrador Sea, I suggest that all of this information should be amalgamated in the Intro, as should most of the description of the NAO (lines 425-430), and Figure 1.

I would specifically identify the main factors that may control phytoplankton – temp, salinity, mixed layer depth, light, nutrients, ice, meltwater. I also think that the Introduction should mention that the cruises occurred at different times of the Spring/Summer, introducing the notion of a temporal sequence, as this was the basis for one of the Conclusions (which surprised me on the first read!). Also that there were some cruises that deviated from the normal transect.

I note that there was another publication by the same authors in the same region this year. I am surprised that there was not a specific reference to how this study relates to the previous one.

Methods:

The inclusion of results in section 2.4 surprised me at first, but I think that this section is peripheral to the main story and is appropriate here.

Results:

I was frustrated by the fact that CHEMTAX results were presented only at the community level as defined through cluster analysis – but what was happening with the individual taxa that comprised these communities? Later I discovered that these results were (sort of) presented in the Discussion. I suggest that the distributions of individual taxa should be presented (with figures) before the distributions of communities.

I would like to see a more detailed analysis of the factors controlling phytoplankton in each water mass. Even though there was considerable data on photosynthetic properties, I didn't get a clear message on the role of light in controlling biomass.

The Results should include a specific section on the temporal sequence, possibly exploring the sequence of events in each region. I note in Fig 3 that the data for 2012 and 2014, which were sampled late in the season, differ from other years, particularly Chl and nutrients in the central region.

Discussion:

Much of the discussion about individual taxa in section 4.1 should be first described in the Results section. Most of sections 4.2 and 4.3 should be saved for another paper.

The Discussion should focus specifically on the results of this paper in relation to the Aim, only referring to other studies to provide context, generally in the style of "Our results match those of Smith and Jones...". Only then should the wider implications of the work be discussed, and there should be clear signals when the narrative extends beyond the current work. Much of this Discussion reads like a review. It was often difficult to determine whether the results being discussed were from this paper or from others.

Conclusions:

Most of the final paragraph seems more appropriate to the Introduction. The authors may also consider any further research questions that arise from this study.

Abstract:

I think the first sentence is redundant and that the second sentence should be extended to include the Aim. The abstract will require revision in line with the changes to the rest of the manuscript.

SPECIFIC COMMENTS:

Line 186 and Table 3: Lutein not used for chlorophytes? (Does the BIO method separate ZEA & LUT?) If not, Table 3 ZEA must be ZEA+LUT

Lines 192-200 and Figure 2: I note that two of the categories include Hex but no Chlc3 – I assume this is a simplification of the text and diagram as this combination does not exist to my knowledge. Figure 2 is unnecessary and should be replaced with a table including all pigments.

Section 3.2: Did the authors try further subdivision of group C3b? This group is by far the biggest, it is widest spread across the S-T diagram (Fig 5a), and its composition is "mixed", yet Fig 4a shows major divisions within the group. Would these subdivisions distinguish communities that were more coherent in composition and habitat?

Line 316: change "Phaeocystis (cluster B)" to "A community dominated by diatoms and Phaeocystis (cluster B)" . This is an important consideration throughout the document —- e.g. lines 328, 329 – there is not a careful distinction between the cluster groups (communities) and the taxa comprising them. I would invent an acronym or abbreviation for each community to avoid this confusion.

Line 527: The possibility that "diatom species from both Arctic and Atlantic waters varied intrinsically in pigment composition" can be supported by consulting Table 3 of this paper, where we see that they do.

Line 551: "chlorophytes were present in high concentrations on the Labrador Shelf, which may explain the discrepancy between these results." Some more details are required to constitute an explanation.

Table 5: This table should be augmented by information on the region in which each

cluster is found, and the major taxonomic components. Also expressing the values like Temperature with standard errors is inappropriate. The values are not based on repeat measurements of a single parameter –e.g. Cluster 3b is listed as 3.4+/-0.2 C, but the actual range is from about -1.3 to +8, the widest of any group. I would be surprised if the standard error given is correct. Even if is, it is meaningless. This table should list the range for each cluster instead. Also: I didn't see any reference to the data for DT:(DT+DD) in text (nor was there any reference to how long the filters were held between sample collection and freezing. This should be < 5-10 min for this parameter to be valid).

Results: I did not notice any indication that the raw pigment data were to be included in Supplementary Material or an online databank. I would hope that this will be the case to increase the value of this data set.

TECHNICAL COMMENTS:

Line 67 and throughout: References should be cited in order of date – oldest to newest

Line 84: change "while" to "but"

Line 118: inset "wide" after "km" (twice)

Line 123: change "fresh" to "low salinity". Rest of same para: three water masses are described as "warm and salty" or "cold, low salinity" but other water masses lack these descriptions (parallel form required– see below). Also, is the warm arrow parallel to the Labrador current in Fig 1 considered to be part of that current?

Line 177: The correct reference for the method ascribed to "Coupel et al. (2015)" is Higgins et al (2011).

Line 316: Add "respectively" after "(IC)" ?

Line 325: Replace "respond strongly to" with "are associated with" and "spatial aspects of the data" with "environmental parameters"

Line 331: The description of Fig 5b could hardly be more obscure: "In Atlantic waters, temporal aspects of the data were also observed (upper and lower right quadrants (Fig. 5b))." There is nothing in that figure that implies a temporal sequence. It was only when the Conclusions mentioned clear temporal differences that I searched the document for "temporal" to find what I had missed and came back to this figure. After some cross-referencing I realised that the description should have read, "In Atlantic waters (upper and lower right quadrants (Fig. 5b)), the phytoplankton community was composed of mixed taxa during May (orange circles), but became dominated by diatoms and dinoflagellates during the bloom in June (red circles), showing a clear temporal succession in these waters". More generally, the authors must not rely on the reader to discern what is in a figure. The reader is not familiar with the data and may not see what the author sees, or they may see something different. Whatever story exists in the figure, it must be stated clearly in text as part of the narrative. The figure supports the narrative, it does not replace it.

Line 368: Replace "lower accessory pigments to TChla ratio" with "lower ratio of accessory pigments to Tchla"

Line 369: Replace "(Fig. 7b). Furthermore, communities from warmer waters (Irminger Current from Atlantic origin), particularly those co-dominated by diatoms and dinoflagellates had " with "(Fig. 7b) than communities from warmer waters (Irminger Current from Atlantic origin), particularly those co-dominated by diatoms and dinoflagellates which had"

Line 376: Replace "$\mu$g C $\mu$g Chla h-1W m-2" with "$\mu$g C $\mu$g Chla h-1 W-1 m2" or " (W m-2) -1 " Also line 378

Lines 375 to 386. Sentences should be rearranged to "parallel form" i.e. talk about the same things in the same order for each case cited.

Line 392: Insert "Atlantic," before "Labrador"

[Figure]

Lines 437 – 450: Reads like a review. Note also that the paragraph starts with "Phaeo-cystis and diatoms ... (Fragoso etal 2016)" but by line 441 it's "PRESUMABLY of Phaeocystis and diatoms (Fragoso etal 2016)". Also is "eastern central Labrador Sea" (line 437) equivalent to "West Greenland Current" (line 440)?

Line 598: Add reference e.g Gieskes and Kraay (1983) Mar. Biol. 75, 179-185.

Line 886: remove "et al" ; page numbers = 78 - 80

Figure 2 is unnecessary and should be replaced with a table including all pigments.

Figure 4b. The colours of the sectors would be much more easily interpreted if they made sense to a phycologist ! Surely cyanobacteria = Cyan, chlorophytes = Dk Green, Prasinophytes = Lt Green, Phaeocystis = Brown, etc. (Leave diatoms white)

Figure 4c. The single circle as a scale is ambiguous. Does the biomass relate to the diameter or the area of the circle? In any case it's difficult to judge. There should be a range of circles representing a biomass scale (if circles are to be used). Also I estimate that about 20% of the data points are hidden in this diagram as they underlie another circle. This could be solved by increasing the breadth of the figure or using vertical bars instead of circles. Could the fronts be marked for each year by dotted lines?

Figure 5: It would be good to see individual taxa plotted in such diagrams.

Table 2 is unnecessary. The individual pigments are not part of the story – simply quote the references.

Table 3: The legend doesn't make it clear that the references cited provided the starting ratios from which these data were calculated. Cyanobacteria is misspelt.

Table 4: The formatting is strange. It looks as if it should be split into A & B, horizontally.

---

## Referee Comment (RC3) · Anonymous Referee #3 · 27 Aug 2016

The manuscript "Spring phytoplankton communities of the Labrador Sea (2005-2014): pigments signatures, photophysiology and elemental ratios" present a time series of pigments and nutrients data in the Labrador Sea from 2005 to 2014. The authors use the CHEMTAX method to interpret the pigment dataset in term of phytoplankton groups and then to describe the distribution of these phytoplankton groups. Oceanographic provinces of the Labrador Sea are identified using on physical and biogeochemical parameters as well as phytoplankton diversity. Several statistical approaches based on clustering, ordination plot and regression were used to link the distribution in time and space of the phytoplankton with the environmental parameters. Finally, several physiological parameters related to the phytoplankton communities were measured (PI

curves, POC/PON, POC/POC Chla) or extract from the pigments distribution (AP/Chla, photoprotective pigments). The physiological information is used to go further in the explanation of the link between the phytoplankton community's distribution and the environmental conditions.

General comments:

The introduction is not well structured and full of too heavy and unclear sentence. But, the manuscript goes better in the result and discussion section. The results section is clear with a good choice of graph. Sometimes, it was difficult to get the point of the use of methods and the information that sort from some data. Finally, the discussion put together in a clear way all the information in the results section and brings interesting information to parameters that were of unclear utility in the result section. The authors highlight the specificity of the species and explained their success in the different regions and use well the comparison with the literature. I recommend important change in the introduction to make it more fluent, to better extract the key information and topics of each sub-paragraph. The sentences are generally way too long and confusing. Most of them could be cut in two parts. There are several mistakes on the use of superlative in the results section. The discussion is well conducted and uses interestingly the results

Specific comments by section

Introduction

L51: better to use "structure"

L51: change the order to "functional role in the community"

L 54 to 59: there is some redundancy with the lines 51-53

L59 to 64: Unclear about the conservation or not of the stoichiometry. You said the "stoichiometry is consistent phylogenetically" and latter you mentioned, "they may vary (. . .) phenotypically within species". Be more precise on when the ratios are conserved

or not.

L70 "shelves and the basin"

L75-76: I don't think the interest to study the phytoplankton is to use it as an index of waters masses since simple parameters as temperature and salinity did a good job. It appears to me more important to highlight the possible importance of the biogeography on the biological pump, carbon export or the energy transfer to upper trophic level.

L78-84: The same idea is repeated. Please reduce the size of the sentence, too much utilization of the conjunction "and".

L82: could simplify "high-latitude Arctic/Atlantic waters" by "polar waters".

L100: redundancy with the line 88-90

L93: Please precise the concept of "functional cell size"

L94-95: "assemblage dominance": wrong, it's the dominance of phytoplankton groups and not assemblages

L95: remove "however"

L99: remove the comma.

L107: "comprehensively understand" is a pleonasm.

L108-L111: you repeat the same information than the line 106-108.

Methods

There is some confusion on the water composition of the Labrador Sea. Moreover the authors depicted as well deep and shallow currents and water masses. The authors should focus on the surface and sub-surface water-masses and circulation since the pigment dataset presented here concerned only the upper 10m.

L115: "transition zone between the Arctic and..."

L115: Newfoundland is not really the southern boundary. The North Atlantic is the southern boundary.

L119: The lower limit of the Greenland Shelf (ie 2500m) sounds very deep to characterize a shelf! I think you characterize the extension of the Greenland Current here.

L122: remove "mostly"

L122: The Irminger current is not the main water masses of the Labrador Basin since this current it is confined on the east and west borders of Labrador Basin at a mid-depth (200-600m). The Labrador Sea Water composes the water of the basin and their characteristics are mainly influenced by the winder convection with the deeper water masses (see the work of Yashayaev et al.).

L123: There is no evidence than the cold fresh after originated from Arctic contribute substantially to the deep basin since the front between the basin and the shelf is very strong. Part of the VITALS program using gliders is actually studying the exchange between the basin and the Labrador Shelf (B. De young, J. Palter et al.).

L134: "Data used in this study"

L134: remove "from stations" and "repeat".

L146: Choose between "surface" or "near-surface" and stick to it all along the manuscript.

L155: Maybe add the underline word "Back in the laboratory, POC/PON samples ..."

L171: I think the good way to describe the CHEMTAX output is "relative abundance" instead of "ratios of abundance"

L173: not clear if all the pigments ratios are from the literature.

L174: Please indicate how the algal groups present in the study area are identified.

L187: remove "that"

L190: explain here the purpose of the fourth-root transforamation.

L195: "higher" than what? Be careful to compare with something when you use a superlative.

Results

L277: "less well stratified" ... "at those stations where"

L278: replace "during" by "in"

L279: "more highly stratify": pleonasm again...

L281: "higher": then superlative to be compared with something.

L288: Not clear if the "pairwise analysis" you mentioned refer to the ANISOM one-way pairwise?

L289: too long sentence, please reduce or cut in two parts. Parentheses are at the wrong place.

L298: "especially" is useless here. In general, there is an over utilisation of adverbs in the text (mostly/especially...).

L313: superlative!! No subject of comparison...

L315: superlative again. Wrong use.

L321-324: Too long sentence make it confusing. Separate in two sentences?

L340: The table 4 is difficult to understand and could earn a better presentation.

L345: there is a problem, the title is the same than 3.3 !!

L344-352: Please present the POC-PON relationships somewhere.

L354: Please quickly explain the purpose of calculating the relationships between POCphyto and POC:PON.

L359: I would say, "... contribute for a high proportion..."

L362: superlative lower (use low or compare to something).

Discussion

L392: as noted earlier in the manuscript, the surface phytoplankton didn't growth in the Irminger water since this water mass is observed only the slope and at great depth.

L396-397: Here the concept of ecological succession should be better presented. Is the variation between a deep and shallow mixed layer associated to the season or the two conditions (shallow/deep mixed layer) can be observed at the same time of the year?

L401-403: A link is missing between this information and the above sentence.

L406: "often" and "as well" mean the same here. Please remove one of the two.

L470: I would prefer to use the mean POCphyto rather than POC>... The latter formulation is not really comparable since we don't know the dispersion of the data.

L475: were also abundant

L512-519: It should be interesting to explain the meaning of the AP/TChla ratio in term of strategy for the adaptation to light regime.

L522-523: Conufsing because you introduce "two parameters" and after you cite three parameters (Nitrate, Silicate and SI).

L540-552: You show interesting difference in the photophysiological characteristics of phytoplankton, especially between the west and east communities. Near Greenland, the communities is composed of species resistant to high light while on the Labrador Shelf, the species are less resistant to photo-inhibition. Is the light conditions are so different between east and west to explain these different adaptations to light? It could be interesting to describe these difference in the light regimes between the two side of

the Labrador Sea. The latter melting of the ice cover on the Labrador Shelf could be an explanation?

L555 to 558: The sentence is confusing. It takes time for me to understand that dinoflagellates bloom in May to avoid higher light levels. Please rephrase or separate in two sentences to improve the clarity.

---

## Author Comment (AC1) · 10 Oct 2016

We thank the reviewer for his/her comments and suggestions, which we feel have greatly improved the manuscript. Below we respond to each comment in detail. RC refers to "Reviewer's Comments" and AC to "Author's comments". We have enumerated the reviewer's comments to organise better our responses.

Reviewer #1:

RC1.1- General comments: This work provides information on the phytoplankton groups found in the surface waters of the Labrador Sea. Pigment signatures determined with HPLC were analyzed with CHEMTAX to obtain the contribution of the various algal groups to the total chlorophyll a concentration. The authors also related the phytoplankton biogeographic distribution to the properties of the various water masses and the photophysiology of cells during the late spring /early summer over a 10 year period. The use of CHEMTAX for this data set is a novel application, however, a previous publication by Fragoso et al. in 2016 described the phytoplankton communities linked to the various hydrographical areas of the Labrador Sea at depths less than 50 meters using microscopy. Although both microscopy and CHEMTAX analytical methods are critical to any biogeographic examination of phytoplankton, I feel the two methods should have been combined into a single manuscript as they complement one another.

AC1.1 - The first manuscript (Fragoso et al 2016, Progress in Oceanography) focused on phytoplankton taxonomy and includes only large phytoplankton ($>4$ $\mu$m). In addition the manuscript includes data from only 4 years (2011- 2014) and not all stations sampled along the AR7W transect line. The current manuscript focuses on additional algal groups, including those that were not considered in Fragoso et al 2016, in addition to covering a much larger dataset (10 years of data), many more stations from the AR7W line and includes biogeochemical aspects of the data. For these reasons, we decided to publish the two papers separately to ensure all aspects of phytoplankton (taxonomy, algal groups, biochemical and physiological aspects) are examined in full. This is explained in line 578.

RC1.2 Therefore, although I consider this work to be of value in its contribution to our understanding of the dynamics of the biogeochemical characteristics of the Labrador Sea, I feel its content fails to merit publication in present form. Key problems that I feel need to be addressed include: 1) the absence of the initial CHEMTAX matrices and RMS errors

AC1.2 - CHEMTAX input matrices have been inserted in the manuscript (Table 3). The output matrices and information about the range of RMS errors (in the legend of the Table S1) has now been inserted in the supplemental material. We have attached a pdf in this response letter that shows the new versions of Table 3.

RC1.3 2) the organization of the methodology section; it is not well structured, it includes CHEMTAX results and lacks information (see specific comments)

AC1.3 - The method section has now been modified according to the reviewer's suggestions. For specific changes, see the specific comments (methods) below.

RC1.4 - 3) the amount of information presented regarding taxonomy; species-specific information for the encountered groups of diatoms would have helped to understand differences on the photoprotective responses observed.

AC1.4 - We agree with the reviewer and additional comments regarding distinct diatom species in influencing differences in the photoprotective response has now been added. See new text below (line 525 in revised manuscript).

Line 525- "Although both communities were co-dominated by diatoms (relative abundance > 70 % of total chlorophyll), the ratio logAP:logTChla varied considerably, suggesting that either 1) diatom species from both Arctic and Atlantic waters varied intrinsically in pigment composition, or 2) temperature had a physiological effect on the logAP:logTChla ratio. Fragoso et al (2016) has previously observed that the diatom species from Arctic and Atlantic waters of the Labrador Sea during spring varied in terms of species composition. According to the study by Fragoso et al. (2016), the diatoms Ephemera planamembranacea and Fragilariopsis atlantica were typically found in Atlantic waters, whereas polar diatoms, including Thalassiosira species (T. hyalina, T. nordenskioeldii, for example), in addition to Bacterosira bathyomphala, Fossula arctica, Nitzschia frigida and Fragilariopsis cylindrus were all found in Arctic-influenced waters. It is possible that the distinct composition of diatoms from these biogeographical regions might have influenced the pigment composition in these waters. Despite the observed trend of logAP:logTChla varying with temperature, a direct physiological temperature-induced effect in logAP:logTChla is currently unknown."

RC1.5 - 4) of all the identified pigments (presented in Table 2) only the (DD+DT): chl a and DT:(DT+DD) are included for discussion on cell physiology. The authors should

at least have included why they did not use the PPC:PSC, PPC:chl a or the pigment chlorophyllide a

AC1.5 - We appreciate the reviewer's suggestion, however ratios of PPC:PSC would vary according to phytoplankton community structure, in addition to cell photophysiology. That is due to the inherent variations of PPC and PSC within different phytoplankton groups. So PPC:PSC ratios would likely reflect community composition, which would be difficult to extract information about cell photophysiology. The DD+DT:chla ratio only pertains to taxa which possess a xanthophyll cycle with a photoprotective (DD) and photosynthetic (DT) component.

RC1.6 - 6) the use of accepted and standardized abbreviations for the marker pigments and the phytoplankton groups in Tables 2 and 3 and throughout the text and finally

AC1.6 - Accepted and standardized abbreviations for the marker pigments according to Higgins et al (2011) as suggested by the reviewer have been updated.

RC1.7 - 7) the correction of any incorrectly assigned references.

AC1.7 - References have now been corrected.

RC1.8 - Specific comments Introduction Line 54: change for Phaeocystis spp. colonies (> 100$\mu$m).

AC1.8 - This sentence has been removed. See the introduction in the new manuscript version.

RC1.9 Line 83: update references.

AC1.9 - This sentence has been removed. See the introduction in the new manuscript version.

RC1.10 - Line 84 what do you mean by "while the influence of phytoplankton composition on photophysiological patterns has not been investigated thoroughly?" please explain further.

AC 1.10 - This sentence has been changed. See the introduction in the new manuscript version.

RC1.11 - Methods In general this section is not well structured and needs clarification and more detail. Sampling and analysis are combined throughout this section and need to be presented with more organization. I recommend organizing this section into separate Study Area, Sampling and Biogeochemical Analyses sub-sections and limiting relevant data to relevant sub-sections.

AC 1.11 An additional subsection named "Biogeochemical analysis" has been added including nutrient, POC:PON and chlorophyll a methodology.

RC1.12 - Line 138: please include the number of stations sampled before fixed stations (was it 28 as in the previous work?). The number of depths sampled at each station should appear in the text as well.

AC1.12 - Information about stations has now been added to the manuscript. See the sentence rewritten below. Samples for this manuscript were collected from the surface only and this information is included in line 147.

Line 138. "Fixed stations (total of 28), as well as some additional non-standard stations, were sampled across shelves and in the deep central basin on the AR7W section or slightly north or south of this transect (Fig. 1)."

RC1.13 - Line 141: please write the specifications of the Seabird CTD system.

AC 1.13 - Specifications of the Seabird CTD system (SBE 911) has been inserted.

RC1.14 - Lines 148-149: the description of how the total chl a was analyzed is presented before explaining how the collected samples for pigment analyses were filtered (probably on board?). Were samples for chl a fluorometric determination kept frozen at -20C until analyses or at -80C is a bit confusing. Was the extraction (90% acetone) performed by keeping the filters at -20C for 24 h? or rather the filters were kept at -20C until analysis (extraction for 24h with 90% acetone)? Was acidification of the samples

performed?

AC 1.14 - We have rewritten this sentence from Line 149 for clarification (see sentence rewritten below). The analysis of chlorophyll a has now been inserted in the other subsection "Biogeochemical analysis". Chlorophyll determination by fluorescence according to the methodology of Holm-Hansen et al (1965) includes acidification of the samples. A sentence explaining this will be added in the new version of the manuscript (see below).

Line 149 – "Filters for chlorophyll a measurements were immediately put in scintillation vials containing 10 ml of 90% acetone, which were placed into a -20oC freezer and extracted for 24 h."

New added line - "Fluorescence was determined on board after 24 h of extraction using a Turner Designs fluorometer (Holm-Hansen et al., 1965). Fluorometric analysis of chlorophyll and phaeo-pigments, using the Turner fluorometer, was always within 48 h."

RC1.15 - Line 151: I recommend changing this line to "samples for detailed pigment analysis were filtered onto 25 mm Whatman GF/F filters".

AC 1.15 - We rewrote this sentence as: "…filtered onto 25 mm glass fibre filters (GF/F Whatman Inc., Clifton, New Jersey)".

RC1.16 - Lines 151-153 How much time passed between storage and analysis for the samples? Were the samples always analyzed in the same laboratory for every cruise over the 10-year period? Information on the maximum time of filtration is not provided and is very important for xanthophyll measurements. If too much passed while doing the filtration, the measurements of diatoxanthin are likely to be meaningless. This is also important for degradation pigment information, however the later data are not presented.

AC 1.16 - We have now added these information in the manuscript. The new sentence

below would replace the sentence from Line 151.

Line 151 – "Samples for detailed pigment analysis were filtered onto 25 mm glass fibre filters (GF/F Whatman Inc., Clifton, New Jersey) and immediately flash frozen in liquid nitrogen and kept frozen in a freezer (at -80° C) until analysis in the BIO (2005-2013) or NOC (2014) laboratories within 2-3 months of collection. Volumes of water sampled for HPLC analysis were adjusted such that samples took less than 10 mins to filter."

RC1.17 - Line 153: were the nutrient samples kept frozen or refrigerated until analysis?

AC 1.17 - We have added this information in the text. The new sentence below would replace the sentence from Line 153.

Line 153 – "Nutrient samples were kept refrigerated at 5°C and analysed at sea (within 12 h of collection) on a SEAL AutoAnalyser III."

RC1.18 - Pigment analysis Line 166: Was calibration done with external pigment standards obtained from DHI? Was the precision of the instrument tested? Is there a variation coefficient? Do you have limits of detection? Please at least provide the limits of detection and quantification and how were they estimated and if the pigments with concentrations below this limit were reported or not. All this information is relevant and missing.

AC 1.18 - Information about the standards, calibration and quantification procedures are described in detail in Stuart and Head (2005). Information about precision, coefficient of variation and limits of detection have now been inserted in the new version of the manuscript. The new sentence below would replace the sentence from Line 160.

Line 160 – "Pigments (chlorophyll a and accessory pigments) were quantified using reverse-phase, High-Performance Liquid Chromatography (HPLC). Methods for 2005-2013 (Hudson cruises), including information about the standards, calibration and quantification procedures are described in detail in Stuart and Head (2005), known as the "BIO method". Methods for samples collected in 2014 (JR302 cruise) are described in Poulton et al (2006). Quality control of both methods was applied according to Aiken et al (2009). Precision of the instruments was tested by running samples and standards and the coefficient of variation for pigments were < 10% of the mean. Limits of detection were ∼0.01 and 0.002 mg m-3 for carotenoids and chlorins, respectively (Head, pers. comm, Poulton et al 2006). Pigments concentrations below detection limits were not reported." RC1.19 - Table 2: In this table and throughout the manuscript the authors should follow the abbreviations for phytoplankton pigments and pigment formulae suggested in the Scientific Council for Oceanic Research (SCOR), Jeffrey et al. 1997 or in Higgins et al. 2011 In: Roy S, Llewellyn CA, Egeland ES, Johnsen G (eds) Phyto- plankton pigments: char-acterization, chemotaxonomy and applications in oceanography. Cambridge University Press, Cambridge, p 257

AC 1.19 - Pigment abbreviations were updated as suggested by the reviewer. We have added the new version of Table 2 as a pdf.

RC1.20 - This table should summarize the distribution of major taxonomically signifi-cant pigments found in the various algal groups during the study. This is poorly done in its current form. The authors should avoid ambiguity. For example when referring to 19'-hexanoyloxyfucocanthin (Hex-fuco), it should be mentioned that is a major pig-ment in haptophytes and dinoflagellates (Type-2, lacking peridinin), instead of "some dinoflagellates" or "various". This information –if provided here-would improve signif-icantly the reading of the few next sections dealing with the marker pigments used for the CHEMTAX analysis. Only if the authors are more specific, the use of the ref-erences Jeffrey et al. 1997 or Higgins et al. 2011 make sense. Please delete the reference column of this table unless is useful (not the case in its present form).

AC 1.20 - This table has been updated. We have included the more specific information requested following Jeffrey et al (1997), Higgins et al (2011) and Vidussi et al (2004). See the new version of the table updated.

RC1.21 - Chlorophyll c1 + c2 should stay as Chlorophyll c1 + c2. Please avoid the use

of CHLC12.

AC 1.21 - Abbreviations were updated in the new version of the manuscript according Higgins et al 2011 as the reviewer suggested.

RC1.22- Zeaxanthin is a minor pigment present in various groups as cyanobacteria, however this group is supposed to be practically absent in polar waters. Although Blais et al. 2012 showed that cyanobacteria may be underestimated in polar regions (Beaufort Sea & Baffin Bay). Did the authors find presence of cyanobacteria using epifluorescence microscopy?

AC 1.22 - We did not count cyanobacteria but referred to information about the presence of Synechococcus in the Labrador Sea from previous reference (Li et al 2016) as stated in line 211.

RC1.23 - Also did the authors perform any correlation analyses between prasinoxanthin and zeaxanthin to prove that the zeaxanthin encountered did or did not correspond to a group of prasinophytes-containing zeaxanthin? Please provide this information.

AC 1.23 - We are not sure if we understand this point raised by the reviewer as a correlation between prasinoxanthin and zeaxanthin would not directly determine whether the zeaxanthin found belongs to prasinophyte-containing zeaxanthin or cyanobacteria. Zeaxanthin, in this study, represented not only prasinophytes type 2, but also chlorophytes and cyanobacteria. Moreover, species representing prasinophyte type 2, such as Pyramimonas and M. pusilla have been observed (qualitatively in our samples, although not directly counted due to difficulties in quantification) in the Labrador Sea from microscope observation of Lugols fixed samples. M. pusilla is abundant in the North Water Polynya in regions near the Labrador Sea (e. g. Buffin Bay) as stated in line 208.

RC1.24 - Pigment interpretation There are major problems with this section. The title itself is more like the title of a results section. Actually the authors use the title "CHEM-

TAX interpretation" as a section included in the results. I suggest the authors change the title of the pigment interpretation section to HPLC pigment data or Clustering of HPLC data for CHEMTAX or CHEMTAX analysis or something similar-

AC 1.24 - The title of the section has been changed to "CHEMTAX analysis" as suggested.

RC1.25 - This section is not well structured and difficult to follow partially because the authors explain the use of the selected initial pigment ratios while presenting the output matrices after the CHEMTAX analyses (Table 3). This is confusing for the reader. The initial ratio matrices used to seed CHEMTAX are not presented or explained with detail. Instead ambiguous information is presented e.g. "diatoms were identified as containing high fucoxanthin to chl a ratios"

AC 1.25 - Initial pigment ratios have now been inserted in the new version of the manuscript (see the pdf with the new version of this table) and output ratio information has been moved to the supplemental material. Explanation for selected ratios are explained in line 197 onwards and we have included a column in the initial pigment ratios table mentioning the source reference where the ratios were taken from to seed initial CHEMTAX analysis.

RC1.26 - Line 171: change it for Mackey et al., 1996, version 1.95.

AC 1.26 - Changes updated.

RC1.27 - The following paragraph is not straightforward. The information on how CHEMTAX works in general and how version 1.95 works lacks clarity. This later version is a significant improvement on CHEMTAX application since the software sets up the multiple (60) initial pigment ratio matrices to obtain the more stable final values (as was recommended for example by Latasa 2007) and was actually used and described by Wright et al. 2009 and other authors before Coupel et al. 2015! Please add the references.

AC 1.27 - This paragraph has been rewritten (see paragraph below) and the earlier references have been included.

Line 171 – "The CHEMTAX software (Mackey et al., 1996) was used to estimate ratios of abundance of distinct micro-algal classes to total chlorophyll a from in situ pigment measurements. The software utilises a factorization program that uses "best guess" ratios of accessory pigments to chlorophyll a that are derived for different classes from the literature available and marker pigment concentrations of algal groups that are known to be present in the study area. The program uses the steepest descent algorithm to obtain the best fit to the data based on assumed pigment to chlorophyll a ratios (for more detail, see Mackey et al 1996). Because CHEMTAX is sensitive to the seed values of the initial ratio matrix (Latasa et al 2007), we used a later version (1.95) to obtain the more stable output matrices. In this CHEMTAX version, the initial matrices are optimized by generating 60 further pigment ratio tables using a random function (RAND in Microsoft Excel) as described in Wright et al., (2009). The results of the six best output matrices (with the smallest residuals, equivalent to 10 % of all matrices) were used to calculate the averages of the abundance estimates and final pigment ratios."

RC1.28 - Line 179 to the end of the paragraph: please use the standardized abbreviations and you should at least explain why you decided to choose these particular marker pigments for the CHEMTAX analysis. Your microscopy results from the previous work should help here in a more detailed way.

AC 1.28 - The explanation of why we used these pigments is mentioned later in the text (line 197). Thus, in the new version of the manuscript, we would include a new paragraph (before the sentence from Line 178), where we would insert sentences from Lines 197 – 204. To finalise this paragraph, we would add the sentence from line 178-181. We believe that this organisation in the text would better guide the reader.

RC1.29 - Line 183: Again, please refer to Mackey et al. 1996 before more recent

studies.

AC 1.29 - Reference now added.

RC1.30 - Line 191 to 197: Is figure 2 referring to the mean relative concentration of the main marker pigments to total accessory pigments (wt:wt) encountered or to chl a or total chl a or is based on the pigments absolute values? Unclear. It would have been helpful to include in this figure the biogeographical region linked with each cluster (as in figure 3).

AC 1.30 - Figure 2 is the first statistical treatment using PRIMER to separate out samples based on a similarity index before applying CHEMTAX. It refers to the mean relative (transformed) concentration of each of the selected pigments (e.g. Fuco) to the total selected pigments (i.e. sum of Fuco + Chl c3 + Hex-fuco + Chl b + Peri + Allo + But-fuco + Pras + Zea+Lut). Further clarification has now been added to the legend of Figure 2 in line 985. We have now added to Figure 2 a biogeographical plot showing the cluster groups as depicted in Figure 3 (Fig. 2b). This figure will be moved to the supplemental material. (Please see the figure in pdf).

RC1.31 - Line 198: you already explained this earlier (lines 173-74). I think this is not very well explained and this may be the reason why you mentioned it again here. Line 199-200: "To satisfy this requirement, initial pigment ratios were carefully selected and applied to each cluster". This should actually be mentioned earlier in this section when you explain and justify why you use the selected pigment markers that best describe the phytoplankton community of your study area.

AC 1.31 - We have reorganised these sentences and the order of explanation as mentioned in the comments above.

RC1.32- Line 204: The authors should justify why they have used the "high light" field ratios from Higgins et al. 2011. Moreover, considering the importance on the photo-physiological results obtained in this study why is there not more information beside the

irradiance of the experimental incubations? Was the PAR incident irradiance measured at the sampling sites?

AC 1.32 - High light field ratios were chosen because the samples in our analysis were collected from surface waters (< 10 m) where they are likely to be exposed to high light levels; for example, in May and June daily irradiance levels may exceed >30 mol m-2 d-1 ( see Figure 2a, Harrison et al 2013). In the new version of the manuscript, we have now added a sentence at line 204 explaining why we chose high light field ratios.

Line 204. "High light field ratios were chosen because samples were collected from surface waters during May and June (average monthly irradiance >30 mol m-2 d-1, Harrison et al 2013)."

RC1.33 - Line 205: "Prasinophytes were separated into type 1 (containing prasinox-anthin) and type 2 (lacking prasinoxanthin)". Both genera were observed in light mi-croscope counts (Fragoso et al. 2016)" What do you mean? Fragoso et al. 2016 enumerated pico- phytoplankton (M. pusilla < 2 um)?

AC 1.33 - We apologise for the confusion. Pyraminomas and M. pusilla were observed qualitately from microscope observations but not enumerated by Fragoso et al (2016). We have now changed the reference to (Fragoso, pers obs) in the text to avoid confu-sion.

RC1.34 - Line 209: Did the authors detect by HPLC the unknown carotenoid that char-acterizes the unique pigment signature of M. pusilla? Did they detect the pigment micromonal in their samples? or micromonol?

AC 1.34 - The pigment micromonal was not identified as part of the HPLC analytical protocol followed (i.e. it was not a pigment peak listed for identification).

RC1.35 - Line 211: "In addition to prasinophytes –type 2 (type 2A in Higgins et al. 2011- I assume), zea is also the major accessory pigment of cyanobacteria etc.. unclear para- graph.

AC 1.35 - The beginning of this sentence has been rewritten for clarification. See sentence below.

Line 211 – "Zea + Lut is not only found in prasinophytes –type 2, but is also the major accessory pigment of cyanobacteria. . ."

RC1.36 - Line 215: "Prasinophytes (type-1, Higgins et al. 2011) indeed contain chl b so do chlorophytes and they can be distinguished by their relative ratios of lutein to chl b (Higgins et al. 2011). Was lutein detected with the HPLC analyses? Again correlations would have helped here.

AC 1.36 - The BIO method does not separate lutein and zeaxanthin so we have now renamed it as Zea + Lut.

RC1.37 - Line 218: I suggest the authors change Dino-2 class for Dino2 (dinoflagellates type-2). Avoid the use of class, use what is suggested by Higgins et al. 2011. As mentioned before, this could have been nicely done in Table 2.

AC 1.37 - We have rewritten this sentence now using the best terminology. Table 2 has been updated (see table ).

Line 218- ". . .(herein defined as dinoflagellates type-2 (DINO-2) according to Higgins et al (2011)). . .".

RC1.38 - Line 220: Why did the authors use the term Cryptophycea instead of cryptophytes?

AC 1.38 - Cryptophytes, instead of cryptophycea, is now used in the revised text.

RC1.39 - Line 256: Please refer to algal groups or phytoplankton groups based on pigment composition instead of "class".

AC 1.39 - "Phytoplankton/algal class" has been changed to "phytoplankton/algal groups".

RC1.40 - Results Line 294-296: Where is cluster C1 mentioned in this section to explain Figure 4?

AC 1.40 - Cluster C1 has now been included in this sentence.

RC1.41 - Line 380: Why do you present saturation irradiances here as Wm-2 when in the methodology (line 237) you mentioned the 30 different irradiance levels is expressed as $\mu$mol quanta m-2s-1. Please use same units everywhere.

AC 1.4 - Irradiance units used throughout are now "Wm-2" in the text.

RC1.42 - Line 382: What was the % contribution of DD, DT and,-carotene to the total PPC for clusters C3b and C2?

AC 1.42 - The percentage contribution of DD, DT and,-carotene to the total PPC would vary according to the total amount of PPC (similar situation as comparing to DD+DT/Chl a, see comments below). Moreover, as mentioned previously, we feel that this information is irrelevant for photophysiology because of the influence of phytoplankton community structure in the overall PPC values.

RC1.43 - Line 381: DD+DT/Chl a; clusters C3b and C2 have also the lowest chl a concentration. However the level of deepoxidation is higher for these two cluster. How do your DDDT/chla and PPC/PSC ratios compare with other studies for the Arctic during spring/summer transition? Actually you don't present PPC/PSC, why?

AC 1.43 - Again, ratios of PPC:PSC would vary according to phytoplankton community structure and not just cell photophysiology. For example, a community dominated by diatoms would have high PSC:PPC, while a community dominated by prasinophytes would have low PSC:PPC. Thus, we believe that this information is insufficient to discuss photophysiology in mixed phytoplankton communities. However, as the reviewer pointed out, the level of de-epoxidation was high, which suggests that these communities were exposed to high light levels. We changed the sentence from line 560, where we cite other studies (i.e., Alou-Font et al 2016) that show similar patterns.

Line 560. "These communities also had high diatoxanthin levels compared with the other phytoplankton communities in this study, suggesting that the community was experiencing higher light intensities (Moisan et al., 1998). Increases in photoprotective pigments, including (DD+DT)/Chla, have been reported to occur in Arctic phytoplankton communities from spring to summer presumably as a response to higher irradiance (Alou-Front et al 2016). Thus, photoprotective capacity can be a key determinant for phytoplankton survival and may also be related to the taxonomic segregation observed in Arctic and Atlantic phytoplankton communities."

RC1.44 - Legend of figure 3: would be better if each variable and parameter is related to the corresponding panel.

AC 1.44 - We have now related each variable and parameter to the corresponding panel. See new sentence below.

Line 988 "Figure 3 – Map with sampling stations and distances from a fixed reference position (Northeast Gulf of St Lawrence) in the x-axis shown by the star (a). Values are given at individual stations sampled between 2005 and 2014 (y-axis) for the following variables: date of sample collection (b), temperature (c), salinity (d), stratification index (SI) (e), chlorophyll a (f), nitrate ($NO_3^-$) (g), phosphate ($PO_4^{3-}$) (h), silicate ($Si(OH)_4$) concentrations (i), ratios of particulate organic carbon (POC) to particulate organic nitrogen (PON) (j), silicate to nitrate ($Si(OH)_4$:$NO_3^-$) ratios (k), and nitrate to phosphate ($NO_3^-$:$PO_4^{3-}$) ratios (l). LSh = Labrador Shelf, LSl = Labrador Slope, CB = Central Basin, GSl = Greenland Slope, GSh = Greenland Shelf."

RC1.45 - Discussion Very little information is discussed about spatial and temporal incident PAR irradiance variation.

AC 1.45 - Unfortunately we do not have PAR measurements for each site during the 10 years of cruise observations. However, we discuss PAR indirectly through mixed layer depth, stratification index and progression of solar incidence from May to June throughout the whole discussion section. We have also extended the discussion of the

effect of irradiance on photosynthetic parameters in the new version of the manuscript (new version of Line 560 as mentioned above).

RC1.46 - Line 405: Chlorophytes have also been associated with land-fast ice in the Arctic (e.g. Palmer et al. 2011).

AC 1.46 - A reference including chlorophytes having been found in land-fast ice in the Arctic has now been added.

RC1.47 - Lines 524-529: I think this is a very interesting result and an interesting point for discussion. Here is where species identification for the diatom groups of Arctic and Atlantic waters would have been helpful. How do these results compare to other Arctic studies?

AC 1.47 - We have now added a sentence discussing the influence of distinct species in the variable AP:Tchla ratios (see comments above). There are few studies that have investigated what causes such distinct AP:TChla ratios. Changes in AP:TChla ratios vary with community structure and comparison with another Arctic study is discussed in line 514 (see new version of this sentence below). However, it is still unclear why this pattern is observed.

Line 514: Changes in the ratios of logAP:logTChla as a function of phytoplankton community composition has been previously observed by Stramska et al. (2006). These authors showed a higher slope of logAP:logTChla when dinoflagellates were dominant during summer in northern polar Atlantic waters as opposed to lower ratios associated with flagellates in spring.

RC1.48 - Lines 540 to 550: This paragraph deserves a better explanation with at least details on the microscopic most abundant genera for diatoms.

AC 1.48 - We agree with the reviewer and a discussion about distinct species of diatoms in influencing the AP:TChla has now been included in the text. See line our response to comment the reviewer comment above (RC 1.4).

RC1.49 - Lines 564 to 575: is more a repeated line of the introduction.

AC 1.49 - This whole paragraph (lines 564 to 575) has been removed.

RC1.50 - Lines 564 to the end: The resulting ratios of the final CHEMTAX analysis should have been discussed here, at least accordance/discrepancies with past studies in the polar environment. The interesting comparison among the carbon biomass- estimated from CHEMTAX and the estimated by microscopic observations- should have been better structured and compared with other studies.

AC 1.50 - We have now included more references that compare the two methods of biomass estimations (CHEMTAX and microscopy) from polar environments. See the new version of the paragraph below.

Line 588 - Phaeocystis (r2 = 0.79) and diatom (r2 = 0.74) biomasses were well correlated when carbon biomasses estimated from microscopic counts when compared with CHEMTAX-derived algal chlorophyll a biomass (data not shown). Diatoms are the group that usually show the best agreement between the two methods of biomass estimations (Vidussi et al. 2004, Coupel et al 2015, Mendes et al 2012). For Phaeocystis, a positive relationship between the two methods of biomass estimation (CHEMTAX and microscopy) confirms that using chlorophyll c3 was appropriate for detecting and quantifying Phaeocystis biomass in the Labrador Sea. Similar associations have been observed for Phaeocystis from boreal waters (e.g. P. pouchetii and P. globosa ; Antajan et al., 2004; Muylaert et al., 2006; Stuart et al., 2000; Wassmann et al., 1990), while other pigment markers have been used elsewhere, e.g. 19- hexanoyloxyfucoxanthin, which is characteristic of Phaeocystis antarctica in austral polar waters (Arrigo et al., 2010, 2014; Fragoso and Smith, 2012; Fragoso, 2009). Dinoflagellates gave a poor correlation between biomass estimates made using the two methods (r2 = 0.12, data not shown). A lack of or weak relationship between both biomass estimations for dinoflagellates has been previously reported in Artic waters (Vidussi et al 2004 Coupel et al 2005). The argument for this inconsistency is that some heterotrophic dinoflagellates, which usually lack photosynthetic pigments unless they ingest a prey that contains them, might have been included in the microscopic counts, and it is possible that the same occurred in Fragoso et al. (2016). Cryptophyte biomass estimates from both methods were not related (data not shown), likely as the biomass of this group was underestimated in microscopic counts. Inconsistences between CHEMTAX and microscopy methods of estimating biomasses have also been observed in nanoflagellates and this is assumed to be because of the low accuracy of visual microscopic counts (Coupel et al 2015).

RC1.51 - Lines 987 to 993: please relate each variable to the corresponding panel.

AC 1.51 - We have now related each variable and parameter with the corresponding panel.

RC1.52 - References need further formatting review. Latasa M (2007) Improving estimations of phytoplankton class abundances using CHEMTAX. Mar Ecol Prog Ser 329:13

Wright SW, Ishikawa A, Marchant HJ, Davidson AT, van den Enden RL, Nash G (2009) Composition and significance of picophytoplankton in Antarctic waters. Polar Biol 32:797

AC 1.52 - References added.

Please also note the supplement to this comment:
http://www.biogeosciences-discuss.net/bg-2016-295/bg-2016-295-AC1-supplement.pdf

―――――――――――――――――

---

## Author Comment (AC2) · 10 Oct 2016

We thank the reviewer for his comments and suggestions, which we feel have greatly improved the manuscript. Below we respond to each comment in detail. RC refers to "Reviewer's Comments" and AC to "Author's comments". We have enumerated the reviewer's comments to organise better our responses.

Reviewer #2, Simon Wright:

RC2.1 - GENERAL COMMENTS:

This paper provides a decadal assessment of phytoplankton communities of the Labrador Sea using pigment markers and CHEMTAX analysis, as well as environ-

mental parameters (T, S, nutrients, MLD, etc) and photosynthetic parameters. A single transect was sampled during each late spring – early summer for 10 years with high geographic resolution. The comprehensive suite of measurements makes this a valuable data set that should provide a useful reference for future cruises. I believe it is appropriate for Biogeosciences.

The analyses appear to have been competently performed and I have no worries about the data. Although the text itself is generally well written, at the broader level the manuscript itself unfortunately has two serious problems. First, it is not well structured – in particular, it lacks a clear Aim.

AC2.1 - The aim of the work has now been reinforced in the last paragraph of the Introduction. Please see the new version of the introduction attached as a pdf.

RC2.2 -Secondly, and perhaps as a consequence, the authors have attempted to cover too much data in a single publication. They describe the entire data set rather than derive a clear story from it. As a result, key parts of the story are insufficiently described despite a huge volume of complex text, and the overall story is confusing. Three subplots are introduced (Accessory pigment:Chl_a ratios, POC:PON ratios, and photosynthetic parameters) that add little to (what I consider to be) the main story but add considerable verbiage and unnecessary confusion. There is possibly sufficient data here for a thesis, in which each of these subplots would warrant a separate chapter. Here they would be better relegated to separate publications, possibly followed by a review paper that integrates this study with previous work in the region. Due to lack of a coherent focus, the data and discussion are not well integrated.

AC2.2 - We have now reaffirmed the aim of the paper which is to compare the biogeochemical and photophysiological properties of phytoplankton communities from contrasting biogeographical regions in the Labrador Sea and to create a baseline of these trends which could be compared with in the future. Although we agree with the reviewer that this manuscript covers a lot of information, we feel that the results from
sections 3.4 and 3.5 are directly linked to the information on phytoplankton groups and that writing a different paper about POC:PON ratios and photochemical aspects on its own becomes largely irrelevant. We hope the reviewer can appreciate our focus and restated aim and view our responses in light of these.

RC2.3 -Despite these problems, this is a very useful study that should be published, but the manuscript requires substantial revision.

STRUCTURAL COMMENTS: Introduction: This paper desperately needs a clear Aim to provide a basis for a narrative, to dictate what is included in (or excluded from) the paper, to provide a focus for the Results, Discussion and Conclusions, and by which to judge the success of the project.

AC2.3 - A clear aim has now been added in the last paragraph of the introduction and the Results, Discussion and Conclusions all follow this aim.

RC2.4 -There is an implicit aim in the sampling regime – "What are the major determinants of phytoplankton composition and abundance in the Labrador Sea?" My comments hereafter will address this aim, and I leave the authors to judge how appropriate they are to the revised paper.

AC2.4 - The major determinants of phytoplankton composition and abundance has already been investigated in Fragoso et al 2016. The uniqueness of this manuscript is that it takes the subject matter a step further to focus on additional algal groups, including those that potentially were not included in Fragoso et al 2016. Moreover, we investigate the biogeochemical (C:N) and photophysiological signatures of these communities that are shown to vary across the biogeochemical regions of the Labrador Sea. Therefore, the current manuscript details a much larger dataset (10 years) and more stations from the AR7W line in order to provide a baseline of the determination of phytoplankton communities and their specific biogeochemical and photophysiological signatures from spring to summer in the Labrador Sea. Following this reasoning, we have decided not to follow the direct suggestions of the reviewer to focus only on

hydrography, but rather we examine other aspects (biogeochemistry and physiology) of the phytoplankton communities. This provides a more holistic understanding of phytoplankton dynamics in the Labrador Sea.

RC2.5 -The Introduction must provide sufficient information to provide the context for the Aim and to allow the reader to understand the significance of the results as they are presented. It must introduce all of the major topics covered in the paper, but nothing else. Thus, the first two paragraphs (lines 42-65) are unnecessary; as is the paragraph on CHEMTAX starting line 86 (which should be replaced by a brief outline on the approach taken to address the Aim).

AC2.5 - A holistic understanding of phytoplankton dynamics requires suitable introduction of phytoplankton photophysiology and biogeochemistry. Therefore, we retain the first two paragraphs as paragraph one refers to the impact of hydrography on community structure and paragraph two refers to the impact of phytoplankton community structure on C:N ratios. However, we do agree with the reviewer that the introduction needs to better guide the reader and provide enough information for the stated aim of the paper. Thus, we have now radically shortened the whole introduction and reorganised it in attempt to make it clearer for the reader.

RC2.6 -The description of the study region is currently split between the Introduction (lines 66- 84), Methods (lines 114-132), and Discussion (lines 409 – 413). Given that the notional paper is now about the Labrador Sea, I suggest that all of this information should be amalgamated in the Intro, as should most of the description of the NAO (lines 425-430), and Figure 1.

AC2.6 - Paragraph three in the introduction gives a brief overview of why it is important to research the Labrador Sea. We consider this information to be crucial for the wider relevance of our study where phytoplankton biogeography is influenced by contrasting hydrography. Phytoplankton biogeography in the Labrador Sea, in turn, influences the contrasting biogeochemical and photophysiological traits observed in distinct water masses. The study area section in the methods focuses mostly on the complex hydrography of the area and provides further details that are crucial for the reader to understand the biogeography of the community. Line 409-413 has now been moved to the study area section (see new version of introduction) as suggested by the reviewer as it is important information about the region. Although the reviewer suggests it would be better to amalgamate the information from the introduction and the study area, we believe that the information is more clearly organised as is. Thus, the introduction provides a brief explanation of why is important to study the Labrador Sea, while the study area section in the methods gives a more in depth description of the complex hydrography of the region. The possible effects of the NAO are not investigated in depth in this manuscript, therefore, it is not relevant to describe it in the Introduction.

RC2.7 -I would specifically identify the main factors that may control phytoplankton – temp, salinity, mixed layer depth, light, nutrients, ice, meltwater. I also think that the Introduction should mention that the cruises occurred at different times of the Spring/Summer, introducing the notion of a temporal sequence, as this was the basis for one of the Conclusions (which surprised me on the first read!). Also that there were some cruises that deviated from the normal transect. I note that there was another publication by the same authors in the same region this year. I am surprised that there was not a specific reference to how this study relates to the previous one.

AC2.7 - We agree with the reviewer that the notion of the temporal sequence of the phytoplankton communities is important to guide the reader. Therefore, we have now added information about the seasonality of late spring/early phytoplankton communities in the introduction (see paragraph three of the new version). However, we left the detailed information about cruises occurring at different times in the Methods section, and have made it clear to the reader that there is temporal variability in the sampling times. See the sentences rewritten below.

Line 137 – "Stations were sampled during late spring and/or early summer, varying mostly within a 6 week window (see sampling dates in Table 1) over a 10 year period

(2005-2014) by scientists from the Canadian Department of Fisheries and Oceans. Fixed stations (total of 28), as well as some additional non-standard stations, were sampled across shelves and in the deep central basin on the AR7W section or slightly north or south of this transect (Fig. 1)."

RC2.8 -Method

The inclusion of results in section 2.4 surprised me at first, but I think that this section is peripheral to the main story and is appropriate here.

AC2.8 - We agree with the reviewer.

RC2.9 -Results: I was frustrated by the fact that CHEMTAX results were presented only at the community level as defined through cluster analysis – but what was happening with the individual taxa that comprised these communities? Later I discovered that these results were (sort of) presented in the Discussion. I suggest that the distributions of individual taxa should be presented (with figures) before the distributions of communities.

AC2.9 - Information about individual taxa has now been inserted as a subsection "4.2 Chemtax interpretation and groups distributions" in the results section as suggested by the reviewer. See the attached pdf with the new figure and the new subsection.

RC2.10 - I would like to see a more detailed analysis of the factors controlling phytoplankton in each water mass. Even though there was considerable data on photosynthetic properties, I didn't get a clear message on the role of light in controlling biomass.

AC2.10 - MLD and stratification are included as indirect indicators of light availability. Unfortunately PAR was not measured at all stations during the 10 years of sampling. One paragraph (photosynthetic parameters) in the discussion (see paragraph rewritten in the response to the reviewer #3, AC3.60), now covers the effect of light on taxonomic segregation. Further analysis of nutrient and light variability across the Labrador Sea and its impact on phytoplankton composition is discussed in Fragoso et al. (2016).

RC2.11 -The Results should include a specific section on the temporal sequence, possibly exploring the sequence of events in each region. I note in Fig 3 that the data for 2012 and 2014, which were sampled late in the season, differ from other years, particularly Chl and nutrients in the central region.

AC2.11 - Unfortunately, there is not enough information to provide a true temporal sequence of data. Although there is some temporal variability in when the section was sampled, as discussed in the paper, the main variability is spatial as Figure 3 clearly shows. A figure has now been added showing the sampling day on the Z axis in figure 3. See the new version of figure 3 in the pdf file attached.

RC2.12 -Discussion: Much of the discussion about individual taxa in section 4.1 should be first described in the Results section.

AC2.12 - Information about individual taxa have been now added to the Results section. See new figure 4 and text in the pdf.

RC2.13 -Most of sections 4.2 and 4.3 should be saved for another paper.

AC2.13 - The reasons why we have kept the biogeochemical and photophysiological data in the revised manuscript are that they directly pertain to the aim and focus of our paper (see previous comments).

RC2.14 -The Discussion should focus specifically on the results of this paper in relation to the Aim, only referring to other studies to provide context, generally in the style of "Our results match those of Smith and Jones...". Only then should the wider implications of the work be discussed, and there should be clear signals when the narrative extends beyond the current work. Much of this Discussion reads like a review. It was often difficult to determine whether the results being discussed were from this paper or from others.

AC2.14 - We agree with the reviewer and have now considerably revised the discussion to focus on our results and improve the interpretation of our data in comparison to other

studies.

RC2.15 -Conclusions: Most of the final paragraph seems more appropriate to the Introduction. The authors may also consider any further research questions that arise from this study.

AC2.15 - This last paragraph was removed from the conclusions.

RC2.16 -Abstract: I think the first sentence is redundant and that the second sentence should be extended to include the Aim. The abstract will require revision in line with the changes to the rest of the manuscript.

AC2.16 - We have now changed the beginning of the abstract to reinforce the aim and the importance of the study. See the beginning of the abstract rewritten below.

Line 12 - "Abstract. The Labrador Sea is an ideal region to study the biogeographical, physiological and biogeochemical implications of phytoplankton communities due to sharp transitions between distinct water masses across its shelves and central basin. The aim of this study is to provide a baseline description of the distributions and biogeochemical traits of phytoplankton communities from distinct biogeographical regions of the Labrador Sea. We have investigated the multi-year (2005-2014) distributions of late spring and early summer (May to June) phytoplankton communities in the various hydrographic settings of the Labrador Sea. Our analysis is based on pigment markers (using CHEMTAX analysis), and photophysiological and biogeochemical characteristics associated with the communities present in the different water masses of the Labrador Sea."

RC2.17 -SPECIFIC COMMENTS: Line 186 and Table 3: Lutein not used for chlorophytes? (Does the BIO method separate ZEA & LUT?) If not, Table 3 ZEA must be ZEA+LUT

AC2.17 - The BIO method does not separate lutein and zeaxanthin so we renamed it to Zea + Lut

RC2.18 -Lines 192-200 and Figure 2: I note that two of the categories include Hex but no Chlc3 – I assume this is a simplification of the text and diagram as this combination does not exist to my knowledge. Figure 2 is unnecessary and should be replaced with a table including all pigments.

AC2.18 - Figure 2 shows the percentage contribution of each pigment to each cluster. In this study, Phaeocystis pouchetii did not contain 19-hex and was identified using Chl C3. This has previously been observed in the Labrador Sea (Stuart et al., 2000) and is stated in line 589 – 594.

RC2.19 -Section 3.2: Did the authors try further subdivision of group C3b? This group is by far the biggest, it is widest spread across the S-T diagram (Fig 5a), and its composition is "mixed", yet Fig 4a shows major divisions within the group. Would these subdivisions distinguish communities that were more coherent in composition and habitat?

AC2.19 - Cluster C3b had the highest level of similarity in terms of sample composition, although it was the most "mixed" in terms of community structure and the most widespread group in the Labrador Sea. Hence, we decided to leave it as is because a further division would bring information that we consider unnecessary, since they are, according to the Bray-Curtis similarity values (73 %), the most equal when compared to the other clusters.

RC2.20 -Line 316: change "Phaeocystis (cluster B)" to "A community dominated by diatoms and Phaeocystis (cluster B)". This is an important consideration throughout the document —- e.g. lines 328, 329 – there is not a careful distinction between the cluster groups (communities) and the taxa comprising them. I would invent an acronym or abbreviation for each community to avoid this confusion.

AC2.20 - Changes updated. We agree with the reviewer that it is important to elucidate that we are referring to multiple taxa in the community in the text. Therefore, we have now rewritten these sentences for clarification.

RC2.21 -Line 527: The possibility that "diatom species from both Arctic and Atlantic waters varied intrinsically in pigment composition" can be supported by consulting Table 3 of this paper, where we see that they do.

AC2.21 - This is true but our argument is that diatom composition (polar versus Atlantic species) might be influencing these discrepancy. See our response to the reviewer #1 (AC1.4).

RC2.22 -Line 551: "chlorophytes were present in high concentrations on the Labrador Shelf, which may explain the discrepancy between these results." Some more details are required to constitute an explanation.

AC2.22 - This sentence has been completely rewritten for clarification. See the response to the reviewer #3 where we include the new version of this paragraph (AC3.60)

RC2.23 -Table 5: This table should be augmented by information on the region in which each cluster is found, and the major taxonomic components.

AC2.23 - We believe that this might confuse the reader. The taxonomic components are already provided in Figure 4 and should be examined in parallel with Table 5.

RC2.24 - Also expressing the values like Temperature with standard errors is inappropriate. The values are not based on repeat measurements of a single parameter –e.g. Cluster 3b is listed as 3.4+/-0.2 C, but the actual range is from about -1.3 to +8, the widest of any group. I would be surprised if the standard error given is correct. Even if is, it is meaningless. This table should list the range for each cluster instead.

AC2.24 - We now included standard deviations rather than standard errors in Table 5. A table of data ranges for the parameters discussed for each cluster would be strongly influenced by outliers, hence we have chosen to retain the averages. However, we have added a table with the parameter ranges in the Supplementary material.

RC2.25 -Also: I didn't see any reference to the data for DT:(DT+DD) in text (nor was there any reference to how long the filters were held between sample collection and

freezing. This should be < 5-10 min for this parameter to be valid).

AC2.25 - We have now included information on the filtering time. See our response to reviewer #1 (AC 1.16).

RC2.26 -Results: I did not notice any indication that the raw pigment data were to be included in Supplementary Material or an online databank. I would hope that this will be the case to increase the value of this data set.

AC2.26 Some of these data (from Bedford Institute of Oceanpgraphy) are publically available online (http://www.dfo-mpo.gc.ca/science/data-donnees/biochem/index-eng.html). We are discussing with the co-authors the possibility of submitting additional data to PANGEA.

RC2.27 -TECHNICAL COMMENTS: Line 67 and throughout: References should be cited in order of date – oldest to newest

AC2.27 - For Biogeosciences, in case of co-authors papers, citation should be first alphabetically according to the second author's last name, and then chronologically within each set of co-authors.

RC2.28 -Line 84: change "while" to "but"

AC2.28 - Changed.

RC2.29 -Line 118: inset "wide" after "km" (twice)

AC2.29 - Changed.

RC2.30 -Line 123: change "fresh" to "low salinity". Rest of same paragraph: three water masses are described as "warm and salty" or "cold, low salinity" but other water masses lack these descriptions (parallel form required– see below). Also, is the warm arrow parallel to the Labrador Current in Fig 1 considered to be part of that current?

AC2.30 - This whole paragraph has changed, however, we have described the waters

masses in an orderly manner as suggested by the reviewer. See the response to the reviewer #3 (AC3.24). The red arrow in Figure 1 (lighter in colour) represents a modified (cooled and freshened) branch of the IC through lateral and vertical mixing following the Labrador slope. We will include the modified figure in the new version of the manuscript for clarification.

RC2.31 -Line 177: The correct reference for the method ascribed to "Coupel et al. (2015)" is Higgins et al (2011).

AC2.31 - We have actually updated this reference to Wright et al., (2009). See response to reviewer #1 (AC 1.27) where we include the new version of this paragraph.

RC2.32 -Line 316: Add "respectively" after "(IC)"?

AC2.32 - Changed.

RC2.33 -Line 325: Replace "respond strongly to" with "are associated with" and "spatial aspects of the data" with "environmental parameters"

AC2.33 - Changed.

RC2.34 -Line 331: The description of Fig 5b could hardly be more obscure: "In Atlantic waters, temporal aspects of the data were also observed (upper and lower right quadrants (Fig. 5b))." There is nothing in that figure that implies a temporal sequence. It was only when the Conclusions mentioned clear temporal differences that I searched the document for "temporal" to find what I had missed and came back to this figure. After some cross-referencing I realised that the description should have read, "In Atlantic waters (upper and lower right quadrants (Fig. 5b)), the phytoplankton community was composed of mixed taxa during May (orange circles), but became dominated by diatoms and dinoflagellates during the bloom in June (red circles), showing a clear temporal succession in these waters". More generally, the authors must not rely on the reader to discern what is in a figure. The reader is not familiar with the data and may not see what the author sees, or they may see something different. Whatever story

exists in the figure, it must be stated clearly in text as part of the narrative. The figure supports the narrative, it does not replace it.

AC2.34 - We have now clarified the temporal succession of the spring bloom in the Labrador Sea in the text (see response AC2.7). Lastly, we have changed this sentence, now following the suggestion of the reviewer.

RC2.35 -Line 368: Replace "lower accessory pigments to TChla ratio" with "lower ratio of accessory pigments to Tchla"

AC2.35 - Changed.

RC2.36 -Line 369: Replace "(Fig. 7b). Furthermore, communities from warmer waters (Irminger Current from Atlantic origin), particularly those co-dominated by diatoms and dinoflagellates had " with "(Fig. 7b) than communities from warmer waters (Irminger Current from Atlantic origin), particularly those co-dominated by diatoms and dinoflagellates which had"

AC2.36 - Changed.

RC2.37 - Line 376: Replace "$\mu$g C $\mu$g Chla h-1W m-2" with "$\mu$g C $\mu$g Chla h-1 W-1 m2" or " (Wm-2)-1 " Also line 378

AC2.37 – All changed.

RC2.38 -Lines 375 to 386. Sentences should be rearranged to "parallel form" i.e. talk about the same things in the same order for each case cited

AC2.38 – We have rewritten this whole paragraph. See the response to the reviewer #3 (AC3.60) where we show how this paragraph would be in the new manuscript version.

RC2.39 -Line 392: Insert "Atlantic," before "Labrador"

AC2.39 - Changed.

RC2.40 -Lines 437 – 450: Reads like a review. Note also that the paragraph starts with

"Phaeo- cystis and diatoms. . . (Fragoso etal 2016)" but by line 441 it's "PRESUMABLY of Phaeocystis and diatoms (Fragoso etal 2016)". Also is "eastern central Labrador Sea" (line 437) equivalent to "West Greenland Current" (line 440)?

AC2.40 - We have now changed the beginning of this paragraph. See below.

Line 437 - In this study, Phaeocystis and diatoms were observed blooming together in waters of the WGC, in the eastern central part of the Labrador Sea. The occurrence of Phaeocystis in these waters has been observed before by several authors (Fragoso et al., 2016; Frajka-Williams and Rhines, 2010; Harrison et al., 2013; Head et al., 2000; Stuart et al., 2000; Wolfe et al., 2000). The eastern part of the Labrador Sea is a region with high eddy kinetic energy during spring (Chanut et al., 2008; Frajka-Williams et al., 2009; Lacour et al., 2015), which causes the accumulation of low-salinity surface waters from the West Greenland Current. This buoyant freshwater layer contains elevated levels of biomass of both Phaeocystis and diatoms (this study, Fragoso et al., 2016).

RC2.41 -Line 598: Add reference e.g Gieskes and Kraay (1983) Mar. Biol. 75, 179-185.

AC 2.41 - Suggested reference added.

RC2.42 -Line 886: remove "et al" ; page numbers = 78 – 80

AC 2.42 - Changed.

RC2.43 -Figure 2 is unnecessary and should be replaced with a table including all pigments.

AC 2.43 - We believe that Figure 2 is key to the CHEMTAX analysis, although we have now included it in the supplemental material rather than the main text.

RC2.44 -Figure 4b. The colours of the sectors would be much more easily interpreted if they made sense to a phycologist! Surely cyanobacteria = Cyan, chlorophytes = Dk

Green, Prasinophytes = Lt Green, Phaeocystis = Brown, etc. (Leave diatoms white)

AC 2.44 - Although we appreciate the colour selection of the reviewer, we have retained the original pastel colours in 4b as the colours suggested are already used elsewhere in Figure 4 (4a, 4c), which could confuse the reader.

RC2.45 -Figure 4c. The single circle as a scale is ambiguous. Does the biomass relate to the diameter or the area of the circle? In any case it's difficult to judge. There should be a range of circles representing a biomass scale (if circles are to be used). Also I estimate that about 20% of the data points are hidden in this diagram as they underlie another circle. This could be solved by increasing the breadth of the figure or using vertical bars instead of circles. Could the fronts be marked for each year by dotted lines?

AC 2.45 - A scale for the bubbles used in this figure has now been added, as well as the width of the figure (see attached pdf). Physical fronts are already discernible through sharp changes in phytoplankton community composition, as mentioned in line 305, whereas dotted lines would become confusing between adjacent years.

RC2.46 -Figure 5: It would be good to see individual taxa plotted in such diagrams.

AC 2.46 –We could add the "arrows" of individual taxa in the figure 5b, however, we decided to leave it as is because adding information on taxa would be confusing to interpret the message of the figure. That is because, diatoms, for example, are the dominant taxa in all communities (except cluster C3b), so the "diatom arrow" in just one direction could bias the interpretation. We are focused in the community, rather than individual taxa in this paper.

RC2.47 -Table 2 is unnecessary. The individual pigments are not part of the story – simply quote the references.

AC 2.47 - We believe that individual pigment information is important for the CHEMTAX analysis, though we have now moved this table to the supplementary material.

RC2.48 -Table 3: The legend doesn't make it clear that the references cited provided the starting ratios from which these data were calculated. Cyanobacteria is misspelt.

AC 2.48 - We have now clarified in the legend that the references provided inform the starting ratios from which the data were calculated. See the pdf attached in the response letter for reviewer #1, where we have added a new version of this table. "Cyanobacteria" has now been spelt correctly.

RC2.49 -Table 4: The formatting is strange. It looks as if it should be split into A & B, horizontally.

AC 2.49 - Table 4 has now been split horizontally into a) and b) for better clarification. See the new version of the table and the response to the reviewer #3 (AC3.46).

Please also note the supplement to this comment:
http://www.biogeosciences-discuss.net/bg-2016-295/bg-2016-295-AC2-supplement.pdf

**Supplement:**

**1. Introduction**

Marine phytoplankton form a taxonomically and functionally diverse group, where communities are structured by a variety of factors, including nutrient and light availability, predation and competition for resources (Litchman and Klausmeier, 2008). Environmental heterogeneity, thus, creates biogeographical patterns of abundance, composition, traits and diversity of phytoplankton communities in the global ocean (Barton et al., 2013; Follows et al., 2007; Hays et al., 2005). Phytoplankton communities within a biogeographical region are subject to similar environmental conditions, such as temperature (Bouman et al., 2003), nutrient concentration (Browning et al., 2014) and irradiance (Arrigo et al., 2010). These environmental factors, along with phytoplankton composition itself (Bouman et al., 2005), affect the overall photophysiological response and bulk primary productivity of the phytoplankton community.

Biogeography of phytoplankton communities and their photophysiological characteristics, consequently, impact the structure of marine ecosystems due to their functional community role in biogeochemical cycling and transfer of energy to higher trophic levels. For example, distinct phytoplankton assemblages have been reported to influence differently particulate (Martiny et al., 2013a, 2013b; Smith and Asper, 2001) and dissolved elemental stoichiometry (C:N:P)(Weber and Deutsch, 2010), the drawdown of gases (Arrigo, 1999; Tortell et al., 2002) and the efficiency of carbon export (Guidi et al., 2009; Le Moigne et al., 2015). Patterns of phytoplankton stoichiometry has been assumed to be consistent phylogenetically and within higher taxonomic levels (Ho et al., 2003; Quigg et al., 2003). Nonetheless, phytoplankton stoichiometry has been reported to vary according to nutrient supply ratios (Bertilsson et al., 2003; Rhee, 1978), as well as phenotypically within species across the same population (Finkel et al., 2006).

The sub-Arctic North Atlantic is a complex system with contrasting hydrography that structures plankton communities within distinct biogeographical provinces (Fragoso et al., 2016; Head et al., 2003; Li and Harrison, 2001; Platt et al., 2005; Sathyendranath et al., 2009, 1995). Biogeographical regions of the Labrador Sea shape phytoplankton community composition (Fragoso et al., 2016), bio-optical properties (Cota, 2003; Lutz et al., 2003; Platt et al., 2005; Sathyendranath et al., 2004; Stuart et al., 2000) and the seasonality of phytoplankton blooms (Frajka-Williams and Rhines, 2010; Lacour et al., 2015; Wu et al., 2008, 2007). Phytoplankton blooms, for example, occur first (April to early May) in the shelves due to haline-driven stratification driven by the input of Arctic-related waters, in addition to rapid sea ice melt in the Labrador Shelf near Canada (Frajka-Williams and Rhines, 2010; Wu et al., 2007). The central Labrador bloom occurs later in the season (late May to June) as result of thermal stratification (Frajka-Williams and Rhines, 2010). Fragoso et al. (2016) showed that the biogeography of phytoplankton communities in the Labrador Sea during spring and early summer is shaped by distinct species found Atlantic or Arctic waters, which may have distinct impact on the biogeochemical cycles and transfer of energy to upper trophic level. However, these authors focused in taxonomy and investigated only larger phytoplankton ($> 4\mu m$). The photophysiological and biochemical signatures, such as stoichiometry (C:N ratio) of these distinct 
[revised manuscript text omitted]
. 4a). Chlorophytes and prasinophytes were common in the center-western part (Fig. 4b,c), whereas *Phaeocystis* was abundant at the eastern part of the Labrador Sea (Fig. 4d). Dinoflagellates were abundant in the center region of the Labrador Sea (Fig. 4e). Other prymnesiophytes, including coccolithophores and *Chrysochromulina* were also common at the center part of the Labrador Sea (Fig. 4f). Overall, chrysophytes and pelagophytes were found in low abundance in the Labrador Sea, except at the center region of the Labrador Sea during 2011 (Fig. 4g). Cyanobacteria was more abundant at the Labrador Slope and Greenland Shelf and during some years (2005 and 2012) at the center Labrador Sea (Fig. 4f). Cryptophytes comprised less than 10% of total phytoplankton chlorophyll concentrations (data not shown).

[Figure]

**Figure 4 – Relative contribution (%) of chlorophyll a from distinct phytoplankton classes at each station from 2005 to 2014 along the section distance from Labrador coast represented in Figure 3a (star symbol in a). LSh = Labrador Shelf, LSl = Labrador Slope, CB = Center Basin, GSl = Greenland Slope, GSh = Greenland Shelf. Note the distinct scales for each group.**

**Table 4 – Results of the Redundancy Analyses (RDA) with the eigen-values, taxa-environmental correlations and percentages of variance explained used in the analysis (a). Automatic forward selection (a *posteriori* analysis) was used to determine the environmental variable(s) that best explain the variance of the data (b). The subset of environmental variable(s) that significantly explained phytoplankton distribution are referred to marginal effects ($\lambda_1$) when analysed individually, or conditional effects ($\lambda a$) when analysed additively in the model (b). Explanatory variables are temperature (°C), salinity, nitrate ($NO_3^-$; µmol $L^{-1}$), phosphate ($PO_4^{3-}$; µmol $L^{-1}$), silicate ($Si(OH)_4$; µmol $L^{-1}$) and Stratification Index (SI) (kg $m^{-4}$). Significant p-values ($p < 0.05$) represents the variables that explain the variation in the analyses.**

| a) Axes | 1 | 2 | 3 | 4 | Total variance |
|---|---|---|---|---|---|
| Eigen-values | 0.26 | 0.04 | 0.005 | 0 | 1 |
| Taxa-environment correlations | 0.68 | 0.4 | 0.321 | 0.25 | |
| Cumulative percentage variance | | | | | |
| of species data | 25.7 | 29.9 | 30.3 | 30.7 | |
| of species-environment relation | 83.5 | 97.2 | 98.8 | 99.8 | |
| | | | | | |
| Sum of all eigenvalues | | | | | 1 |
| Sum of all canonical eigenvalues | | | | | 0.31 |

| b) Marginal Effects | | Conditional Effects | | | |
|---|---|---|---|---|---|
| Variable | $\lambda_1$ | Variable | $\lambda_a$ | *P* | *F* |
| $Si(OH)_4$ | 0.2 | $Si(OH)_4$ | 0.2 | 0.001 | 61.7 |
| $NO_3^-$ | 0.19 | Temperature | 0.05 | 0.001 | 17.3 |
| $PO_4^{3-}$ | 0.17 | Salinity | 0.02 | 0.002 | 6.94 |
| Salinity | 0.09 | $NO_3^-$ | 0.01 | 0.016 | 4.31 |
| Temperature | 0.07 | $PO_4^{3-}$ | 0.02 | 0.002 | 7.22 |
| SI | 0.06 | SI | 0.01 | 0.153 | 1.72 |

**Table 5** – Average, standard deviations and number of observations (in parenthesis) of environmental and biological variables of each cluster group. MLD = mixed layer depth, SI= Stratification index, $NO_3^-$ = nitrate, $PO_4^{3-}$ = phosphate, $Si(OH)_4$ = silicate, DT= diatoxanthin, DD= diadinoxanthin, POC= particulate organic carbon, PON= particulate organic nitrogen, $POC_{phyto}$ = phytoplankton-derived particulate organic carbon, $\alpha^B$ = initial slope of the photosynthesis-irradiance curve, $P_m^B$ = maximum normalised photosynthesis, $E_k$ = half-saturation irradiance, $Es$ = saturation irradiance.

| | Cluster A | | Cluster B | | Cluster C3a | | Cluster C3b | | Cluster C2 | | Cluster C1 | |
|---|---|---|---|---|---|---|---|---|---|---|---|---|
| Temperature (°C) | 2.8 ± 2.4 | (17) | 2.0 ± 1.8 | (46) | 1.6 ± 1.9 | (62) | 3.4 ± 1.9 | (92) | 4.8 ± 1.5 | (32) | 1.4 ± 1.7 | (4) |
| Salinity | 33.4 ± 1.5 | (17) | 33.7 ± 0.8 | (46) | 33.1 ± 1.2 | (62) | 34.1 ± 1.0 | (92) | 34.4 ± 0.5 | (32) | 33.0 ± 1.6 | (4) |
| MLD (m) | 32.2 ± 43.8 | (17) | 32.6 ± 23.4 | (46) | 31.2 ± 28.5 | (62) | 59 ± 71.1 | (92) | 29.8 ± 17.0 | (32) | 16.0 ± 4.2 | (4) |
| SI x $10^{-3}$ (kg m$^{-4}$) | 9.1 ± 6.3 | (17) | 6.3 ± 5.7 | (46) | 10.7 ± 8.5 | (62) | 5.0 ± 6.8 | (92) | 6.1 ± 4.5 | (31) | 6.6 ± 8.5 | (4) |
| $NO_3^-$ (µmol L$^{-1}$) | 2.9 ± 4.7 | (17) | 2.7 ± 3.5 | (46) | 3.4 ± 4.3 | (58) | 8.4 ± 4.1 | (83) | 3.7 ± 3.9 | (32) | 3.8 ± 6.8 | (4) |
| $Si(OH)_4$ (µmol L$^{-1}$) | 2.2 ± 2.7 | (17) | 2.8 ± 2.1 | (46) | 3.5 ± 2.4 | (58) | 5.4 ± 2.2 | (83) | 3.0 ± 2.2 | (32) | 2.3 ± 3.4 | (4) |
| $PO_4^{3-}$ (µmol L$^{-1}$) | 0.3 ± 0.3 | (17) | 0.3 ± 0.2 | (45) | 0.4 ± 0.2 | (55) | 0.7 ± 0.2 | (79) | 0.3 ± 0.2 | (32) | 0.4 ± 0.3 | (4) |
| $Si(OH)_4$:$NO_3^-$ | 6.0 ± 11.8 | (14) | 3.6 ± 7.9 | (37) | 8.5 ± 18.2 | (54) | 1.1 ± 1.5 | (82) | 1.6 ± 1.8 | (32) | 3.9 ± 4.4 | (4) |
| $NO_3^-$:$PO_4^{3-}$ | 8.2 ± 6.7 | (11) | 5.2 ± 5.0 | (45) | 5.9 ± 5.8 | (55) | 11.4 ± 4.1 | (79) | 8.7 ± 4.6 | (32) | 5.5 ± 7.1 | (4) |
| Chlorophyll a (mgChl$a$ m$^{-3}$) | 3.8 ± 4.7 | (17) | 5.5 ± 4.8 | (45) | 7.7 ± 5.6 | (59) | 2.0 ± 1.7 | (91) | 4.0 ± 1.8 | (31) | 8.8 ± 9.6 | (4) |
| DT:(DT+DD) | 0.01 ± 0.03 | (16) | 0.02 ± 0.05 | (44) | 0.04 ± 0.05 | (62) | 0.10 ± 0.10 | (92) | 0.08 ± 0.07 | (32) | 0.02 ± 0.04 | (4) |
| (DD+DT):Chla | 0.08 ± 0.07 | (17) | 0.03 ± 0.03 | (46) | 0.04 ± 0.02 | (62) | 0.07 ± 0.03 | (92) | 0.12 ± 0.03 | (32) | 0.07 ± 0.04 | (4) |
| POC (mgC m$^{-3}$) | 245 ± 90 | (4) | 498 ± 198 | (27) | 533 ± 198 | (45) | 234 ± 145 | (63) | 512 ± 179 | (15) | 392 ± 418 | (2) |
| PON (mgN m$^{-3}$) | 39 ± 16 | (4) | 65 ± 23 | (27) | 74 ± 30 | (45) | 37 ± 26 | (64) | 84 ± 33 | (15) | 42 ± 41 | (2) |
| $POC_{phyto}$ (%) | 23.0 ± 5.2 | (4) | 49.2 ± 29.5 | (26) | 60.9 ± 25.6 | (44) | 33.3 ± 10.1 | (64) | 36.0 ± 11.4 | (15) | 37.8 ± 1.3 | (2) |
| POC:PON | 6.5 ± 1.2 | (4) | 7.8 ± 2.1 | (27) | 7.5 ± 2.1 | (45) | 6.6 ± 1.3 | (64) | 6.2 ± 0.9 | (15) | 8.6 ± 1.6 | (2) |
| $\alpha^B$ (mgC mgChl$a^{-1}$ h$^{-1}$ W$^{-1}$m$^2$) x$10^{-2}$ | - | | 6.8 ± 6 | (9) | 9.2 ± 5 | (10) | 7.1 ± 4 | (18) | 7.1 ± 1.5 | (4) | - | |
| $P_m^B$ (mgC mgChl$a^{-1}$ h$^{-1}$) | - | | 2.9 ± 1.1 | (9) | 2.3 ± 0.8 | (10) | 2.3 ± 0.6 | (18) | 3.2 ± 0.7 | (4) | - | |
| $E_k$ (Wm$^{-2}$) | - | | 60 ± 33 | (9) | 29 ± 13 | (10) | 39 ± 14 | (18) | 46 ± 5 | (4) | - | |
| $Es$ (Wm$^{-2}$) | - | | 62 ± 32 | (9) | 35 ± 18 | (10) | 43 ± 18 | (18) | 56 ± 8 | (4) | - | |
| $\beta$ (mgC mgChl$a^{-1}$ h$^{-1}$ W$^{-1}$m$^2$) x$10^{-4}$ | - | | 4 ± 7 | (9) | 16 ± 23 | (10) | 10 ± 16 | (18) | 29 ± 24 | (4) | - | |

30 **Supplemental Material - Table S2. Range of environmental and biological variables of each cluster group. MLD = mixed layer depth, SI= Stratification index, $NO_3^-$ = nitrate, $PO_4^{3-}$ = phosphate, $Si(OH)_4$ = silicate, DT= diatoxanthin, DD= diadinoxanthin, POC= particulate organic carbon, PON= particulate organic nitrogen, $POC_{phyto}$ = phytoplankton-derived particulate organic carbon, $\alpha^B$ = initial slope of the photosynthesis-irradiance curve, $P_m^B$ = maximum normalised photosynthesis, $E_k$ = half-saturation irradiance, $Es$ = saturation irradiance.**

| | Cluster A | Cluster B | Cluster C3a | Cluster C3b | Cluster C2 | Cluster C1 |
|---|---|---|---|---|---|---|
| Temperature (°C) | - 1.1 - 6.7 | - 1.1 - 5.7 | - 1.2 - 5.6 | - 1.4 - 7.9 | 0.9 - 6.8 | - 0.6 - 3.4 |
| Salinity | 30.2 - 34.9 | 31.8 - 34.8 | 30.4 - 34.8 | 31.3 - 35 | 32.8 - 34.8 | 31.4 - 34.7 |
| MLD (m) | 14 - 196 | 11 - 105 | 11 - 156 | 11 - 531 | 11 - 64 | 12 - 21 |
| SI x $10^{-3}$ (kg $m^{-4}$) | 0.03 - 24 | 0.1 - 27 | 0.008 - 40 | 0.008 - 32 | 0.5 - 19 | 1 - 19 |
| $NO_3^-$ (µmol $L^{-1}$) | 0 - 15.2 | 0 - 16.0 | 0 - 15.9 | 0 - 15.1 | 0.2 - 13.7 | 0.1 - 13.9 |
| $Si(OH)_4$ (µmol $L^{-1}$) | 0 - 7.8 | 0.3 - 8.5 | 0 - 8.6 | 0.9 - 8.8 | 0.4 - 8.0 | 0.2 - 7.4 |
| $PO_4^{3-}$ (µmol $L^{-1}$) | 0.1 - 1.1 | 0.1 - 1.0 | 0 - 1.0 | 0.2 - 1.0 | 0.1 - 0.9 | 0.2 - 0.9 |
| $Si(OH)_4:NO_3^-$ | 0.2 - 44.1 | 0.4 - 45.4 | 0 - 87.3 | 0.1 - 9.3 | 0.6 - 9.5 | 0.2 - 9.4 |
| $NO_3^-:PO_4^{3-}$ | 0.3 - 19.1 | 0 - 15.9 | 0 - 17.3 | 0 - 20.8 | 1.1 - 16.2 | 0.2 - 15.4 |
| Chlorophyll a (mgChl$a$ $m^{-3}$) | 0.6 - 20.2 | 0.7 - 19.7 | 1.1 - 24.1 | 0.4 - 9.0 | 0.4 - 9.1 | 0.9 - 22.8 |
| DT:(DT+DD) | 0 - 0.1 | 0 - 0.2 | 0 - 0.2 | 0 - 0.4 | 0 - 0.3 | 0 - 0.1 |
| (DD+DT):Chla | 0 - 0.3 | 0 - 0.1 | 0 - 0.1 | 0 - 0.2 | 0.1 0.2 | 0 - 0.1 |
| POC (mgC $m^{-3}$) | 119 - 331 | 178 - 952 | 160 - 960 | 55 - 658 | 211 - 796 | 97 - 688 |
| PON (mgN $m^{-3}$) | 20 - 54 | 29 - 109 | 24 - 154 | 4 - 138 | 38 - 154 | 13 - 71 |
| $POC_{phyto}$ (%) | 17.9 - 29.0 | 14.9 - 100* | 8.0 - 100* | 12.2 - 64.6 | 18.8 - 56.6 | 36.9 - 38.7 |
| POC:PON | 5.2 - 7.9 | 5 - 12.5 | 4.6 - 13.4 | 4.7 - 12.4 | 5.2 - 8.5 | 7.5 - 9.7 |
| $\alpha^B$ (mgC mgChl$a^{-1}$ $h^{-1}$ $W^{-1}m^2$) x$10^{-2}$ | - | 2 - 18 | 4 - 17 | 2 - 17 | 5 - 9 | - |
| $P_m^B$ (mgC mgChl$a^{-1}$ $h^{-1}$) | - | 1.0 - 4.7 | 1.2 - 4.0 | 0.9 - 3.2 | 2.5 - 4.0 | - |
| $E_k$ ($Wm^{-2}$) | - | 20 - 127 | 15 - 52 | 16 - 67 | 41 - 51 | - |
| $Es$ ($Wm^{-2}$) | - | 21 - 127 | 15 - 59 | 17 - 76 | 45 - 62 | - |
| $\beta$ (mgC. mgChl$a^{-1}$ $h^{-1}$ $W^{-1}m^2$) x$10^{-4}$ | - | 0 - 16 | 0 - 75 | 0 - 49 | 5 - 60 | - |

* Values > 100% due to variability of the data was set to a maximum value of 100%.

---

## Author Comment (AC3) · 10 Oct 2016

We thank the reviewer for his/her comments and suggestions, which we feel have greatly improved the manuscript. Below we respond to each comment in detail. RC refers to "Reviewer's Comments" and AC to "Author's comments". We have enumerated the reviewer's comments to organise better our responses.

Reviewer #3:

RC3.1 - The manuscript "Spring phytoplankton communities of the Labrador Sea (2005-2014): pigments signatures, photophysiology and elemental ratios" present a time series of pigments and nutrients data in the Labrador Sea from 2005 to 2014. The

authors use the CHEMTAX method to interpret the pigment dataset in term of phytoplankton groups and then to describe the distribution of these phytoplankton groups. Oceanographic provinces of the Labrador Sea are identified using on physical and biogeochemical parameters as well as phytoplankton diversity. Several statistical approaches based on clustering, ordination plot and regression were used to link the distribution in time and space of the phytoplankton with the environmental parameters. Finally, several physiological parameters related to the phytoplankton communities were measured (P curves, POC/PON, POC/POC Chla) or extract from the pigments distribution (AP/Chla, photoprotective pigments). The physiological information is used to go further in the explanation of the link between the phytoplankton community's distribution and the environmental conditions.

General comments: The introduction is not well structured and full of too heavy and unclear sentence.

AC3.1 - We have now rewritten and reduced the introduction to provide better focus. Please see the response to reviewer #2. We have attached the new version of the introduction as a pdf file.

RC3.2 - But, the manuscript goes better in the result and discussion section. The results section is clear with a good choice of graph. Sometimes, it was difficult to get the point of the use of methods and the information that sort from some data.

AC3.2 - We have now changed and improved the methods section for better clarification by adding further explanation of the use of the different methods to examine the data.

RC3.3 - Finally, the discussion put together in a clear way all the information in the results section and brings interesting information to parameters that were of unclear utility in the result section. The authors highlight the specificity of the species and explained their success in the different regions and use well the comparison with the literature. I recommend important change in the introduction to make it more fluent, to better extract the key information and topics of each sub-paragraph. The sentences are

generally way too long and confusing. Most of them could be cut in two parts. There are several mistakes on the use of superlative in the results section. The discussion is well conducted and uses interestingly the results

AC3.3 – Thanks for the suggestions. The introduction has been shortened and sentences are now condensed.

RC3.4 - Specific comments by section Introduction L51: better to use "structure"

AC3.4 - Changed.

RC3.5 - L51: change the order to "functional role in the community"

AC3.5 - We have removed "in the community" to avoid redundancy.

RC3.6 - L 54 to 59: there is some redundancy with the lines 51-53

AC3.6 - We have now changed/reduced the introduction and shortened the sentences, so this redundancy does not exist anymore.

RC3.7 - L59 to 64: Unclear about the conservation or not of the stoichiometry. You said the "stoichiometry is consistent phylogenetically" and latter you mentioned, "they may vary (. . .) phenotypically within species". Be more precise on when the ratios are conserved or not.

AC3.7 - Further clarification, these sentences have been rewritten. They would be rewritten as such:

Line 59 – "Patterns of phytoplankton stoichiometry may be consistent phylogenetically and within higher taxonomic levels (Ho et al., 2003; Quigg et al., 2003). However, phytoplankton stoichiometry has also been reported to vary according to nutrient supply ratios (Bertilsson et al., 2003; Rhee, 1978), as well as phenotypically within species from the same population (Finkel et al., 2006)." RC3.8 - L70 "shelves and the basin"

AC3.8 - This sentence has been removed.

RC3.9 - L75-76: I don't think the interest to study the phytoplankton is to use it as an index of waters masses since simple parameters as temperature and salinity did a good job. It appears to me more important to highlight the possible importance of the biogeography on the biological pump, carbon export or the energy transfer to upper trophic level.

AC3.9 - We agree with the reviewer and have rewritten these lines to focus on ocean biogeochemistry and marine ecosystems. See below. Line 73. "Fragoso et al. (2016) showed that the biogeography of phytoplankton communities in the Labrador Sea during spring and early summer is shaped by distinct species found in Atlantic or Arctic waters, which may have a distinct impact on the biogeochemical cycles and transfer of energy to higher trophic levels. However, these authors focused on species taxonomy and investigated only the larger phytoplankton (> 4 $\mu$m). The photo-physiological and biogeochemical signatures, such as elemental stoichiometry (C:N ratio), of these spring phytoplankton communities occurring in distinct sectors of the Labrador Sea has not been previously investigated." RC3.10 - L78-84: The same idea is repeated. Please reduce the size of the sentence, too much utilization of the conjunction "and".

AC3.10 - This paragraph has been removed to shorten the introduction overall.

RC3.11 -L82: could simplify "high-latitude Arctic/Atlantic waters" by "polar waters".

AC3.11 - This paragraph has been removed.

RC3.12 -L100: redundancy with the line 88-90

AC3.12 - Line 88-90 refers to analysis of pigments using the HPLC while line 100 refers to CHEMTAX analysis of pigment data; hence we do not see them as redundant.

RC3.13 -L93: Please precise the concept of "functional cell size"

AC3.13 - This sentence has been removed from the introduction.

RC3.14 -L94-95: "assemblage dominance": wrong, it's the dominance of phytoplankton

groups and not assemblages

AC3.14 - This sentence has been removed from the introduction.

RC3.15 -L95: remove "however"

AC3.15 - Changed.

RC3.16 -L99: remove the comma.

AC3.16 - Changed.

RC3.17 -L107: "comprehensively understand" is a pleonasm.

AC3.17 - The word "comprehensively" was removed.

RC3.18 -L108-L111: you repeat the same information than the line 106-108.

AC3.18 - his paragraph has been reduced.

RC3.19 -Methods

There is some confusion on the water composition of the Labrador Sea. Moreover the authors depicted as well deep and shallow currents and water masses. The authors should focus on the surface and sub-surface water-masses and circulation since the pigment dataset presented here concerned only the upper 10m.

AC3.19 - We believe that the reviewer is referring to the Irminger Current (IC). The IC is described as a surface current (see Hauser et al., 2015; Yashayaev and Seidov, 2015), however the WGC may occasionally "slide" over the IC in the central-eastern part of the Labrador Sea and form a "tongue" of fresh, cold and less dense water. The lateral advection of this tongue (i.e. how offshore it goes) varies inter-annually during spring. We have used a T-S diagram to discern these water masses (IC, LC and WGC, see Figure 5a). As the reviewer has noted there were some relatively warm (> 3°C) and salty (> 34) water found at the surface. We refer to this as part of the IC, although it might have been slightly modified due to the highly dynamic features of surface waters, which includes influence of precipitation/evaporation, meltwater, riverine input and mesoscale eddies. Although the IC is "conserved" at mid-depth waters (200-600 m), it does reach the surface, however it becomes "modified" due to the factors already mentioned.

RC3.20 - L115: "transition zone between the Arctic and ..."

AC3.20 - Changed.

RC3.21 - L115: Newfoundland is not really the southern boundary. The North Atlantic is the southern boundary.

AC3.21 - We have now defined the limits of the Labrador Sea according to the International Hydrography Organisation. See below.

Line 115 – "It is bounded by Davis Strait to the north, a line from Cape St. Francis in Newfoundland (47°45' N, 52°27'W) to Cape Farewell (southern tip of Greenland) to the southeast and the coast of Labrador and Newfoundland to the west (Fig. 1) (International Hydrographic Organization, 1953)."

RC3.22 - L119: The lower limit of the Greenland Shelf (ie 2500m) sounds very deep to characterize a shelf! I think you characterize the extension of the Greenland Current here.

AC3.22 - We apologise for the confusion. We were referring to the Greenland shelf and slope and not just the shelf. We have corrected this now. See the sentence rewritten below.

Line 116 – "The bathymetry of the Labrador Sea can be subdivided into the wide continental shelf and relatively gentle continental slope on its western side (the Labrador Shelf, > 500 km and < 250 m deep) and narrow shelf and very steep continental slope on the eastern side (the Greenland Shelf and Slope, < 100 km and < 2500 m deep)."

RC3.23 - L122: remove "mostly"

AC3.23 - Changed.

RC3.24 - L122: The Irminger current is not the main water masses of the Labrador Basin since this current it is confined on the east and west borders of Labrador Basin at a mid-depth (200-600m). The Labrador Sea Water composes the water of the basin and their characteristics are mainly influenced by the winder convection with the deeper water masses (see the work of Yashayaev et al.).

AC3.24 - We apologise for the confusion in this section. We were referring to surface hydrography only. As discussed above (AC 3.19), the WGC often "slides" over the IC, creating a broad and thin layer of fresh and cold water, usually observed in the central-eastern section of the AR7W transects. On the western part of the section the IC intrudes into upper waters. This is observed in the T-S diagrams when salty (> 34) and warm (> 3°C) waters of Atlantic origin are found at the surface. We have rewritten this paragraph for clarification. See below.

Line 121. "The upper Labrador Sea (< 200 m) is comprised of waters originating from the North Atlantic and the Arctic (Yashayaev, 2009). Atlantic-influenced waters occur mostly in the central Labrador Sea, where waters are relatively warm, salty and mainly identified as the Irminger Current (IC). Cold, low salinity waters originate from the Arctic via the surrounding shelves and are mainly identified as the Labrador Current (LC) and the West Greenland Current (WGC) (Fig 1). Circulation in the central basin of the Labrador Sea is complex, often showing a gyre-like flow system that alternates in direction (Palter et al. 2016, Wang et al, 2016). The inshore branch of the LC overlies the Labrador Shelf and includes Arctic waters originating from Baffin Bay and the Canadian Arctic Archipelago via Davis Strait and from Hudson Bay via Hudson Strait, together with inputs of melting sea ice, which originate locally or from farther north. The main branch of the LC flows along the Labrador slope from north to south and is centered around the 1000 m depth contour. It is composed of a mixture of Arctic water from Baffin Bay via Davis Strait and the branch of the WGC that flows west across the mouth of Davis Strait. The WGC, which flows from south to north over the

Greenland shelf and along the adjacent slope, is a mixture of cold, low salinity Arctic water exiting the Nordic Seas with the East Greenland Current (EGC) (Yashayaev, 2007), together with sea ice and glacial melt water (Fig 1). The WGC often spreads westwards, forming a "tongue" of buoyant fresher water, with the accumulation of low salinity waters, driven by high eddy kinetic activity in the central eastern Labrador Sea during spring (Frajka-Williams and Rhines, 2010). The WGC often floats over the IC in the central-eastern part of the Labrador Sea, however, the IC is usually observed in surface waters of the central-western Labrador Sea during spring. More detailed descriptions of the hydrography of the Labrador Sea can be found elsewhere (Fragoso et al., Head et al. 2013, Yashayaev and Seidov, Yashayaev 2007)."

RC3.25 - L123: There is no evidence than the cold fresh after originated from Arctic contribute substantially to the deep basin since the front between the basin and the shelf is very strong. Part of the VITALS program using gliders is actually studying the exchange between the basin and the Labrador Shelf (B. De young, J. Palter et al.).

AC3.25 - We have changed this paragraph for clarification. We now refer to the upper Labrador Sea layers (< 200 m) that are comprised of waters originating from the North Atlantic (IC) and the Arctic (LC and WGC). See the response above (AC3.23).

RC3.26 - L134: "Data used in this study"

AC3.26 - Changed.

RC3.27 - L134: remove "from stations" and "repeat".

AC3.27 - Changed.

RC3.28 - L146: Choose between "surface" or "near-surface" and stick to it all along the manuscript.

AC3.28 - We have chosen to refer to surface waters throughout the entire manuscript.

RC3.29 - L155: Maybe add the underline word "Back in the laboratory, POC/PON

samples. . .”

AC3.29 - Changed.

RC3.30 - L171: I think the good way to describe the CHEMTAX output is "relative abundance" instead of "ratios of abundance"

AC3.30 - Changed.

RC3.31 - L173: not clear if all the pigments ratios are from the literature.

AC3.31 - We have added a sentence to the Table legend mentioning that the pigment ratios were extracted from the literature. See the pdf attached in the response letter for reviewer #1, where we have added a new version of this table and legend.

RC3.32 - L174: Please indicate how the algal groups present in the study area are identified.

AC3.32 - The identification is described in full in Fragoso et al. (2016). We have now included this reference in this sentence of the manuscript. See below.

Line 174 – ". . .pigment concentrations of algal groups that are known to be present in the study area as reported in Fragoso et al (2016)".

RC3.33 - L187: remove "that"

AC3.33 - Changed.

RC3.34 - L190: explain here the purpose of the fourth-root transforamation.

AC3.34 - An explanation has now been included. See line the sentence rewritten below. Line 189. – ". . .were standardized and fourth-root transformed before being analysed. Due to the high abundance of diatoms in the data, we have decided to apply a fourth-root transformation to increase the importance of less abundant groups, which would allow us to better discerning the spatial-temporal patterns of the phytoplankton communities in the Labrador Sea."

RC3.35 - L195: "higher" than what? Be careful to compare with something when you use a superlative.

AC3.35 - This word was changed to "high".

RC3.36 - Results L277: "less well stratified"..."at those stations where"

AC3.36 - Changed.

RC3.37 - L278: replace "during" by "in"

AC3.37 - Changed.

RC3.38 - L279: "more highly stratify": pleonasm again...

AC3.38 - We have removed the word "more" from the sentence.

RC3.39 - L281: "higher": then superlative to be compared with something.

AC3.39 - We have removed the parenthesis in this sentence so it is changed to: "...POC:PON ratios were also higher > 8..."

RC3.40 - L288: Not clear if the "pairwise analysis" you mentioned refer to the ANISOM one-way pairwise?

AC3.40 - We have changed the sentence to: "Pairwise one-way analysis of similarity (ANOSIM) between clusters..."

RC3.41 - L289: too long sentence, please reduce or cut in two parts. Parentheses are at the wrong place.

AC3.41 - We have now split the sentence into two. See below.

Line 287 - "Pairwise one-way analysis of similarity (ANOSIM) between clusters suggested that they were significantly different in terms of algal pigment composition (p = 0.001). However, pairwise analysis of clusters C3a and C3b showed that these groups were more similar in composition (R statistic = 0.33) than other clusters (R statistic

values approached 1) (see Clarke and Warwick, 2001)."

RC3.42 - L298: "especially" is useless here. In general, there is an over utilisation of adverbs in the text (mostly/especially. . .).

AC3.42 - The word "especially" was removed from this sentence.

RC3.43 - L313: superlative!! No subject of comparison. . .

AC3.43 - We have rewritten this whole paragraph. See comment below (AC.3.44).
RC3.44 - L315: superlative again. Wrong use.

AC3.44 - We have rewritten this paragraph. See lines the rewritten paragraph below.

Line 311. "In general, chlorophytes and diatoms (cluster C3a) were associated with the inshore branch of the Labrador Current (LC), on the Labrador Shelf. Surface waters from the LC were the coldest (temperature < 2°C) and least saline with the lowest density ($\sigma$Æ§ of most stations approximately < 26.5 kg m-3) of all the surface water masses of the Labrador Sea (Fig. 5a). Mixed assemblages (cluster C3b), as well as blooms (chlorophyll average = 4 mg Chla m-3) of dinoflagellates and diatoms (cluster C2) were associated with the Atlantic water mass, and the Irminger Current (IC) (Fig. 5a). These were the warmest (temperature > 3°C), saltiest (salinity > 34) and densest ($\sigma$Æ§ of most 315 stations < 27 kg m-3) surface waters of the Labrador Sea (Fig. 5a)."

RC3.45 - L321-324: Too long sentence make it confusing. Separate in two sentences?

AC3.45 - This sentence was split into two. See below.

Line 321. "The ordination diagram revealed that stations from each distinct clusters are concentrated in different quadrants (Fig. 5b). The arrows in the ordination diagram represent the environmental variables. Positive or negative correlations indicate that the arrows are orientated parallel to the distribution of cluster stations with the strength of the correlation proportional to the arrow length."

RC3.46 - L340: The table 4 is difficult to understand and could earn a better presentation.

AC3.46 - We have now reorganised Table 4, separating it into Table 4a and Table 4b. See the response to the reviewer #2 (AC2.48), where we add a new version of this table. Further explanation is given in new paragraph of the manuscript and in the revised legend of Table 4 (Line 963).

Line 339. "Table 4a indicates that the first axis (x-axis) of the redundancy analysis explained most of the variance (83.5 % of species-environment relationship; taxa-environmental correlation = 0.68). Summed, the canonical axes explained 99.8 % of the variance (axis 1, p = 0.002; all axes, p = 0.002) (Table 4a), which indicates that the environmental variables included in this analysis explained almost 100 % of the variability. Forward selection showed that five of the six environmental factors (silicate, temperature, salinity, nitrate and phosphate) included in the analysis best explained the variance in phytoplankton community composition when analysed together (p<0.05, Table 4b). When all variables were analysed together (conditional effects, referred to as $\lambda$a in Table 4b), silicate was the most significant explanatory variable ($\lambda$a = 0.2, p = 0.001), followed by temperature ($\lambda$a = 0.05, p = 0.001), salinity ($\lambda$a = 0.02, p = 0.002), nitrate concentration ($\lambda$a = 0.01, p = 0.016) and phosphate concentration ($\lambda$a = 0.02, p = 0.002) (Table 4). Stratification Index (SI) was the only explanatory variable that had no statistical significance in explaining the distribution of phytoplankton communities (Table 4b)." Line 963. "Table 4 – Results of the Redundancy Analyses (RDA) with the eigen-values, taxa-environmental correlations and percentages of variance explained used in the analysis (a). Automatic forward selection (a posteriori analysis) was used to determine the environmental variable(s) that best explain the variance of the data (b). The subset of environmental variable(s) that significantly explained phytoplankton distribution are referred to marginal effects ($\lambda$1) when analysed individually, or conditional effects ($\lambda$a) when analysed additively in the model (b). Explanatory variables are temperature (°C), salinity, nitrate (NO3-; $\mu$mol L-1), phosphate (PO43-; $\mu$mol L-1), silicate (Si(OH)4; $\mu$mol L-1) and Stratification Index (SI) (kg m-4). Significant p-values

(p < 0.05) represents the variables that explain the variation in the analyses." RC3.47 - L345: there is a problem, the title is the same than 3.3!!

AC3.47 - The title has now been updated to "Phytoplankton distributions and elemental stoichiometry".

RC3.48 - L344-352: Please present the POC-PON relationships somewhere.

AC3.48 - We are not sure what the reviewer means by this comment, but POC:PON relationships are shown in Figure 6a and has been referred to in line 351.

RC3.49 - L354: Please quickly explain the purpose of calculating the relationships between POCphyto and POC:PON.

AC3.49 - We have now added a short explanation of the purpose of studying the relationships between POCphyto and POC:PON. See the sentences rewritten below.

Line 354 - To investigate the influence of phytoplankton community structure on the stoichiometry of particulate organic material of surface Labrador Sea waters, the relationships between POCphyto (the estimated proportion of POC from phytoplankton) and the ratio of POC to PON were examined. In general, different phytoplankton communities had distinct relationships between POCphyto and POC:PON. Stations in the shelf regions . . ."

RC3.50 - L359: I would say, ". . .contribute for a high proportion. . ."

AC3.50 - Changed.

RC3.51 - L362: superlative lower (use low or compare to something).

AC3.51 - We have now included an object of comparison in this sentence. See the sentence rewritten below.

Line 362. "Stations influenced by Atlantic waters had generally lower contributions of POCphyto compared to Arctic-related waters, with most stations having POC:PON

ratios < 6.6 (Fig. 6c)."

RC3.52 - Discussion L392: as noted earlier in the manuscript, the surface phytoplankton didn't growth in the Irminger water since this water mass is observed only the slope and at great depth.

AC3.52 - In the central-eastern part of the Labrador Sea, the IC is found below the WGC "tongue", as the reviewer mentioned. However, in the central-western region the IC is found at the surface so phytoplankton are growing in these different water masses (IC, LC, WGC). Phytoplankton species found in the IC are usually found in Atlantic waters, while polar species are found in the LC and WGC (see Fragoso et al 2016).

RC3.53 - L396-397: Here the concept of ecological succession should be better presented. Is the variation between a deep and shallow mixed layer associated to the season or the two conditions (shallow/deep mixed layer) can be observed at the same time of the year?

AC3.53 - Part of this paragraph has been rewritten to clarify the seasonal and temporal patterns of phytoplankton communities. See below.

Line 390 – "In this study, our assessment of phytoplankton pigments from surface waters of the Labrador Sea during spring/early summer are based on a decade of observations and show that the distribution of phytoplankton communities varied primarily with distinct waters masses (Labrador, Irminger and Greenland Currents). However, a temporal succession of phytoplankton communities from the central region of the Labrador Sea was observed as waters became thermally stratified from May to June. Major blooms (Chla concentrations > 3 mg Chla m-3) occurred on or near the shelves in shallower mixed layers (< 33 m, Table 5). Diatoms were abundant in these blooms, however they co-dominated with 1) chlorophytes in the west (mostly in the Labrador Current) and 2) Phaeocystis in the east in the West Greenland Current. A more diverse community with low chlorophyll values (average Chla concentrations ∼2 mg Chla m-3,

[Figure]

Table 5) was found earlier in the season (May) in deeper mixed layers (> 59 m, Table 5) of the central basin. Once these waters of the central basin became thermally-stratified (June), a third bloom co-dominated by diatoms and dinoflagellates occurred, revealing an ecological succession from mixed flagellate communities. These patterns are similar to those seen in other shelf and basin regions of Arctic/subarctic waters (Coupel et al., 2015; Fujiwara et al., 2014; Hill et al., 2005)."

RC3.54 - L401-403: A link is missing between this information and the above sentence.

AC3.54 - This sentence has been rewritten for clarification. See lines this paragraph rewritten below.

Line 398. "It is well known that diatoms tend to dominate in high-nutrient regions of the ocean due to their high growth rates, while their low surface area to volume ratios mean that they do not do as well as smaller nano- or picoplankton in low nutrient conditions (Gregg et al., 400 2003; Sarthou et al., 2005). The Labrador Sea is a high-nutrient region during early spring due to the deep winter mixing (200 – 2300 m) that provides nutrients to the surface layers. High nutrient concentration supports phytoplankton spring blooms, particularly those dominated by diatoms, once light becomes available (Fragoso et al., 2016; Harrison et al., 2013; Yashayaev and Loder, 2009)."

RC3.55 L406: "often" and "as well" mean the same here. Please remove one of the two.

AC3.55 -We have changed the word "often" to "occasionally" to clarify the sentence.

RC3.56 L470: I would prefer to use the mean POCphyto rather than POC>...The latter formu- lation is not really comparable since we don't know the dispersion of the data.

AC3.56 -The dispersion of the data of POCphyto/total POC and POC:PON ratios are shown in Figure 6c and 6b, respectively. We have now referred to the figure in the text. See line the sentence rewritten below.

Line 469. "In this study, highly productive surface waters of Arctic origin (near or over

the shelves) had higher phytoplankton-derived particulate organic carbon (POCphyto > 43 % of total POC, Fig. 6c), as well as higher and more variable POC:PON ratios (average > 6.9, Fig. 6b) compared with stations influenced by Atlantic water (average POC:PON < 6.3, POCphyto > 35 %, Fig. 6b)."

RC3.57 - L475: were also abundant

AC3.57 - We are not sure what the reviewer is referring to here.

RC3.58 - L512-519: It should be interesting to explain the meaning of the AP/TChla ratio in term of strategy for the adaptation to light regime.

AC3.58 - Few studies have examined this in any depth and hence we can conclude very little in the present study. AP/TChla ratio varied according to community composition and species adaptation to light environments, mixing regimes, competition for light with other dissolved substances (etc) could explain the observed trend. Further in depth physiological work is needed. We have extended the discussion a little bit in the paper in attempt to explain why such trend is observed.

RC3.59 - L522-523: Conufsing because you introduce "two parameters" and after you cite three parameters (Nitrate, Silicate and SI).

AC3.59 - The word "two" has been removed from this sentence.

RC3.60 - L540-552: You show interesting difference in the photophysiological charac-teristics of phytoplankton, especially between the west and east communities. Near Greenland, the communities is composed of species resistant to high light while on the Labrador Shelf, the species are less resistant to photo-inhibition. Is the light conditions are so different between east and west to explain these different adaptations to light? It could be interesting to describe these difference in the light regimes between the two side of the Labrador Sea. The latter melting of the ice cover on the Labrador Shelf could be an explanation?

AC3.60 - We have now improved the discussion about the influence of PAR in separating the polar phytoplankton communities observed. See the rewritten paragraphs below.

Line 540 – "Polar phytoplankton communities from shelf waters (east versus west) observed in this study had distinctive photo-physiological characteristics. Comparing these blooms, diatom/chlorophyte communities (west) had higher photosynthetic efficiency ($\alpha$B = 9.2 $\times$ 10-2 $\mu$g C $\mu$g Chla h-1 W m-2), lower light-saturation irradiance (Es = 35 W m-2) and higher photo-inhibition ($\beta$ = 16 $\times$ 10-4 $\mu$g C $\mu$g Chla h-1 W m-2) than communities from the east. This suggests that the community located in the Labrador Shelf waters (west) was more light-stressed compared to the community observed in the east (diatom/Phaeocystis). Haline-stratification due to the influence of Arctic waters occur in both regions during spring, contributing to the shallow mixed layer depth (<33 m) observed (Table 5). However, waters from the Labrador Shelf (west, Cluster C3a) were more stratified than the Greenland Shelf (cluster B, see stratification index (SI) values, Table 5) because of the local sea ice melt observed in this area, which contributes to increased stratification in this region. The diatom species observed on the Labrador Shelf were mostly sea-ice related (Fragilariopsis cylindrus, Fossula arctica, Nitzschia frigida) compared to pelagic species observed in the Greenland Shelf waters (Thalassiosira gravida, for example) (Fragoso et al., 2016). Sensitivity of sea-ice related diatoms to irradiance > 15 $\mu$mol photons m$-2$ s$-1$ has been reported (Alou-Font et al., 2016), which can help explaining why phytoplankton communities from the west were photo-inhibited. Phaeocystis/diatoms located near Greenland (east) had the inverse pattern: low photosynthetic efficiency (average $\alpha$B = 6.8 $\times$ 10-2 $\mu$g C $\mu$g Chla h-1 W m-2) and high light-saturation irradiances (Es = 62 W m-2). This pattern in diatom/Phaeocystis dominated communities mean that photosynthetic rates were relatively low at high light intensities, although photo-inhibition was low ($\beta$ = 4 $\times$ 10-4 $\mu$g C $\mu$g Chla h-1 W m-2). Phaeocystis antarctica, widespread in Antarctic waters, relies heavily on photo-damage recovery, such as D1 protein repair (Kropuenske et al., 2009), which could explain how these communities overcome photo-inhibition. Stuart et al. (2000), however, found a high photosynthetic efficiency ($\alpha$B) for a population

dominated by Phaeocystis near Greenland and attributed this to the small cell size of Phaeocystis. However, in addition to the exposure of ice-related diatoms to high light levels due to increased stratification, the high concentration of chlorophytes and prasinophytes, which are also small in cell size, might also explain the higher $\alpha B$ observed in the Labrador Shelf waters (west, cluster C3a) when compared to values from diatom/Phaeocystis blooms (east, cluster B).

RC3.61 - L555 to 558: The sentence is confusing. It takes time for me to understand that dinoflagellates bloom in May to avoid higher light levels. Please rephrase or separate in two sentences to improve the clarity. AC3.61 - The beginning of this paragraph has been rewritten for better elucidation. See the sentence below.

Line 555 - "Days are longer and solar incidence is higher in June compared to May at these latitudes (Harrison et al., 2013). Dinoflagellates were found to bloom in the central Labrador Sea in June as a consequence of increased thermal stratification. To cope with high light levels and potential photo-damage, this phytoplankton community appeared. . ."

Hauser, T., Demirov, E., Zhu, J., Yashayaev, I., 2015. North Atlantic atmospheric and ocean inter-annual variability over the past fifty years – Dominant patterns and decadal shifts. Prog. Oceanogr. 132, 197–219. doi:10.1016/j.pocean.2014.10.008 Yashayaev, I., 2007. Hydrographic changes in the Labrador Sea, 1960-2005. Prog. Oceanogr. 73, 242–276. doi:10.1016/j.pocean.2007.04.015 Yashayaev, I., Seidov, D., 2015. The role of the Atlantic Water in multidecadal ocean variability in the Nordic and Barents Seas. Prog. Oceanogr. 132, 68–127. doi:10.1016/j.pocean.2014.11.009
* * *

---

## Author Response (AR1)

**UNIVERSITY OF Southampton**

**24**th November, 2016 Editor, **Biogeosciences**

Dear Editor:

I, with my co-authors, would like to resubmit our article entitled "Spring phytoplankton communities of the Labrador Sea (2005-2014): pigment signatures, photophysiology and elemental ratios" for publication in **Biogeosciences**. The paper has been extensively revised in accordance with the suggestions of the reviewers, and a sheet detailing the changes made is included in this letter. A revised manuscript (with and without track changes) and new supplemental material are attached.

Thank you for your consideration, and we hope the paper is now acceptable to you.

Sincerely,

Glaucia Fragoso, PhD
Ocean and Earth Science, National Oceanography Centre Southampton
Email: glaucia.fragoso@noc.soton.ac.uk

Title: Spring phytoplankton communities of the Labrador Sea (2005-2014): pigment signatures, photophysiology and elemental ratios. Resubmitted to Biogeosciences.

**We thank the three reviewers for their comments and suggestions, which we feel have greatly improved the manuscript. Below we respond to each comment in detail. RC refers to "Reviewer's Comments" and AC to "Author's comments". We have enumerated the reviewer's comments to organise better our responses.**

Reviewer #1:

RC1.1- General comments: This work provides information on the phytoplankton groups found in the surface waters of the Labrador Sea. Pigment signatures determined with HPLC were analyzed with CHEMTAX to obtain the contribution of the various algal groups to the total chlorophyll a concentration. The authors also related the phytoplankton biogeographic distribution to the properties of the various water masses and the photophysiology of cells during the late spring /early summer over a 10 year period. The use of CHEMTAX for this data set is a novel application, however, a previous publication by Fragoso et al. in 2016 described the phytoplankton communities linked to the various hydrographical areas of the Labrador Sea at depths less than 50 meters using microscopy. Although both microscopy and CHEMTAX analytical methods are critical to any biogeographic examination of phytoplankton, I feel the two methods should have been combined into a single manuscript as they complement one another.

**AC1.1 - The first manuscript (Fragoso et al 2016, Progress in Oceanography) only focused on taxonomy of large phytoplankton (>4 µm), identifiable by light microscopy. Moreover, it only included data from 4 years (2011- 2014), and not all the stations sampled along the AR7W transect line. The current manuscript focuses on additional algal groups not considered in Fragoso et al (2016) through analysis of pigments. The new manuscript also covers a much larger dataset (10 years of data rather than only 4), many more stations from the AR7W line, and includes important biogeochemical aspects of the data not examined before. For these reasons we have kept the two papers separate and distinct, in order to ensure all aspects of phytoplankton (taxonomy, algal groups, biogeochemical and physiological aspects) are examined in full. This is now explained in line 71 and 628.**

RC1.2 Therefore, although I consider this work to be of value in its contribution to our understanding of the dynamics of the biogeochemical characteristics of the Labrador Sea, I feel its content fails to merit publication in present form.

Key problems that I feel need to be addressed include: 1) the absence of the initial
CHEMTAX matrices and RMS errors

**AC1.2 - CHEMTAX input matrices have now been inserted in the manuscript (Table 3). The output matrices and information about the range of RMS errors (in the legend of the Table S1) has now in the supplemental material.**

RC1.3 2) the organization of the methodology section; it is not well structured, it includes CHEMTAX results and lacks information (see specific comments)

**AC1.3 - The method section has now been modified according to the reviewer's suggestions. For specific changes, see the response to specific comments (methods, comments RC1.11 onwards) below.**

RC1.4 - 3) the amount of information presented regarding taxonomy; species-specific information for the encountered groups of diatoms would have helped to understand differences on the photoprotective responses observed.

**AC1.4 - We agree with the reviewer comment and have now added additional comments identifying the distinct diatom species influencing differences in the photoprotective response. See the line 564 in the revised manuscript.**

RC1.5 - 4) of all the identified pigments (presented in Table 2) only the (DD+DT): chl a and DT:(DT+DD) are included for discussion on cell physiology. The authors should at least have included why they did not use the PPC:PSC, PPC:chl a or the pigment chlorophyllide a

**AC1.5 - Although we appreciate the reviewer's suggestion, the ratios of PPC:PSC in this study varied according to phytoplankton community structure due to the inherent variations of PPC and PSC within different phytoplankton groups. For this reason, we decided to not include PPC:PSC ratios. We focused on AP:TChla ratios as a function of phytoplankton community structure. This relationship not been explored in detail previously. In contrast, the DD+DT:chla ratio only pertains to taxa which possess a xanthophyll cycle with a photoprotective (DD) and photosynthetic (DT) component, hence we have retained our focus on the DD and DT patterns.**

RC1.6 - 6) the use of accepted and standardized abbreviations for the marker pigments and the phytoplankton groups in Tables 2 and 3 and throughout the text and finally

**AC1.6 – We apologise for this oversight and have now updated and added the commonly accepted and standardized abbreviations for the marker pigments as suggested by the reviewer. See Tables 2 and 3 in the revised manuscript.**

RC1.7 - 7) the correction of any incorrectly assigned references.

**AC1.7 - References have now all been corrected.**

RC1.8 - Specific comments
Introduction Line 54: change for Phaeocystis spp. colonies (> 100µm).

**AC1.8 - This sentence has been removed as the introduction was rewritten following another reviewers comments.**

RC1.9 Line 83: update references.

**AC1.9 - This sentence has been removed.**

RC1.10 - Line 84 what do you mean by "while the influence of phytoplankton composition on photophysiological patterns has not been investigated thoroughly?" please explain further.

**AC 1.10 - This sentence has been removed.**

RC1.11 - Methods In general this section is not well structured and needs clarification and more detail.

Sampling and analysis are combined throughout this section and need to be presented with more organization. I recommend organizing this section into separate Study Area, Sampling and Biogeochemical Analyses sub-sections and limiting relevant data to relevant sub-sections.

**AC 1.11 An additional subsection named "Biogeochemical analysis" has been added including nutrient, POC:PON and chlorophyll *a* methodology.**

RC1.12 - Line 138: please include the number of stations sampled before fixed stations (was it 28 as in the previous work?). The number of depths sampled at each station should appear in the text as well.

**AC1.12 - Information about the number of stations has now been added to the manuscript (line 131). Samples for this manuscript were collected from the surface only and this information is now included in line 140.**

RC1.13 - Line 141: please write the specifications of the Seabird CTD system.

**AC 1.13 - Specifications of the Seabird CTD system (SBE 911) has been inserted (line 134).**

RC1.14 - Lines 148-149: the description of how the total chl a was analyzed is presented before explaining how the collected samples for pigment analyses were filtered (probably on board?). Were samples for chl a fluorometric determination kept frozen at -20C until analyses or at -80C is a bit confusing. Was the extraction (90% acetone) performed by keeping the filters at -20C for 24 h? or rather the filters were kept at -20C until analysis (extraction for 24h with 90% acetone)? Was acidification of the samples performed?

**AC 1.14 - We have now explained how chlorophyll a was sampled and analysed (see lines 143 and 153) and have cited the methodology of Holm-Hansen et al (1965) for chlorophyll determination by fluorescence, which includes acidification of the samples.**

RC1.15 - Line 151: I recommend changing this line to "samples for detailed pigment analysis were filtered onto 25 mm Whatman GF/F filters".

**AC 1.15 - We rewrote this sentence as: "…filtered onto 25 mm glass fibre filters (GF/F Whatman Inc., Clifton, New Jersey)" (see line 143).**

RC1.16 - Lines 151-153 How much time passed between storage and analysis for the samples? Were the samples always analyzed in the same laboratory for every cruise over the 10-year period? Information on the maximum time of filtration is not provided and is very important for xanthophyll measurements. If too much passed while doing the filtration, the measurements of diatoxanthin are likely to be meaningless. This is also important for degradation pigment information, however the later data are not presented.

**AC 1.16 – This information is now inserted in lines 143: "Samples for detailed pigment analysis were filtered onto 25 mm glass fibre filters (GF/F Whatman Inc., Clifton, New Jersey) and immediately flash frozen in liquid nitrogen, kept frozen in a freezer (at -80° C) until analysis in the BIO (2005-2013) or NOC (2014) laboratories within 2-3 months of collection. Volumes of water sampled for HPLC analysis were adjusted, such that samples were filtered as quickly as possible (< 10 mins)."**

RC1.17 - Line 153: were the nutrient samples kept frozen or refrigerated until analysis?

**AC 1.17 – Refrigerated (5°C and analyse at sea within 12 h of collection). We have now added this information in the text (line 147).**

RC1.18 - Pigment analysis Line 166: Was calibration done with external pigment standards obtained from DHI? Was the precision of the instrument tested? Is there a variation coefficient? Do you have limits of detection? Please at least provide the limits of detection and quantification and how were they estimated and if the pigments with concentrations below this limit were reported or not. All this information is relevant and missing.

**AC 1.18 - Information about the standards, calibration and quantification procedures are described in detail in Stuart and Head (2005) and Poulton et al (2006), which we have now cited in this section. We have also added information about precision, coefficient of variation and limits of detection (lines 158 - 165).**

RC1.19 - Table 2: In this table and throughout the manuscript the authors should follow the abbreviations for phytoplankton pigments and pigment formulae suggested in the Scientific Council for Oceanic Research (SCOR), Jeffrey et al. 1997 or in Higgins et al. 2011 In: Roy S, Llewellyn CA, Egeland ES, Johnsen G (eds) Phyto- plankton pigments: char-acterization, chemotaxonomy and applications in oceanography. Cambridge University
Press, Cambridge, p 257

**AC 1.19 - Pigment abbreviations in Table 2 were updated as suggested by the reviewer.**

RC1.20 - This table should summarize the distribution of major taxonomically significant pigments found in the various algal groups during the study. This is poorly done in its current form. The authors should avoid ambiguity. For example when referring to 19'-hexanoyloxyfucocanthin (Hex-fuco), it should be mentioned that is a major pigment in haptophytes and dinoflagellates (Type-2, lacking peridinin), instead of "some dinoflagellates" or "various". This information –if provided here- would improve significantly the reading of the few next sections dealing with the marker pigments used for the CHEMTAX analysis. Only if the authors are more specific, the use of the references Jeffrey et al. 1997 or Higgins et al. 2011 make sense. Please delete the reference column of this table unless is useful (not the case in its present form).

**AC 1.20 - This table has been updated. We have included the more specific information requested following Jeffrey et al (1997), Higgins et al (2011) and Vidussi et al (2004).**

RC1.21 - Chlorophyll c1 + c2 should stay as Chlorophyll c1 + c2. Please avoid the use of CHLC12.

**AC 1.21 - Abbreviations updated.**

RC1.22- Zeaxanthin is a minor pigment present in various groups as cyanobacteria, however this group is supposed to be practically absent in polar waters. Although Blais et al. 2012 showed that cyanobacteria may be underestimated in polar regions (Beaufort Sea & Baffin Bay). Did the authors find presence of cyanobacteria using epifluorescence microscopy?

**AC 1.22 - We did not count cyanobacteria but referred to information confirming the presence of *Synechococcus* in the Labrador Sea (Atlantic waters) from Li et al (2016) as stated in line 215.**

RC1.23 - Also did the authors perform any correlation analyses between prasinoxanthin and zeaxanthin to prove that the zeaxanthin encountered did or did not correspond to a group of prasinophytes-containing zeaxanthin? Please provide this information.

**AC 1.23 - We are not sure if we understand this point raised by the reviewer as a correlation between prasinoxanthin and zeaxanthin would not directly determine whether the zeaxanthin found belongs to prasinophyte-containing zeaxanthin or cyanobacteria. Zeaxanthin, in this study, represented not only prasinophytes type 2, but also chlorophytes and cyanobacteria. Moreover, species representing prasinophyte type 2, such as *Pyramimonas* and *M. pusilla* have been observed (qualitatively in our samples, although not directly counted due to difficulties in quantification) in the Labrador Sea from microscope observation of Lugols fixed samples. *M. pusilla* is abundant in the North Water Polynya in regions near the Labrador Sea as stated in line 213.**

RC1.24 - Pigment interpretation There are major problems with this section. The title itself is more like the title of a results section. Actually the authors use the title "CHEMTAX interpretation" as a section included in the results. I suggest the authors change the title of the pigment interpretation section to HPLC pigment data or Clustering of HPLC data for CHEMTAX or CHEMTAX analysis or something similar-

**AC 1.24 - The title of the section has been changed to "CHEMTAX analysis" as suggested (line 167).**

RC1.25 - This section is not well structured and difficult to follow partially because the authors explain the use of the selected initial pigment ratios while presenting the output matrices after the CHEMTAX analyses (Table 3). This is confusing for the reader. The initial ratio matrices used to seed CHEMTAX are not presented or explained with detail. Instead ambiguous information is presented e.g. "diatoms were identified as containing high fucoxanthin to chl a ratios"

**AC 1.25 - Initial pigment ratios (Table 3) have now been inserted in the new version of the manuscript and the output ratio information has been moved to the supplemental material. Explanation for the selected ratios are explained in lines 179 – 189 and we have now included a column in the initial pigment ratios table (Table 3) mentioning the source reference (\*Ref) where the ratios were taken from to seed the initial CHEMTAX analysis.**

RC1.26 - Line 171: change it for Mackey et al., 1996, version 1.95.

**AC 1.26 – Changed.**

RC1.27 - The following paragraph is not straightforward. The information on how CHEMTAX works in general and how version 1.95 works lacks clarity. This later version is a significant improvement on CHEMTAX application since the software sets up the multiple (60) initial pigment ratio matrices to obtain the more stable final values (as was recommended for example by Latasa 2007) and was actually used and described by Wright et al. 2009 and other authors before Coupel et al. 2015! Please add the references.

**AC 1.27 - This paragraph has been rewritten (lines 168 - 177) and the earlier references have now been included.**

RC1.28 - Line 179 to the end of the paragraph: please use the standardized abbreviations and you should at least explain why you decided to choose these particular marker pigments for the CHEMTAX analysis. Your microscopy results from the previous work should help here in a more detailed way.

**AC 1.28 – We have now reorganised this paragraph. The explanation for the selection of the pigments are found in lines 182 – 189.**

RC1.29 - Line 183: Again, please refer to Mackey et al. 1996 before more recent studies.

**AC 1.29 - Reference now added (line 192).**

RC1.30 - Line 191 to 197: Is figure 2 referring to the mean relative concentration of the main marker pigments to total accessory pigments (wt:wt) encountered or to chl a or total chl a or is based on the pigments absolute values? Unclear. It would have been helpful to include in this figure the biogeographical region linked with each cluster (as in figure 3).

**AC 1.30 – Figure 2 has been moved to the supplemental material (now Fig S1a) and refers to the percentage contribution of each diagnostic pigment to the observed statistical Bray-Curtis similarity between samples (at the 60% level) after fourth root transformation (see explanation of the statistics in lines 197 - 201). Thus, it is the mean relative (%) fourth-root transformed concentration of each of the selected pigments to the total (selected) for each cluster (see**

**revised figure legend in supplemental material). We have now added to Figure S1 a biogeographical plot showing the cluster groups as depicted in Figure 3 (Fig. S1b).**

RC1.31 - Line 198: you already explained this earlier (lines 173-74). I think this is not very well explained and this may be the reason why you mentioned it again here. Line 199-200:
"To satisfy this requirement, initial pigment ratios were carefully selected and applied to each cluster". This should actually be mentioned earlier in this section when you explain and justify why you use the selected pigment markers that best describe the phytoplankton community of your study area.

**AC 1.31 - We have reorganised these sentences into the suggested order of explanation from the reviewer.**

RC1.32- Line 204: The authors should justify why they have used the "high light" field ratios from
Higgins et al. 2011. Moreover, considering the importance on the photo-physiological results obtained in this study why is there not more information beside the irradiance of the experimental incubations? Was the PAR incident irradiance measured at the sampling sites?

**AC 1.32 – "High light" field ratios were chosen because the samples were from surface waters (see explanation now in line 185). PAR incident irradiance was not measured at all sampling sites.**

RC1.33 - Line 205: "Prasinophytes were separated into type 1 (containing prasinoxanthin) and type 2 (lacking prasinoxanthin)". Both genera were observed in light microscope counts
(Fragoso et al. 2016)" What do you mean? Fragoso et al. 2016 enumerated pico- phytoplankton (M. pusilla < 2 um)?

**AC 1.33 - We apologise for the confusion. *Pyraminomas* and *M. pusilla* were observed qualitatively from microscope observations but not enumerated by Fragoso et al (2016). We have now changed the reference to (Fragoso, pers obs) in the text to avoid this confusion (line 213).**

RC1.34 - Line 209: Did the authors detect by HPLC the unknown carotenoid that characterizes the unique pigment signature of *M. pusilla*? Did they detect the pigment micromonal in their samples? or micromonol?

**AC 1.34 - The pigment micromonal was not identified as part of the HPLC analytical protocol followed (i.e. it was not a pigment peak listed for identification in the analysis).**

RC1.35 - Line 211: "In addition to prasinophytes –type 2 (type 2A in Higgins et al. 2011- I assume), zea is also the major accessory pigment of cyanobacteria etc.. unclear para- graph.

**AC 1.35 - The beginning of this sentence has been rewritten for clarification (line 214).**

RC1.36 - Line 215: "Prasinophytes (type-1, Higgins et al. 2011) indeed contain chl b so do chlorophytes and they can be distinguished by their relative ratios of lutein to chl b
(Higgins et al. 2011). Was lutein detected with the HPLC analyses? Again correlations would have helped here.

**AC 1.36 - The BIO method does not separate lutein and zeaxanthin so we have now renamed it as Zea + Lut.**

RC1.37 - Line 218: I suggest the authors change Dino-2 class for Dino2 (dinoflagellates type-2).
Avoid the use of class, use what is suggested by Higgins et al. 2011. As mentioned before, this could have been nicely done in Table 2.

**AC 1.37 - We have now rewritten this sentence using "… "dinoflagellate type 2" (Dino-2)" as was suggested by Higgins et al (2011) (line 221).**

RC1.38 - Line 220: Why did the authors use the term Cryptophycea instead of cryptophytes?

**AC 1.38 – We have now rephrased as cryptophytes rather than cryptophycea.**

RC1.39 - Line 256: Please refer to algal groups or phytoplankton groups based on pigment composition instead of "class".

**AC 1.39 - "Phytoplankton/algal class" has been changed to "phytoplankton/algal groups".**

RC1.40 - Results Line 294-296: Where is cluster C1 mentioned in this section to explain Figure 4?

**AC 1.40 - Cluster C1 has now been included in this sentence (line 309).**

RC1.41 - Line 380: Why do you present saturation irradiances here as Wm-2 when in the methodology (line 237) you mentioned the 30 different irradiance levels is expressed as µmol quanta m-2s-1. Please use same units everywhere.

**AC 1.4 - Irradiance units used throughout have been changed to "Wm$^{-2}$".**

RC1.42 - Line 382: What was the % contribution of DD, DT and,-carotene to the total PPC for clusters C3b and C2?

**AC 1.42 - The percentage contribution of DD, DT and,-carotene to the total PPC would vary according to the total amount of PPC (similar situation as comparing to DD+DT/Chl a, see comments below). Moreover, as mentioned previously, we feel that this information is difficult to interpret in a simple photo-physiological sense due to the influence of phytoplankton community structure in the overall PPC values.**

RC1.43 - Line 381: DD+DT/Chl a; clusters C3b and C2 have also the lowest chl a concentration. However the level of deepoxidation is higher for these two cluster. How do your DDDT/chla and PPC/PSC ratios compare with other studies for the Arctic during spring/summer transition? Actually you don't present PPC/PSC, why?

**AC 1.43 - Ratios of PPC:PSC would vary according to variability in phytoplankton community structure and not just cell photophysiology. For example, a community dominated by diatoms would have high PSC:PPC, while a community dominated by prasinophytes would have low PSC:PPC. Figure 7b shows that variability in pigment ratios is mainly driven by community structure. Thus, we believe that this information is insufficient and potentially misleading to discuss the photophysiology in mixed phytoplankton communities. However, as the reviewer pointed out, the level of de-epoxidation was high, which suggests that these communities were exposed to high light levels. We have now changed the sentence from line 619 - 623, and cited other studies (i.e., Alou-Font et al 2016) that show similar patterns.**

RC1.44 - Legend of figure 3: would be better if each variable and parameter is related to the corresponding panel.

**AC 1.44 - We have now revised the figure legend so that each variable and parameter is related to the corresponding panel (line 996).**

RC1.45 - Discussion Very little information is discussed about spatial and temporal incident PAR irradiance variation.

**AC 1.45 - Unfortunately we do not have PAR measurements from the ship for each site during the 10 years of cruise observations. However, we do discuss PAR indirectly through mixed layer depth, stratification index and progression of solar incidence from May to June throughout the discussion section.**

RC1.46 - Line 405: Chlorophytes have also been associated with land-fast ice in the Arctic (e.g. Palmer et al. 2011).

**AC 1.46 - The suggested reference has now been added (line 450).**

RC1.47 - Lines 524-529: I think this is a very interesting result and an interesting point for discussion. Here is where species identification for the diatom groups of Arctic and Atlantic waters would have been helpful. How do these results compare to other Arctic studies?

**AC 1.47 - We have now added a discussion of the influence of distinct species in the variable AP:Tchla ratios (see comments above) (line 561 - 570).**

RC1.48 - Lines 540 to 550: This paragraph deserves a better explanation with at least details on the microscopic most abundant genera for diatoms.

**AC 1.48 - The distinct species of diatoms potentially influencing the AP:TChla have now been included (line 561 - 570).**

RC1.49 - Lines 564 to 575: is more a repeated line of the introduction.

**AC 1.49 - This whole paragraph has been removed.**

RC1.50 - Lines 564 to the end: The resulting ratios of the final CHEMTAX analysis should have been discussed here, at least accordance/discrepancies with past studies in the polar environment. The interesting comparison among the carbon biomass-estimated from
CHEMTAX and the estimated by microscopic observations- should have been better structured and compared with other studies.

**AC 1.50 - We have now included more references that compare the two methods of biomass estimations (CHEMTAX and microscopy) from polar environments (lines 637 – 654) to further clarify these points.**

RC1.51 - Lines 987 to 993: please relate each variable to the corresponding panel.

**AC 1.51 - We have now related each variable and parameter with the corresponding panel (line 996).**

RC1.52 - References need further formatting review.
Latasa M (2007) Improving estimations of phytoplankton class abundances using CHEMTAX. Mar Ecol Prog Ser 329:13

Wright SW, Ishikawa A, Marchant HJ, Davidson AT, van den Enden RL, Nash G (2009)
Composition and significance of picophytoplankton in Antarctic waters. Polar Biol 32:797

**AC 1.52 - References added.**

RC2.1 - GENERAL COMMENTS:

This paper provides a decadal assessment of phytoplankton communities of the Labrador Sea using pigment markers and CHEMTAX analysis, as well as environmental parameters (T, S, nutrients, MLD, etc) and photosynthetic parameters. A single transect was sampled during each late spring – early summer for 10 years with high geographic resolution. The comprehensive suite of measurements makes this a valuable data set that should provide a useful reference for future cruises. I believe it is appropriate for Biogeosciences.

The analyses appear to have been competently performed and I have no worries about the data. Although the text itself is generally well written, at the broader level the manuscript itself unfortunately has two serious problems. First, it is not well structured – in particular, it lacks a clear Aim.

**AC2.1 – We thank the reviewer for his comments have now rewritten large sections of the manuscript. The introduction now has a clear aim of the work identified and this has now been reinforced in the last paragraph.**

RC2.2 -Secondly, and perhaps as a consequence, the authors have attempted to cover too much data in a single publication. They describe the entire data set rather than derive a clear story from it. As a result, key parts of the story are insufficiently described despite a huge volume of complex text, and the overall story is confusing. Three subplots are introduced (Accessory pigment:Chl_a ratios, POC:PON ratios, and photosynthetic parameters) that add little to (what I consider to be) the main story but add considerable verbiage and unnecessary confusion. There is possibly sufficient data here for a thesis, in which each of these subplots would warrant a separate chapter. Here they would be better relegated to separate publications, possibly followed by a review paper that integrates this study with previous work in the region. Due to lack of a coherent focus, the data and discussion are not well integrated.

**AC2.2 - We have now reaffirmed the aims of the paper, which is to compare the biogeochemical and photo-physiological properties of phytoplankton communities from contrasting biogeographical regions in the Labrador Sea and to create a baseline of these trends which could be compared with in the future. These aims require that we comprehensively cover the various aspects ("sub-plots") of the data, and warrants that they are considered equally within the main theme of the paper: the analysis of the phytoplankton community composition from pigment analysis. Although we agree with the reviewer that this manuscript covers a lot of information, we feel that the results from sections 3.4 and 3.5 are directly linked to the information on phytoplankton groups and that writing a different paper about POC:PON ratios and photochemical aspects on their own would lead to them being watered down in importance or relevance to one another. We hope that the reviewer can appreciate our focus and that the restated aim and revised text address his concerns.**

RC2.3 -Despite these problems, this is a very useful study that should be published, but the manuscript requires substantial revision.

STRUCTURAL COMMENTS:
Introduction:
This paper desperately needs a clear Aim to provide a basis for a narrative, to dictate what is included in (or excluded from) the paper, to provide a focus for the Results,
Discussion and Conclusions, and by which to judge the success of the project.

**AC2.3 - Clear aims have now been added in the last paragraph of the introduction and the Results, Discussion and Conclusions all link to these aims.**

RC2.4 -There is an implicit aim in the sampling regime – "What are the major determinants of phytoplankton composition and abundance in the Labrador Sea?" My comments hereafter will address this aim, and I leave the authors to judge how appropriate they are to the revised paper.

**AC2.4 - The major determinants of phytoplankton composition and abundance have already been investigated in detail by Fragoso et al (2016). The uniqueness of the current manuscript is that it takes the subject matter a step further to focus on additional algal groups, including those not included in Fragoso et al (2016). Moreover, we investigate the biogeochemical (POC:PON) and photo-physiological signatures of these different communities and show that these all vary across the biogeochemical domains of the Labrador Sea. The current manuscript includes a much larger dataset (10 years) and more stations from the AR7W line than Fragoso et al. (2016). Following this reasoning we have decided not to follow the direct suggestions of the reviewer to focus only on hydrography, but rather we continue to examine other aspects (biogeochemistry and physiology) of the phytoplankton communities. We hope that this provides a more holistic understanding of phytoplankton dynamics in the Labrador Sea.**

RC2.5 -The Introduction must provide sufficient information to provide the context for the Aim and to allow the reader to understand the significance of the results as they are presented. It must introduce all of the major topics covered in the paper, but nothing else.
Thus, the first two paragraphs (lines 42-65) are unnecessary; as is the paragraph on
CHEMTAX starting line 86 (which should be replaced by a brief outline on the approach taken to address the Aim).

**AC2.5 – We retained the first two paragraphs as they refer to the impact of hydrography on community structure and photophysiology and to the impact of phytoplankton community structure on C:N ratios. However, we do agree with the reviewer that the introduction needs to better guide the reader and provide enough information for the stated aims of the paper. Thus, we have now radically shortened the whole introduction and reorganised it in attempt to make it clearer for the reader.**

RC2.6 -The description of the study region is currently split between the Introduction (lines 66-
84), Methods (lines 114-132), and Discussion (lines 409 – 413). Given that the notional paper is now about the Labrador Sea, I suggest that all of this information should be amalgamated in the Intro, as should most of the description of the NAO (lines 425-430), and Figure 1.

**AC2.6 – We believe that merging this information would result in a very long introduction and have decided to give only a brief overview of the regions of the Labrador Sea in the introduction (see paragraph 3), while we focus on the complex hydrography in the study area section (2.1) of the methods. The possible effects of the NAO are not investigated in depth in this manuscript and therefore we have retained it only in the discussion.**

RC2.7 -I would specifically identify the main factors that may control phytoplankton – temp, salinity, mixed layer depth, light, nutrients, ice, meltwater. I also think that the Introduction should mention that the cruises occurred at different times of the Spring/Summer, introducing the notion of a temporal sequence, as this was the basis for one of the Conclusions (which surprised me on the first read!). Also that there were some cruises that deviated from the normal transect. I note that there was another publication by the same authors in the same region this year. I am surprised that there was not a specific reference to how this study relates to the previous one.

**AC2.7 – The main factors that control phytoplankton bloom is now mentioned in line 65. Information about the seasonality of late spring/early phytoplankton communities is also inserted in the introduction (see third paragraph of the introduction), including findings from the previous study by Fragoso et al (2016) (line 65 - 71). We have also made it clear to the reader that there is (slight) temporal variability in the sampling times (line 128, Fig. 2b, Table 1). Line 420 – 431 also now includes a summary of the temporal progression of communities observed in this study.**

RC2.8 -Method

The inclusion of results in section 2.4 surprised me at first, but I think that this section is peripheral to the main story and is appropriate here.

**AC2.8 - We agree with the reviewer.**

RC2.9 -Results:
I was frustrated by the fact that CHEMTAX results were presented only at the community level as defined through cluster analysis – but what was happening with the individual taxa that comprised these communities? Later I discovered that these results were (sort of) presented in the Discussion. I suggest that the distributions of individual taxa should be presented (with figures) before the distributions of communities.

**AC2.9 - Information about individual taxa has now been inserted in the results section (lines 291 - 299, see also Fig. 3)**

RC2.10 - I would like to see a more detailed analysis of the factors controlling phytoplankton in each water mass. Even though there was considerable data on photosynthetic properties, I didn't get a clear message on the role of light in controlling biomass.

**AC2.10 – Unfortunately we do not have PAR measurements for each site during the 10 years of cruise observations. However, we do discuss PAR indirectly through mixed layer depth, stratification index and progression of solar incidence from May to June throughout the discussion section. Further analysis of nutrient and light variability across the Labrador Sea and its impact on phytoplankton composition is also discussed in Fragoso et al. (2016).**

RC2.11 -The Results should include a specific section on the temporal sequence, possibly exploring the sequence of events in each region. I note in Fig 3 that the data for 2012 and 2014, which were sampled late in the season, differ from other years, particularly Chl and nutrients in the central region.

**AC2.11 - Unfortunately, we do not feel that there is not enough information to provide a true temporal sequence of data or the succession of phytoplankton community composition. However, Fig. 2b does shows the temporal variability in sampling period.**

RC2.12 -Discussion:
Much of the discussion about individual taxa in section 4.1 should be first described in the Results section.

**AC2.12 - Information about individual taxa has now been added to the Results section (lines 291 – 299, Fig. 3)**

RC2.13 -Most of sections 4.2 and 4.3 should be saved for another paper.

**AC2.13 – Here we do not agree with the reviewer (see initial comment AC 2.2 and 2.4) and have retained them in the revised manuscript.**

RC2.14 -The Discussion should focus specifically on the results of this paper in relation to the
Aim, only referring to other studies to provide context, generally in the style of "Our results match those of Smith and Jones…". Only then should the wider implications of the work be discussed, and there should be clear signals when the narrative extends beyond the current work. Much of this Discussion reads like a review. It was often difficult to determine whether the results being discussed were from this paper or from others.

**AC2.14 - We agree with the reviewer and have now considerably revised the discussion to focus on our aims and our results, and have improved the interpretation of our data in comparison to other studies.**

RC2.15 -Conclusions:

Most of the final paragraph seems more appropriate to the Introduction. The authors
may also consider any further research questions that arise from this study.

**AC2.15 - This last paragraph was removed from the conclusions.**

RC2.16 -Abstract:
I think the first sentence is redundant and that the second sentence should be extended to include the Aim. The abstract will require revision in line with the changes to the rest of the manuscript.

**AC2.16 - We have now changed the beginning of the abstract to reinforce the aims and the relevance of the study (line 14 – 18).**

RC2.17 -SPECIFIC COMMENTS:
Line 186 and Table 3: Lutein not used for chlorophytes? (Does the BIO method separate ZEA & LUT?) If not, Table 3 ZEA must be ZEA+LUT

**AC2.17 - The BIO method does not separate lutein and zeaxanthin so we have renamed it to Zea + Lut as suggested.**

RC2.18 -Lines 192-200 and Figure 2: I note that two of the categories include Hex but no Chlc3
– I assume this is a simplification of the text and diagram as this combination does not exist to my knowledge. Figure 2 is unnecessary and should be replaced with a table including all pigments.

**AC2.18 – This figure has now been moved to the supplemental material (Fig. S1). In this study, *Phaeocystis pouchetii* was not associated with 19-hex and was identified using Chl C3. This has previously been observed in the Labrador Sea (Stuart et al., 2000) and is stated in line 640.**

RC2.19 -Section 3.2: Did the authors try further subdivision of group C3b? This group is by far the biggest, it is widest spread across the S-T diagram (Fig 5a), and its composition is "mixed", yet Fig 4a shows major divisions within the group. Would these subdivisions distinguish communities that were more coherent in composition and habitat?

**AC2.19 - Cluster C3b had the highest level of internal Bray-Curtis similarity in terms of sample composition (i.e. samples in this group were more similar (73%) to one another than to other groups). Hence, we decided not to further divide it as we could in theory continue to subdivide until each subgroup contains very few samples.**

RC2.20 -Line 316: change "Phaeocystis (cluster B)" to "A community dominated by diatoms and Phaeocystis (cluster B)". This is an important consideration throughout the document —- e.g. lines 328, 329 – there is not a careful distinction between the cluster groups (communities) and the taxa comprising them. I would invent an acronym or abbreviation for each community to avoid this confusion.

**AC2.20 - Changes have now been added and updated here (line 331) and throughout the manuscript.**

RC2.21 -Line 527: The possibility that "diatom species from both Arctic and Atlantic waters varied intrinsically in pigment composition" can be supported by consulting Table 3 of this paper, where we see that they do.

**AC2.21 - This is true and we have cited the table showing the final matrix ratio, where fuco:chla varies among Artic and Atlantic diatoms (line 561). Diatom composition (polar versus Atlantic species) might be influencing these discrepancy and we have now added a line discussing this possibility (line 564 - 570).**

RC2.22 -Line 551: "chlorophytes were present in high concentrations on the Labrador Shelf, which may explain the discrepancy between these results." Some more details are required to constitute an explanation.

**AC2.22 - These sentences have been completely rewritten for clarification (line 605 – 610):**

RC2.23 -Table 5: This table should be augmented by information on the region in which each cluster is found, and the major taxonomic components.

**AC2.23 – We added the main taxonomic components to the table (See Table 5).**

RC2.24 - Also expressing the values like Temperature with standard errors is inappropriate. The values are not based on repeat measurements of a single parameter –e.g. Cluster 3b is listed as 3.4+/-0.2 C, but the actual range is from about -1.3 to +8, the widest of any group. I would be surprised if the standard error given is correct. Even if is, it is meaningless. This table should list the range for each cluster instead.

**AC2.24 – Presenting only the ranges for each cluster makes it very difficult to identify patterns of similarity between the environmental conditions associated with these clusters. To aid in interpretation we have now added standard deviations rather than standard errors to Table 5 and have also added a table with the parameter ranges (Table S2) to the Supplementary material.**

RC2.25 -Also: I didn't see any reference to the data for DT:(DT+DD) in text (nor was there any reference to how long the filters were held between sample collection and freezing. This should be < 5-10 min for this parameter to be valid).

**AC2.25 - We have now included information on the filtering time (<10 mins) in the methods (line 146).**

RC2.26 -Results: I did not notice any indication that the raw pigment data were to be included in
Supplementary Material or an online databank. I would hope that this will be the case to increase the value of this data set.

**AC2.26 - Data from Bedford Institute of Oceanography) are publically available online (http://www.dfo-mpo.gc.ca/science/data-donnees/biochem/index-eng.html). We are discussing with the co-authors the possibility of submitting additional data to PANGEA.**

RC2.27 -TECHNICAL COMMENTS:
Line 67 and throughout: References should be cited in order of date – oldest to newest

**AC2.27 - For Biogeosciences, the order can be based on relevance, as well as chronological or alphabetical listing, depending on the author's preference. This is state in the "Reference" tab in "Manuscript preparation guidelines for authors" (http://www.biogeosciences.net/for_authors/manuscript_preparation.html ).**

RC2.28 -Line 84: change "while" to "but"

**AC2.28 - This sentence was removed.**

RC2.29 -Line 118: inset "wide" after "km" (twice)

**AC2.29 – Inserted (lines 102, 103).**

RC2.30 -Line 123: change "fresh" to "low salinity". Rest of same paragraph: three water masses are described as "warm and salty" or "cold, low salinity" but other water masses lack these descriptions (parallel form required– see below). Also, is the warm arrow parallel to the Labrador Current in Fig 1 considered to be part of that current?

**AC2.30 - This whole paragraph has changed, although we have described the water masses in an orderly manner as suggested by the reviewer (lines 105 - 110). The light red arrow in Figure 1 represents the extended branch of the IC, which is a modified (cooled and freshened) water mass caused by lateral and vertical mixing along the Labrador slope. We have now clarified this information in the Figure legend (lines 992).**

RC2.31 -Line 177: The correct reference for the method ascribed to "Coupel et al. (2015)" is
Higgins et al (2011).

**AC2.31 - We have actually updated this reference to Wright et al., (2009) (line 176).**

RC2.32 -Line 316: Add "respectively" after "(IC)"?

**AC2.32 - Both communities were related to the Irminger Current (an Atlantic water mass) so it is unsuitable to add "respectively" as suggested. However, we have rewritten the end of this sentence for further clarification (line 330).**

RC2.33 -Line 325: Replace "respond strongly to" with "are associated with" and "spatial aspects of the data" with "environmental parameters"

**AC2.33 – Changed (line 352).**

RC2.34 -Line 331: The description of Fig 5b could hardly be more obscure: "In Atlantic waters, temporal aspects of the data were also observed (upper and lower right quadrants (Fig. 5b))." There is nothing in that figure that implies a temporal sequence. It was only when the Conclusions mentioned clear temporal differences that I searched the document for "temporal" to find what I had missed and came back to this figure. After some cross-referencing I realised that the description should have read, "In Atlantic waters (upper and lower right quadrants (Fig. 5b)), the phytoplankton community was composed of mixed taxa during May (orange circles), but became dominated by diatoms and dinoflagellates during the bloom in June (red circles), showing a clear temporal succession in these waters". More generally, the authors must not rely on the reader to discern what is in a figure. The reader is not familiar with the data and may not see what the author sees, or they may see something different. Whatever story exists in the figure, it must be stated clearly in text as part of the narrative. The figure supports the narrative, it does not replace it.

**AC2.34 - We have now clarified the temporal succession of the spring bloom in the Labrador Sea in the text and changed this sentence following the suggestion of the reviewer (lines 358 – 361).**

RC2.35 -Line 368: Replace "lower accessory pigments to TChla ratio" with "lower ratio of accessory pigments to Tchla"

**AC2.35 – Changed (line 397).**

RC2.36 -Line 369: Replace "(Fig. 7b). Furthermore, communities from warmer waters (Irminger
Current from Atlantic origin), particularly those co-dominated by diatoms and dinoflagellates had " with "(Fig. 7b) than communities from warmer waters (Irminger Current from Atlantic origin), particularly those co-dominated by diatoms and dinoflagellates which had"

**AC2.36 – Changed (line 395 – 399).**

RC2.37 - Line 376: Replace "µg C µg Chla h-1W m-2" with "µg C µg Chla h-1 W-1 m2" or " (Wm-2)-1 " Also line 378

**AC2.37 – All changed.**

RC2.38 -Lines 375 to 386. Sentences should be rearranged to "parallel form" i.e. talk about the same things in the same order for each case cited

**AC2.38 – We have rewritten this whole paragraph (see lines 404 – 416).**

RC2.39 -Line 392: Insert "Atlantic," before "Labrador"

**AC2.39 – We are not sure what the reviewer is referring to here.**

RC2.40 -Lines 437 – 450: Reads like a review. Note also that the paragraph starts with "Phaeo-cystis and diatoms… (Fragoso etal 2016)" but by line 441 it's "PRESUMABLY of Phaeocystis and diatoms (Fragoso etal 2016)". Also is "eastern central Labrador Sea" (line 437) equivalent to "West Greenland Current" (line 440)?

**AC2.40 - We have now changed the beginning of this paragraph. See lines 472 - 478.**

RC2.41 -Line 598: Add reference e.g Gieskes and Kraay (1983) Mar. Biol. 75, 179-185.

**AC 2.41 - Suggested reference added (line 653).**

RC2.42 -Line 886: remove "et al" ; page numbers = 78 – 80

**AC 2.42 – Changed (line 887).**

RC2.43 -Figure 2 is unnecessary and should be replaced with a table including all pigments.

**AC 2.43 - We believe that Figure 2 is key to the CHEMTAX analysis. However, we have now moved it to the supplemental material.**

RC2.44 -Figure 4b. The colours of the sectors would be much more easily interpreted if they made sense to a phycologist! Surely cyanobacteria = Cyan, chlorophytes = Dk Green, Prasinophytes = Lt Green, Phaeocystis = Brown, etc. (Leave diatoms white)

**AC 2.44 - Although we appreciate the colour selection of the reviewer, we have retained the original pastel colours in 4b as the colours suggested are already used elsewhere in Figure 4 (4a, 4c) and this  could confuse the reader.**

RC2.45 -Figure 4c. The single circle as a scale is ambiguous. Does the biomass relate to the diameter or the area of the circle? In any case it's difficult to judge. There should be a range of circles representing a biomass scale (if circles are to be used). Also I estimate that about 20% of the data points are hidden in this diagram as they underlie another circle. This could be solved by increasing the breadth of the figure or using vertical bars instead of circles. Could the fronts be marked for each year by dotted lines?

**AC 2.45 – We have now increased the breadth of Fig 4c and added a scale for the bubbles. Physical fronts are already discernible through sharp changes in phytoplankton community composition, as mentioned in line 319, whereas dotted lines would become confusing between adjacent years.**

RC2.46 -Figure 5: It would be good to see individual taxa plotted in such diagrams.

**AC 2.46 –We could add the "arrows" of individual taxa in the figure 5b, however, we decided to leave the figure unchanged as adding information on taxa would be confusing to interpret the main message of the figure, which is the**

**effect of environmental factors on distinct phytoplankton communities. That is because, diatoms, for example, are the dominant taxa in all communities (except cluster C3b), so the "diatom arrow" in just one direction could bias the interpretation. We are focused on the whole community rather than individual taxa.**

RC2.47 -Table 2 is unnecessary. The individual pigments are not part of the story – simply quote the references.

**AC 2.47 - We believe that this Table is important for the CHEMTAX analysis so we have it in the manuscript.**

RC2.48 -Table 3: The legend doesn't make it clear that the references cited provided the starting ratios from which these data were calculated. Cyanobacteria is misspelt.

**AC 2.48 - We have now clarified the legend of Table 3. "Cyanobacteria" has now been spelt correctly.**

RC2.49 -Table 4: The formatting is strange. It looks as if it should be split into A & B, horizontally.

**AC 2.49 - Table 4 has now been split horizontally into a) and b) for better clarity.**

Reviewer #3:

RC3.1 - The manuscript "Spring phytoplankton communities of the Labrador Sea (2005-2014): pigments signatures, photophysiology and elemental ratios" present a time series of pigments and nutrients data in the Labrador Sea from 2005 to 2014. The authors use the CHEMTAX method to interpret the pigment dataset in term of phytoplankton groups and then to describe the distribution of these phytoplankton groups. Oceanographic provinces of the Labrador Sea are identified using on physical and biogeochemical parameters as well as phytoplankton diversity. Several statistical approaches based on clustering, ordination plot and regression were used to link the distribution in time and space of the phytoplankton with the environmental parameters. Finally, several physiological parameters related to the phytoplankton communities were measured (P curves, POC/PON, POC/POC Chla) or extract from the pigments distribution (AP/Chla, photoprotective pigments). The physiological information is used to go further in the explanation of the link between the phytoplankton community's distribution and the environmental conditions.

General comments:
The introduction is not well structured and full of too heavy and unclear sentence.

**AC3.1 - We have now completely rewritten and reduced the introduction to provide a better focus.**

RC3.2 - But, the manuscript goes better in the result and discussion section. The results section is clear with a good choice of graph. Sometimes, it was difficult to get the point of the use of methods and the information that sort from some data.

**AC3.2 - We have now changed and improved the methods section for better clarification by adding further explanation on the use of the different methods to examine the data.**

RC3.3 - Finally, the discussion put together in a clear way all the information in the results section and brings interesting information to parameters that were of unclear utility in the result section. The authors highlight the specificity of the species and explained their success in the different regions and use well the comparison with the literature. I recommend important

change in the introduction to make it more fluent, to better extract the key information and topics of each sub-paragraph. The sentences are generally way too long and confusing.
Most of them could be cut in two parts. There are several mistakes on the use of superlative in the results section. The discussion is well conducted and uses interestingly the results

**AC3.3 – We thank the reviewer for their helpful comments and suggestions. The introduction has been shortened and sentences are now condensed throughout.**

RC3.4 - Specific comments by section
Introduction
L51: better to use "structure"

**AC3.4 – Changed (line 52).**

RC3.5 - L51: change the order to "functional role in the community"

**AC3.5 - We have removed "in the community" to avoid redundancy (line 52).**

RC3.6 - L 54 to 59: there is some redundancy with the lines 51-53

**AC3.6 - We have now changed/reduced the introduction and shortened the sentences, so this redundancy does not exist anymore.**

RC3.7 - L59 to 64: Unclear about the conservation or not of the stoichiometry. You said the "stoichiometry is consistent phylogenetically" and latter you mentioned, "they may vary (…) phenotypically within species". Be more precise on when the ratios are conserved or not.

**AC3.7 - This sentence has been rewritten (lines 56).**

RC3.8 - L70 "shelves and the basin"

**AC3.8 - This sentence has been removed.**

RC3.9 - L75-76: I don't think the interest to study the phytoplankton is to use it as an index of waters masses since simple parameters as temperature and salinity did a good job. It appears to me more important to highlight the possible importance of the biogeography on the biological pump, carbon export or the energy transfer to upper trophic level.

**AC3.9 - We agree with the reviewer and have rewritten these lines to focus on ocean biogeochemistry and marine ecosystems (lines 70).**

RC3.10 - L78-84: The same idea is repeated. Please reduce the size of the sentence, too much utilization of the conjunction "and".

**AC3.10 - This paragraph has been removed to shorten the introduction overall.**

RC3.11 -L82: could simplify "high-latitude Arctic/Atlantic waters" by "polar waters".

**AC3.11 - This paragraph has been removed.**

RC3.12 -L100: redundancy with the line 88-90

**AC3.12 - Lines 77 - 79 refers to analysis of pigments using the HPLC while line 85 refers to CHEMTAX analysis of pigment data; hence we do not see them as redundant.**

RC3.13 -L93: Please precise the concept of "functional cell size"

**AC3.13 - This sentence has been removed from the introduction.**

RC3.14 -L94-95: "assemblage dominance": wrong, it's the dominance of phytoplankton groups and not assemblages

**AC3.14 - This sentence has been removed from the introduction.**

RC3.15 -L95: remove "however"

**AC3.15 – Changed (line 79, 80).**

RC3.16 -L99: remove the comma.

**AC3.16 – Changed (line 84).**

RC3.17 -L107: "comprehensively understand" is a pleonasm.

**AC3.17 - The word "comprehensively" was removed and this last paragraph of the introduction has been rewritten.**

RC3.18 -L108-L111: you repeat the same information than the line 106-108.

**AC3.18 - This paragraph (last of introduction) has been reduced.**

RC3.19 -Methods

There is some confusion on the water composition of the Labrador Sea. Moreover the authors depicted as well deep and shallow currents and water masses. The authors should focus on the surface and sub-surface water-masses and circulation since the pigment dataset presented here concerned only the upper 10m.

**AC3.19 - We believe that the reviewer is referring to the Irminger Current (IC). The IC is described as a surface current (see Hauser et al., 2015; Yashayaev and Seidov, 2015), however the Western Greenland Current may occasionally "slide" over the IC in the central-eastern part of the Labrador Sea and form a "tongue" of fresh, cold and less dense water. The lateral advection of this tongue (i.e. how offshore it goes) varies inter-annually during spring. We have used a T-S diagram to discern these water masses (IC, Labrador Current and WGC, see Figure 5a). As the reviewer has noted there were some relatively warm (> 3°C) and salty (> 34) water found at the surface. We refer to this as part of the IC, although it might have been slightly modified due to the dynamic features of surface waters, which includes the influence of precipitation/evaporation, meltwater, riverine input and mesoscale eddies. Although the IC is "conserved" at mid-depth waters (200-600 m), it does reach the surface, however it becomes "modified" due to the factors mentioned.**

RC3.20 - L115: "transition zone between the Arctic and …"

**AC3.20 – Changed (lines 97, 98).**

RC3.21 - L115: Newfoundland is not really the southern boundary. The North Atlantic is the southern boundary.

**AC3.21 - We have now defined the limits of the Labrador Sea according to the International Hydrography Organization (line 100).**

RC3.22 - L119: The lower limit of the Greenland Shelf (ie 2500m) sounds very deep to characterize a shelf! I think you characterize the extension of the Greenland Current here.

**AC3.22 - We apologise for the confusion. We were referring to the Greenland shelf and slope and not just the shelf. We have corrected this now (line 102).**

RC3.23 - L122: remove "mostly"

**AC3.23 – This whole paragraph has changed (lines 105 – 110).**

RC3.24 - L122: The Irminger current is not the main water masses of the Labrador Basin since this current it is confined on the east and west borders of Labrador Basin at a mid-depth (200-600m). The Labrador Sea Water composes the water of the basin and their characteristics are mainly influenced by the winder convection with the deeper water masses (see the work of Yashayaev et al.).

**AC3.24 - We apologise for the confusion in this section. We were referring to surface hydrography only. As discussed above (AC 3.19), the WGC often "slides" over the IC, creating a broad and thin layer of fresh and cold water, usually observed in the central-eastern section of the AR7W transects. On the western part of the section the IC intrudes into upper waters. This is observed in the T-S diagrams when salty (> 34) and warm (> 3°C) waters of Atlantic origin are found at the surface. We have rewritten this paragraph for clarification (lines 112 – 123).**

RC3.25 - L123: There is no evidence than the cold fresh after originated from Arctic contribute substantially to the deep basin since the front between the basin and the shelf is very strong. Part of the VITALS program using gliders is actually studying the exchange between the basin and the Labrador Shelf (B. De young, J. Palter et al.).

**AC3.25 - We have changed this paragraph to aid clarification. We now refer to the upper Labrador Sea layers (< 200 m) that are comprised of waters originating from the North Atlantic (IC) and the Arctic (LC and WGC) (line 105). See the response above (AC3.23).**

RC3.26 - L134: "Data used in this study"

**AC3.26 – Changed (line 126).**

RC3.27 - L134: remove "from stations" and "repeat".

**AC3.27 – Changed (line 126).**

RC3.28 - L146: Choose between "surface" or "near-surface" and stick to it all along the manuscript.

**AC3.28 - We have chosen to refer to *surface* waters throughout the entire manuscript.**

RC3.29 - L155: Maybe add the underline word "Back in the laboratory, POC/PON samples…"

**AC3.29 – Changed (line 153).**

RC3.30 - L171: I think the good way to describe the CHEMTAX output is "relative abundance" instead of "ratios of abundance"

**AC3.30 – Changed (line 168).**

RC3.31 - L173: not clear if all the pigments ratios are from the literature.

**AC3.31 – Line 183 indicates that all pigment ratios are from the literature. We have now added a sentence to the legend of Table 3 mentioning the exact references for each pigment ratio.**

RC3.32 - L174: Please indicate how the algal groups present in the study area are identified.

**AC3.32 - The identification is described in full in Fragoso et al. (2016). We have now included this reference in this sentence of the manuscript (line 182). See below.**

RC3.33 - L187: remove "that"

**AC3.33 – Changed (line 196).**

RC3.34 - L190: explain here the purpose of the fourth-root transforamation.

**AC3.34 - An explanation has now been included (lines 197 - 201): "Due to the high abundance of diatoms in the data, we have decided to apply a fourth-root transformation to increase the importance of less abundant groups, which would allow us to better discerning the spatial-temporal patterns of the phytoplankton communities in the Labrador Sea."**

RC3.35 - L195: "higher" than what? Be careful to compare with something when you use a superlative.

**AC3.35 - This word was changed to "*high*" (line 206).**

RC3.36 - Results
L277: "less well stratified"…"at those stations where"

**AC3.36 – Words "less" and "those" were removed. (line 281).**

RC3.37 - L278: replace "during" by "in"

**AC3.37 – Changed (line 282).**

RC3.38 - L279: "more highly stratify": pleonasm again…

**AC3.38 - We have removed the word "*more*" from the sentence (line 284).**

RC3.39 - L281: "higher": then superlative to be compared with something.

**AC3.39 - We have removed the parenthesis in this sentence so it has changed to: "…POC:PON ratios were also higher > 8…" (line 286).**

RC3.40 - L288: Not clear if the "pairwise analysis" you mentioned refer to the ANISOM one-way pairwise?

**AC3.40 - We have changed the sentence to: "*Pairwise one-way analysis* of similarity (*ANOSIM*) between clusters…" (line 302).**

RC3.41 - L289: too long sentence, please reduce or cut in two parts. Parentheses are at the wrong place.

**AC3.41 - We have now split the sentence into two (line 302 - 305).**

RC3.42 - L298: "especially" is useless here. In general, there is an over utilisation of adverbs in the text (mostly/especially…).

**AC3.42 - The word "especially" was removed from this sentence (line 312).**

RC3.43 - L313: superlative!! No subject of comparison…

**AC3.43 - We have rewritten this whole paragraph. See comment below (AC.3.44).**

RC3.44 - L315: superlative again. Wrong use.

**AC3.44 - We have rewritten this paragraph (lines 326 – 332).**

RC3.45 - L321-324: Too long sentence make it confusing. Separate in two sentences?

**AC3.45 - This sentence was split into two (lines 336 – 340).**

RC3.46 - L340: The table 4 is difficult to understand and could earn a better presentation.

**AC3.46 - We have now reorganised Table 4, separating it into Table 4a and Table 4b. Further explanation is given in lines 336 – 350 and in the revised legend of Table 4 (Line 962).**

RC3.47 - L345: there is a problem, the title is the same than 3.3!!

**AC3.47 - The title has now been updated to "Phytoplankton distributions and elemental stoichiometry".**

RC3.48 - L344-352: Please present the POC-PON relationships somewhere.

**AC3.48 - We are not sure what the reviewer means by this comment, but POC:PON relationships are shown in Figure 6a and has been referred to in line 372.**

RC3.49 - L354: Please quickly explain the purpose of calculating the relationships between POCphyto and POC:PON.

**AC3.49 - We have now added a short explanation of the purpose of studying the relationships between POCphyto and POC:PON (lines 375 – 377).**

RC3.50 - L359: I would say, "…contribute for a high proportion…"

**AC3.50 – Changed (line 382).**

RC3.51 - L362: superlative lower (use low or compare to something).

**AC3.51 - We have now included an object of comparison in this sentence (lines 385).**

RC3.52 - Discussion

L392: as noted earlier in the manuscript, the surface phytoplankton didn't growth in the Irminger water since this water mass is observed only the slope and at great depth.

**AC3.52 - In the central-eastern part of the Labrador Sea, the IC is found below the WGC "tongue", as the reviewer mentioned. However, in the central-western region the IC is found at the surface so phytoplankton do grow in these different water masses (IC, LC, WGC). Phytoplankton species found in the IC are usually found in Atlantic waters, while polar species are found in the LC and WGC (see Fragoso et al 2016).**

RC3.53 - L396-397: Here the concept of ecological succession should be better presented. Is the variation between a deep and shallow mixed layer associated to the season or the two conditions (shallow/deep mixed layer) can be observed at the same time of the year?

**AC3.53 – This whole paragraph has been rewritten to clarify the seasonal and temporal patterns of phytoplankton communities (lines 420 – 431).**

RC3.54 - L401-403: A link is missing between this information and the above sentence.

**AC3.54 - This sentence has been rewritten for clarification (lines 433 – 438).**

RC3.55 L406: "often" and "as well" mean the same here. Please remove one of the two.

**AC3.55 -We have changed the word "often" to "occasionally" to clarify the sentence (lines 441).**

RC3.56 L470: I would prefer to use the mean POCphyto rather than POC>…The latter formulation is not really comparable since we don't know the dispersion of the data.

**AC3.56 -The spread of the data of POCphyto/total POC and POC:PON ratios are shown in Figure 6c and 6b, respectively. We have now specifically referred to the figure in the text (lines 505 – 508).**

RC3.57 - L475: were also abundant

**AC3.57 - We are not sure what the reviewer is referring to here.**

RC3.58 - L512-519: It should be interesting to explain the meaning of the AP/TChla ratio in term of strategy for the adaptation to light regime.

**AC3.58 - Few studies have examined this in any depth and hence we can conclude very little in the present study. AP/TChla ratio varied according to community composition and species adaptation to light environments, mixing regimes, competition for light with other dissolved substances (etc) could explain the observed trend. Further in depth physiological work is really needed to full elucideate the meaning of the variability in AP/TChla. We have extended the discussion a little bit in the paper in attempt to explain why such trend is observed (lines 571-574).**

RC3.59 - L522-523: Conufsing because you introduce "two parameters" and after you cite three parameters (Nitrate, Silicate and SI).

**AC3.59 - The word "two" has been removed from this sentence (lines 558).**

RC3.60 - L540-552: You show interesting difference in the photophysiological characteristics of phytoplankton, especially between the west and east communities. Near Greenland, the communities is composed of species resistant to high light while on the Labrador Shelf, the species are less resistant to photo-inhibition. Is the light conditions are so different between east

and west to explain these different adaptations to light? It could be interesting to describe these difference in the light regimes between the two side of the Labrador Sea. The latter melting of the ice cover on the Labrador Shelf could be an explanation?

**AC3.60 - We have now improved the discussion about the influence of PAR in separating the polar phytoplankton communities observed. See the rewritten paragraphs (lines 586 – 610).**
RC3.61 - L555 to 558: The sentence is confusing. It takes time for me to understand that dinoflagellates bloom in May to avoid

higher light levels. Please rephrase or separate in two sentences to improve the clarity.

**AC3.61 - The beginning of this paragraph has been rewritten for better elucidation (lines 614 – 617).**

Hauser, T., Demirov, E., Zhu, J., Yashayaev, I., 2015. North Atlantic atmospheric and ocean inter-annual variability over the past fifty years – Dominant patterns and decadal shifts. Prog. Oceanogr. 132, 197–219. doi:10.1016/j.pocean.2014.10.008

[revised manuscript text omitted]

The other main requirement of the One of the main assumptions of the CHEMTAX method is that pigment ratios remain

310 constant across the subset of samples that are being analysed (Mackey et al 1996) (Swan et al., 2015). To satisfy this

assumption, *a priori* analysis was performed, where pigment data were sub-divided into groups using cluster analysis (Bray-Curtis similarity) and each group was processed separately by the CHEMTAX program (Table 3; for the final ratio matrix, see supplemental material). This approach was used because distinct phytoplankton communities have been observed in the Labrador Sea (Fragoso et al., 2016) so that the ratio of accessory pigment to chlorophyll *a* probably varies within different

315 water masses across the Labrador Sea (LC, IC and WGC). Pigment concentration data (BUT19But-fuco, Hex-fucoHEX19,

ALLOXAllo, Chl *a*CHLA, Chl *b*CHLB, Chl *c₃*CHLC3, FucoFUCOX, PERIDPeri, PrasPRASINO and Zea + LutZEA) 
[revised manuscript text omitted]
| Salinity | $33.4 \pm 0.4$ | (17) | $33.7 \pm 0.1$ | (46) | $33.1 \pm 0.2$ | (62) | $34.1 \pm 0.1$ | (92) | $34.4 \pm 0.1$ | (32) | $33.0 \pm 0.8$ | (4) |
| MLD (m) | $32.2 \pm 10.6$ | (17) | $32.6 \pm 3.4$ | (46) | $31.2 \pm 3.6$ | (62) | $59 \pm 7.4$ | (92) | $29.8 \pm 3.0$ | (32) | $16.0 \pm 2.1$ | (4) |
| SI × $10^{-3}$ (kg m$^{-4}$) | $9.1 \pm 1.5$ | (17) | $6.3 \pm 0.8$ | (46) | $10.7 \pm 1.1$ | (62) | $5.0 \pm 0.7$ | (92) | $6.1 \pm 0.8$ | (31) | $6.6 \pm 4.3$ | (4) |
| NO$_3^-$ (µmol L$^{-1}$) | $2.9 \pm 1.1$ | (17) | $2.7 \pm 0.5$ | (46) | $3.4 \pm 0.6$ | (58) | $8.4 \pm 0.5$ | (83) | $3.7 \pm 0.7$ | (32) | $3.8 \pm 3.4$ | (4) |
| Si(OH)$_4$ (µmol L$^{-1}$) | $2.2 \pm 0.7$ | (17) | $2.8 \pm 0.3$ | (46) | $3.5 \pm 0.3$ | (58) | $5.4 \pm 0.2$ | (83) | $3.0 \pm 0.4$ | (32) | $2.3 \pm 1.7$ | (4) |
| PO$_4^{3-}$ (µmol L$^{-1}$) | $0.3 \pm 0.1$ | (17) | $0.3 \pm 0$ | (45) | $0.4 \pm 0$ | (55) | $0.7 \pm 0$ | (79) | $0.3 \pm 0$ | (32) | $0.4 \pm 0.2$ | (4) |
| Si(OH)$_4$:NO$_3^-$ | $6.0 \pm 3.2$ | (14) | $3.6 \pm 1.3$ | (37) | $8.5 \pm 2.5$ | (54) | $1.1 \pm 0.2$ | (82) | $1.6 \pm 0.3$ | (32) | $3.9 \pm 2.2$ | (4) |
| NO$_3^-$:PO$_4^{3-}$ | $8.2 \pm 2.0$ | (11) | $5.2 \pm 0.7$ | (45) | $5.9 \pm 0.8$ | (55) | $11.4 \pm 0.5$ | (79) | $8.7 \pm 0.8$ | (32) | $5.5 \pm 3.5$ | (4) |
| Chlorophyll $a$ (mg Chl$a$ m$^{-3}$) | $3.8 \pm 1.1$ | (17) | $5.5 \pm 0.7$ | (45) | $7.7 \pm 0.7$ | (59) | $2.0 \pm 0.2$ | (91) | $4.0 \pm 0.3$ | (31) | $8.8 \pm 4.8$ | (4) |
| DT:(DT+DD) | $0.01 \pm 0.006$ | (16) | $0.02 \pm 0.01$ | (44) | $0.04 \pm 0.01$ | (62) | $0.10 \pm 0.01$ | (92) | $0.08 \pm 0.01$ | (32) | $0.02 \pm 0.02$ | (4) |
| (DD+DT):Chl$a$ | $0.08 \pm 0.02$ | (17) | $0.03 \pm 0.004$ | (46) | $0.04 \pm 0.003$ | (62) | $0.07 \pm 0.004$ | (92) | $0.12 \pm 0.01$ | (32) | $0.07 \pm 0.02$ | (4) |
| POC (mg C m$^{-3}$) | $245 \pm 45$ | (4) | $498 \pm 38$ | (27) | $533 \pm 30$ | (45) | $234 \pm 18$ | (63) | $512 \pm 46$ | (15) | $393 \pm 296$ | (2) |
| PON (mg N m$^{-3}$) | $39 \pm 8$ | (4) | $65 \pm 4$ | (27) | $74 \pm 4$ | (45) | $38 \pm 3$ | (64) | $83 \pm 9$ | (15) | $42 \pm 29$ | (2) |
| POC$_{phyto}$ (%) | $23.0 \pm 2.6$ | (4) | $49.2 \pm 5.8$ | (26) | $60.9 \pm 3.9$ | (44) | $33.3 \pm 1.3$ | (64) | $36.0 \pm 3.0$ | (15) | $37.8 \pm 0.9$ | (2) |
| POC:PON | $6.5 \pm 0.6$ | (4) | $7.8 \pm 0.4$ | (27) | $7.5 \pm 0.3$ | (45) | $6.6 \pm 0.2$ | (64) | $6.2 \pm 2.0$ | (15) | $8.6 \pm 1.1$ | (2) |
| $\alpha^B$ × $10^{-2}$ (µg C µg Chl$a$ h$^{-1}$ W m$^{-2}$) | - | - | $6.8 \pm 2$ | (9) | $9.2 \pm 2$ | (10) | $7.1 \pm 1$ | (18) | $7.1 \pm 1$ | (4) | - | - |
| $P_m^B$ (µg C µg Chl$a$ h$^{-1}$ W m$^{-2}$) | - | - | $2.9 \pm 0.4$ | (9) | $2.3 \pm 0.3$ | (10) | $2.3 \pm 0.1$ | (18) | $3.2 \pm 0.4$ | (4) | - | - |
| $E_k$ (W m$^{-2}$) | - | - | $60 \pm 11$ | (9) | $29 \pm 4$ | (10) | $38 \pm 3$ | (18) | $46 \pm 3$ | (4) | - | - |
| $E_s$ (W m$^{-2}$) | - | - | $62 \pm 11$ | (9) | $35 \pm 6$ | (10) | $43 \pm 4$ | (18) | $56 \pm 4$ | (4) | - | - |
| $\beta$ × $10^{-4}$ (µg C µg Chl$a$ h$^{-1}$ W m$^{-2}$) x10 | - | - | $4 \pm 2$ | (9) | $16 \pm 7$ | (10) | $10 \pm 4$ | (18) | $29 \pm 10$ | (4) | - | - |

**FIGURE LEGENDS**

[revised manuscript text omitted]

a)

[Figure]

b)

[Figure]

c)

[Figure]

1350

**Figure 5**

[Figure]

1355    **Figure 6**

[Figure]

**Figure 7**

[Figure]

Total chlorophyll *a* (mg TChl*a* m⁻³)

---

## Editor Decision (ED1)

Reviewer 1

In general, the authors have improved the manuscript by incorporating the comments suggested by the reviewers. The new introduction and methodological sections and the use of the standardized abbreviations for pigments make the manuscript easier to read in its present form. The initial matrices used for the CHEMTAX analyses and the RMS are now presented. However I still have a few comments on the manuscript in its present form:

Abstract
Line 26, do the authors mean 70% of total chl a (TChla)?

Methods
Line 140, first time referring to chlorophyll a, add the abbreviation after chlorophyll a (chl a) and use the abbreviation through the rest of the manuscript. Please check through the manuscript for usage of Tchla or chl a (e.g. line 254, the abbreviation should be Tchla). Similarly, when the authors refer to the photosynthetically active radiation (PAR, line 481), please first define the abbreviation then use it through the manuscript, not the other way around.

Line 143, add extracted "in the dark" for 24 h-

Line 152, remove samples and change to chl a fluorescence of water samples was determined on board after…

Line 186, do the authors refer to the daily incident irradiance (integrated irradiance) averaged per month? Unclear.

Table 2
Chl c3: diatoms type 2 contain this pigment (see Higgins et al. 2011, Coupel et al. 2015).
TChl a row change the Chlide "alpha" to Chlide a.
Last row of the table, please change CHLA to Chl a.

Table 3
Why are chlorophytes (CHLORO-1) and haptophytes (HAPTO-6) in capital letters?

According to Higgins et al. 2011, dinoflagellates containing peridinin should be named type 1, please change Dinoflagellates to dino-1 in your matrices. The use of dinoflagellates and then dino-2 is confusing otherwise.

Why did you assume that the diatoms present in your study did not contain chl c3? I think you should re-consider this. At least for the communities that are located in the shelves where the authors found high concentration of fucoxanthin together with high chl c3 and diatoms dominated the community.
Seeing the variety of diatom species found in the previous study (lines 565-570), it seems reasonable to think that some of the chl c3 to the total chl a should be attributed to the presence of diatoms type 1 (containing chl c3) and not only to diatoms type-2.

Line 621, change Front with Font. If you cite this work you need to add the reference in the list.

Spring phytoplankton communities of the Labrador Sea (2005- 2014): pigment signatures, photophysiology and elemental ratios

Glaucia M. Fragoso, Alex J. Poulton, Igor M. Yashayaev, Erica J. H. Head , Duncan A. Purdie

GENERAL COMMENTS

This paper has been greatly improved from the previous version. Issues that other reviewers picked up issues with pigment analysis and CHEMTAX appear to have been sorted and I am happy with responses to my comments for the most part. I reiterate my earlier comment that this is a useful study that should be published, but unfortunately my major concern, the structure of the manuscript, is not quite there yet – in particular, the Aims are wrong* – they are too vague and do not fully cover the material presented. They should be re-written as clear EXPLICIT goals for the paper, and other sections should be checked to ensure that the findings are clearly reported. A simple strategy is suggested below that would greatly improve the paper.

*I make that provocative assertion deliberately to focus the following discussion and hopefully (if I may be so bold) to offer some practical guidance to a young scientist. Given that the previous version lacked an aim, and the problems with the aim of the current version (discussed below), I think a fuller explanation of the problem is warranted in the hope of improving subsequent papers as well as this one. Of course I know that in this case, the Aim was written after the rest of the paper, which is exactly what not we are supposed to do. Ideally the Aim (or far preferably, the Hypothesis) should be rigorously defined before designing the experiment, and the content of the resultant paper(s) should be planned as part of the research proposal. It makes the papers much easier to write. Of course that doesn't always happen, and it certainly didn't in many of my experimental papers. Often for surveys or monitoring programs, such as in the current study, the story is discovered as part of the analysis. Nevertheless the paper must be structured as clearly as if the Aim or Hypothesis was clear from the outset.

The Aim or Hypothesis is the single most important statement in a paper – not only does it dictate the content of the entire paper, but it also tells the reader what to expect, and primes him or her to recognise the importance of results as they are presented and the relevance of discussion. Thus it is critical that it is VERY carefully considered, explicit, and crystal clear.

In my review of the previous version of this manuscript, I commented: *"This paper desperately needs a clear Aim to provide a basis for a narrative, to dictate what is included in (or excluded from ) the paper, to provide a focus for the Results, Discussion and Conclusions, and by which to judge the success of the project."*

The authors responded: **AC2.3 - Clear aims have now been added in the last paragraph of the introduction and the Results, Discussion and Conclusions all link to these aims.**

These aims are set out in the Introduction (lines 88-93):

**The aim of this study is to provide a baseline description of the current distributions and biogeochemical traits of phytoplankton communities from distinct biogeographical regions of the**

**Labrador Sea. …….. In addition, we also examine the overall photophysiological and biogeochemical traits associated with these different phytoplankton communities.**

So what is wrong with this Aim? First, it is not clear. It is jargon – general hand-waving. Secondly, it is not explicit. What exactly is meant by terms like baseline description? What photophysiological and biogeochemical traits are examined? (Note the repetition of biogeochemical as well). These general terms should be replaced by explicit reference to P-E curves, fluorometry, POC/PON analysis etc. (many photophysiological and biogeochemical traits were NOT examined!). Thirdly, it is incomplete. There is no mention of the analysis of hydrographic variables that explain the distribution of phytoplankton communities, and thus technically these sections are irrelevant to the paper. In this regard, the Aims have failed the authors because if such an aim was explicitly identified, then they would have mentioned the fact that environmental variables explained 99.8 % of the variability in phytoplankton communities in the Conclusions and the Abstract, which they did not. Thus, what I consider to be an important result of the paper will not be apparent if I read the Title, Abstract, keywords, Aim and Conclusions.

My suggestion for this type of paper, involving post-hoc analysis of a data set, is to set out the Aims as a series of detailed questions that will be addressed in the paper. A useful strategy is to set out, in plain language, what questions the authors asked themselves when they were setting up the statistical analyses presented. Did they say, "Can we provide a baseline description of the current distributions and biogeochemical traits of phytoplankton communities from distinct biogeographical regions of the Labrador Sea"? I doubt it. How about, "Are there distinct communities of phytoplankton within the study region and if so, what are their main constituents? Where do these communities occur and are they stable year-to-year? Can the variability in phytoplankton communities be explained by environmental factors? *Et cetera*"? I hope it is obvious that clear explicit aims like these make it easier for the reader to understand the paper (and thus more likely to cite it), but also make it easier for the authors to write a clear, focused paper in the first place.

(By the way, for my students, I normally suggest writing the questions for each test clearly while doing the analyses, as well as their results. Doing it at the time while immersed in the data simplifies the write-up. I also recommend that the statement of Aims in the Introduction should be followed by a brief paragraph describing the approach taken and another outlining the major findings.)

This can be fixed fairly easily without much rewriting. I suggest going through the paper, listing the major results, framing them as questions in the Aims, ensuring that the answers are clear in the Results, and highlighted in the Conclusions and Abstract, then getting on with the next paper and thesis!

I would discard the first sentence of the Aims (Line 88). Start the para, "Here, we investigate the multi-year (2005-2014) distributions" etc using the second and third sentences. Even though I complained about the third sentence, it can be used because it will be explained and expanded in the following sentence: "We address the following questions:" or similar, using the questions as above. The further hand-waving in the final sentence should be replaced by a brief synopsis of the major findings: e.g." We show that several distinct communities exist … etc. "

More generally, beware of jargon and nominalization – they reduce readability and clarity.

OTHER COMMENTS

1. In my previous review, I asked: *RC2.19 -Section 3.2: Did the authors try further subdivision of group C3b? This group is by far the biggest, it is widest spread across the S-T diagram (Fig 5a), and its composition is "mixed", yet Fig 4a shows major divisions within the group. Would these subdivisions distinguish communities that were more coherent in composition and habitat?* The authors responded: **AC2.19 - Cluster C3b had the highest level of internal Bray-Curtis similarity in terms of sample composition (i.e. samples in this group were more similar (73%) to one another than to other groups). Hence, we decided not to further divide it as we could in theory continue to subdivide until each subgroup contains very few samples.** This is the authors' decision of course. My question was raised out of curiosity. However, I will make the more general point that with this sort of analysis, I suggest exploring the patterns and fully understand what they are showing about the data, rather than sticking to arbitrary cutoffs. Then, having understood the data, decide what groupings are appropriate to address the aims of the paper.

2. Line 74:  Change "has" to "have"

3. Line 154" "pelletised" misspelt

4. Line 215 and elsewhere: There is a problem with the statement "Zea + Lut is not only found in prasinophytes–type 2, but is also the major accessory pigment of cyanobacteria". It is important to note that Zeaxanthin and Lutein are two separate pigments, despite the fact that they cannot be resolved in the author's HPLC system and elute as a single peak. It is wrong to say THE major accessory pigment of cyanobacteria, first because Zea + Lut is not a single pigment, but also because Cyanobacteria have Zeaxanthin but no Lutein. Likewise, it would be better to say Zea is also a minor pigment in chlorophytes, while Lutein is often the dominant carotenoid (Line 216). Note also that Zeax may be derived from non-photosynthetic bacteria (e.g. Flavobacteria)

5. Line 323: "in the Labrador Sea during spring and early summer (2005-2014)" could be replaced by "defined above"   (this redundant detail just makes it harder to read)

6. Line 331: Shouldn't the sigma theta be greater than 27 kg m-3 rather than less than?

7. Line 352: Comma after variance

8. Line 604: Presumably "relies" should not be italicised

9. Line 621: Alou-Front et al (2016) is not in References, and presumably should be Alou-Font

10. Line 750: Gieskes and Kraay (1983) missing

---

## Author Response (AR2)

UNIVERSITY OF
Southampton

**20**th January, 2017 Editor, **Biogeosciences**

Dear Editor:

I, with my co-authors, would like to resubmit our article entitled "*Spring phytoplankton communities of the Labrador Sea (2005-2014): pigment signatures, photophysiology and elemental ratios*" for publication in **Biogeosciences**. We are pleased that the reviewers (Simon Wright and Reviewer #1) are now satisfied with most of our revisions. The paper has been further revised in accordance with both reviewers' suggestions and the pending issues raised have now been addressed. A sheet detailing the changes made is included in this letter as well as in the revised manuscript (with and without track changes).

Thank you for your consideration, and we hope the paper is now acceptable.

Sincerely,

Glaucia Fragoso

Glaucia Fragoso, PhD
Ocean and Earth Science, National Oceanography Centre Southampton
Email: glaucia.fragoso@noc.soton.ac.uk

Ms. Ref. No.: bg-2016-295
Title: Spring phytoplankton communities of the Labrador Sea (2005-2014): pigment signatures, photophysiology and elemental ratios. Resubmitted to Biogeosciences.

**We thank the two reviewers for their comments and suggestions, which we feel have greatly improved the manuscript. Below we respond to each comment in detail. RC refers to "Reviewer's Comments" and AC to "Author's comments". We have enumerated the reviewer's comments to organise our responses.**

Reviewer #1:

RC1.1: In general, the authors have improved the manuscript by incorporating the comments suggested by the reviewers. The new introduction and methodological sections and the use of the standardized abbreviations for pigments make the manuscript easier to read in its present form. The initial matrices used for the CHEMTAX analyses and the RMS are now presented. However I still have a few comments on the manuscript in its present form:

Abstract
Line 26, do the authors mean 70% of total chl a (TChla)?

**AC1.1 – Yes. We have now added "*a*" after total chlorophyll in the sentence (line 24).**

RC1.2 - Methods
Line 140, first time referring to chlorophyll a, add the abbreviation after chlorophyll a (chl a) and use the abbreviation through the rest of the manuscript. Please check through the manuscript for usage of Tchla or chl a (e.g. line 254, the abbreviation should be Tchla).
**AC1.2 – We have now defined the types of chlorophyll *a* measurements included in the manuscript and the abbreviation for each of them: chlorophyll *a* pigment (Chl *a*) (Table 2), fluorometric chlorophyll *a* (Chl*a*) (line 147) and HPLC-based total chlorophyll *a*, including chlorophyllide (TChl*a*) (line 195) and changed the abbreviations in the text accordingly.**

RC1.3 - Similarly, when the authors refer to the photosynthetically active radiation (PAR, line 481), please first define the abbreviation then use it through the manuscript, not the other way around.

**AC1.3 – Changed. Abbreviation of PAR is defined in line 193, where it is mentioned for the first time.**

RC1.4 - Line 143, add extracted "in the dark" for 24 h-

**AC1.4 – Changed. Line 150.**

RC1.5 - Line 152, remove samples and change to chl a fluorescence of water samples was determined on board after…

**AC1.5 – We rewrote the sentence as "Chlorophyll *a* concentrations were determined fluorometrically after 24 h of extraction…". Line 159.**

R.C1.6 Line 186, do the authors refer to the daily incident irradiance (integrated irradiance) averaged per month? Unclear.

**AC1.6 – We refer to the daily incident irradiance averaged over each month. We have now added this information in the text (line 193).**

RC1.7 - Table 2
Chl c3: diatoms type 2 contain this pigment (see Higgins et al. 2011, Coupel et al. 2015).

**AC1.7 – We used Vidussi et al. (2004) as a reference for diatoms in this study. Also, Coupel et al. (2015) did not allocate Chl c3 to diatoms, although Higgins et al. (2011) specifically referred to a second diatom group. A further explanation of why we have not considered the pigment ratio for diatoms according to Higgins et al. (2011) is included in comment AC1.12 (see below).**

RC.1.8 - TChl a row change the Chlide "alpha" to Chlide a.

**AC1.8 – Changed.**

RC.1.9 - Last row of the table, please change CHLA to Chl a.

**AC1.9 – Changed.**

RC1.10 - Table 3
Why are chlorophytes (CHLORO-1) and haptophytes (HAPTO-6) in capital letters?

**AC1.10 – We decided to keep the same terminology used in other references (where the pigment ratios were taken from) because we believe it is easier for the reader to go back to the original reference if needed. Higgins et al. (2011)**

refer to chlorophytes, such as *Dunaliella tertiolecta*, as CHLORO-1 (page 269 of Roy et al., 2011) and HAPTO-6, as *Emiliana huxleyi* (page 281 of Roy et al., 2011).

RC1.11 - According to Higgins et al. 2011, dinoflagellates containing peridinin should be named type 1, please change Dinoflagellates to dino-1 in your matrices. The use of dinoflagellates and then dino-2 is confusing otherwise.

**RC1.11 – Although it is true that dinoflagellates that contain peridinin are termed type 1 by Higgins et al. (2011), the initial pigment ratio for this group was taken from another reference - Coupel et al. (2015), who term them "dinoflagellates", as well as Vidussi et al. (2004) where the pigment ratio in Coupel et al. (2015) originally came from. We are concerned that if we call it dino-1 it would infer that the pigment ratio was from Higgins et al. (2011), which is not true. For this reason, we have decided to keep the terminology consistent with the source of our pigment ratios. The methods section explains clearly the difference between dinoflagellates and dino-2.**

RC1.12 - Why did you assume that the diatoms present in your study did not contain chl c3? I think you should re-consider this. At least for the communities that are located in the shelves where the authors found high concentration of fucoxanthin together with high chl c3 and diatoms dominated the community.
Seeing the variety of diatom species found in the previous study (lines 565-570), it seems reasonable to think that some of the chl c3 to the total chl a should be attributed to the presence of diatoms type 1 (containing chl c3) and not only to diatoms type-2.

**AC1.12 – We agree with the reviewer that some of the chl c3 could come from diatoms because of the high diversity and abundance of this group found on the shelves. However, the initial pigment ratio selected for diatoms in this study (Vidussi et al. 2004) was carefully selected to avoid the underestimation of *Phaeocystis pouchetii*, given that Chl c3 has been shown to be the main marker of *Phaeocystis* in boreal waters (see explanation in lines 240 and 650) and this species was very abundant in the microscopic counts shown by Fragoso et al. (2016). Thus, making the changes suggested by the reviewer could result in a significant underestimation of *Phaeocystis* in our study. Moreover, the correlation between algal biomass (carbon estimated from microscopic counts versus CHEMTAX-derived algal chlorophyll *a*) for *Phaeocystis* ($r^2 = 0.79$) and diatoms ($r^2 = 0.74$) (line 639) confirms that the pigment ratios used were appropriate in this study.**

RC1.13 - Line 621, change Front with Font. If you cite this work you need to add the reference in the list.

**AC1.13 – Changed (lines 607, 631, 701).**

Reviewer #2, Simon Wright:

RC2.1 - GENERAL COMMENTS

This paper has been greatly improved from the previous version. Issues that other reviewers picked up issues with pigment analysis and CHEMTAX appear to have been sorted and I am happy with responses to my comments for the most part. I reiterate my earlier comment that this is a useful study that should be published, but unfortunately my major concern, the structure of the manuscript, is not quite there yet - in particular, the Aims are wrong* - they are too vague and do not fully cover the material presented. They should be re-written as clear EXPLICIT goals for the paper, and other sections should be checked to ensure that the findings are clearly reported. A simple strategy is suggested below that would greatly improve the paper.

**AC2.1 – We appreciate and have accepted the reviewer's suggestions (see comments below).**

RC2.2 - *I make that provocative assertion deliberately to focus the following discussion and hopefully (if I may be so bold) to offer some practical guidance to a young scientist. Given that the previous version lacked an aim, and the problems with the aim of the current version (discussed below), I think a fuller explanation of the problem is warranted in the hope of improving subsequent papers as well as this one. Of course I know that in this case, the Aim was written after the rest of the paper, which is exactly what not we are supposed to do. Ideally the Aim (or far preferably, the Hypothesis) should be rigorously defined before designing the experiment, and the content of the resultant paper(s) should be planned as part of the research proposal. It makes the papers much easier to write. Of course that doesn't always happen, and it certainly didn't in many of my experimental papers. Often for surveys or monitoring programs, such as in the current study, the story is discovered as part of the analysis. Nevertheless the paper must be structured as clearly as if the Aim or Hypothesis was clear from the outset. The Aim or Hypothesis is the single most important statement in a paper - not only does it dictate the content of the entire paper, but it also tells the reader what to expect, and primes him or her to recognise the importance of results as they are presented and the relevance of discussion. Thus it is critical that it is VERY carefully considered, explicit, and crystal clear.

In my review of the previous version of this manuscript, I commented: *"This paper desperately needs a clear Aim to provide a basis for a narrative, to dictate what is included in (or excluded from) the paper, to provide a focus for the Results, Discussion and Conclusions, and by which to judge the success of the project."*

The authors responded: **AC2.3 - Clear aims have now been added in the last paragraph of the introduction and the Results, Discussion and Conclusions all link to these aims.**

These aims are set out in the Introduction (lines 88-93):

**The aim of this study is to provide a baseline description of the current distributions and biogeochemical traits of phytoplankton communities from distinct biogeographical regions of the Labrador Sea ......... In addition, we also examine the overall photophysiological and biogeochemical traits associated with these different phytoplankton communities.**

So what is wrong with this Aim? First, it is not clear. It is jargon - general hand-waving. Secondly, it is not explicit. What exactly is meant by terms like baseline description? What photophysiological and biogeochemical traits are examined? (Note the repetition of biogeochemical as well). These general terms should be replaced by explicit reference to P-E curves, fluorometry, POC/PON analysis etc. (many photophysiological and biogeochemical traits were NOT examined!). Thirdly, it is incomplete. There is no mention of the analysis of hydrographic variables that explain the distribution of phytoplankton communities, and thus technically these sections are irrelevant to the paper. In this regard, the Aims have failed the authors because if such an aim was explicitly identified, then they would have mentioned the fact that environmental variables explained 99.8 % of the variability in phytoplankton communities in the Conclusions and the Abstract, which they did not. Thus, what I consider to be an important result of the paper will not be apparent if I read the Title, Abstract, keywords, Aim and Conclusions.

My suggestion for this type of paper, involving post-hoc analysis of a data set, is to set out the Aims as a series of detailed questions that will be addressed in the paper. A useful strategy is to set out, in plain language, what questions the authors asked themselves when they were setting up the statistical analyses presented. Did they say, "Can we provide a baseline description of the current distributions and biogeochemical traits of phytoplankton communities from distinct biogeographical regions of the Labrador Sea"? I doubt it. How about, "Are there distinct communities of phytoplankton within the study region and if so, what are their main constituents? Where do these communities occur and are they stable year-to-year? Can the variability in phytoplankton communities be explained by environmental factors? *Et cetera"?*

**AC2.2 – We have now fully followed the reviewer's suggestions. We have been through our result section again and listed all the potential specific questions relevant to our main findings. We have, however, avoided a long list, shortened these questions and added them to the end of the introduction to keep it short and direct (line 90).**

RC 2.3 - I hope it is obvious that clear explicit aims like these make it easier for the reader to understand the paper (and thus more likely to cite it), but also make it easier for the authors to write a clear, focused paper in the first place. (By the way, for my students, I normally suggest writing the questions for each test clearly while doing the analyses, as well as their results. Doing it at the time while immersed in the data simplifies the write-up. I also recommend that the statement of Aims in the Introduction should be followed by a brief paragraph describing the approach taken and another outlining the major findings.)

**AC2.3 – We have added a few lines after the questions describing our approach and highlighting our main findings (Line 96).**

RC2.4 - This can be fixed fairly easily without much rewriting. I suggest going through the paper, listing the major results, framing them as questions in the Aims, ensuring that the answers are clear in the Results, and highlighted in the Conclusions and Abstract, then getting on with the next paper and thesis!

**AC2.4 – As stated above, we have now framed the questions based in our results and highlighted our main findings in the conclusions (line 671).**

RC2.5 - I would discard the first sentence of the Aims (Line 88). Start the para, "Here, we investigate the multi-year (2005-2014) distributions" etc using the second and third sentences. Even though I complained about the third sentence, it can be used because it will be explained and expanded in the following sentence: "We address the following questions:" or similar, using the questions as above. The further hand-waving in the final sentence should be replaced by a brief synopsis of the major findings: e.g." We show that several distinct communities exist ... etc. " More generally, beware of jargon and nominalization - they reduce readability and clarity.

**AC2.5 – We have discarded the first sentence of the aims (line 87) and included the research questions (line 90) and a brief synopsis of the main findings (line 96).**

OTHER COMMENTS

RC2.6 - 1. In my previous review, I asked: *RC2.19 -Section 3.2: Did the authors try further subdivision of group C3b? This group is by far the biggest, it is widest spread across the S-T diagram (Fig 5a), and its composition is "mixed", yet Fig 4a shows major divisions within the group. Would these subdivisions distinguish communities that were more coherent in composition and habitat?* The authors responded: **AC2.19 - Cluster C3b had the highest level of internal Bray-Curtis similarity in terms of sample composition (i.e. samples in this group were more similar (73%) to one another than to other groups). Hence, we decided not to further divide it as we could in theory continue to subdivide until each subgroup contains very few samples.** This is the authors' decision of course. My question was raised out of curiosity. However, I will make the more general point that with this sort of analysis, I suggest exploring the patterns and fully understand what they are showing about the data, rather than sticking to arbitrary cutoffs. Then, having understood the data, decide what groupings are appropriate to address the aims of the paper.

**AC2.6 – We appreciate the reviewer's suggestions and we will take it into consideration in future publications.**

2. Line 74: Change "has" to "have"

**AC2.3 - Changed. Line 73.**

3. Line 154" "pelletised" misspelt

**AC2.4 - Changed. Line 161.**

4. Line 215 and elsewhere: There is a problem with the statement "Zea + Lut is not only found in prasinophytes-type 2, but is also the major accessory pigment of cyanobacteria". It is important to note that Zeaxanthin and Lutein are two separate pigments, despite the fact that they cannot be resolved in the author's HPLC system and elute as a single peak. It is wrong to say THE major accessory pigment of cyanobacteria, first because Zea + Lut is not a single pigment, but also because Cyanobacteria have Zeaxanthin but no Lutein. Likewise, it would be better to say Zea is also a minor pigment in chlorophytes, while Lutein is often the dominant carotenoid (Line 216). Note also that Zeax may be derived from non-photosynthetic bacteria (e.g. Flavobacteria)

**AC2.5 – We have now added a sentence that clarifies that Zea and Lut are different pigments that were co-eluted by the methods used in this study (line 197). We have rewritten the following sentences for clarification: "Zea is not only found in "prasinophytes type 2", but is also the major accessory pigment of cyanobacteria (such as *Synechococcus* spp.) who have been observed in the Labrador Sea (particularly in Atlantic waters; Li et al., 2006). Zea is also a minor pigment in chlorophytes, while Lut is often the dominant carotenoid in this group (MacIntyre et al., 2010; Vidussi et al., 2004)", (line 223).**

5. Line 323: "in the Labrador Sea during spring and early summer (2005-2014)" could be replaced by "defined above" (this redundant detail just makes it harder to read)

**AC2.6 - Changed. Line 333.**

6. Line 331: Shouldn't the sigma theta be greater than 27 kg m-3 rather than less than?

**AC2.7 - Changed. Line 340.**

7. Line 352: Comma after variance

**AC2.8 – Changed. Line 362.**

8. Line 604: Presumably "relies" should not be italicised

**AC2.9 – Changed. Line 614.**

9. Line 621: Alou-Front et al (2016) is not in References, and presumably should be Alou-Font

**AC2.10 – Changed. Lines 607, 631, 701.**

10. Line 750: Gieskes and Kraay (1983) missing

**AC2.11 – Reference added. Line 768.**

[revised manuscript text omitted]

Distance from fixed reference position (km)

1075

**Figure 3**

[Figure]

 **Figure 4**

[Figure]

**Figure 5**

[Figure]

**Figure 6**

[Figure]

**Figure 7**

[Figure]

1100